# Granularity-Aware Adaptive Classifier Expansion via Zero-Shot Learning

**Xiangyu Wang** [1]  **Yanze Gao** [1]  **Changxin Rong** [1]  **Lyuzhou Chen** [1]  **Xiren Zhou** [1]  **Huanhuan Chen** [1]

## Abstract

Zero-shot classifier expansion aims to recognize unseen classes by learning a shared mechanism to map semantics of all classes to classifier weights without access to images. However, existing methods rely on a shared mapping, which is difficult to classify in scenarios containing a mixture of distinct and similar classes, especially with the continuous expansion of classes. Since this mapping prioritizes general attributes for distinct classes while neglecting subtle attributes for similar ones, this granularity mismatch, compounded by sensitivity to noise, induces optimization interference where gradients from distinct classes dominate the learning process. To overcome this limitation, a granularity-aware adaptive framework with interventions is introduced to balance them. Specifically, this method first generates multi-source semantics by intervening on non-causal noise, then discovers latent class structure to separate distinct classes, and finally refine similar classes to synthesize weights with invariance to non-causal noise. The effectiveness is demonstrated through theoretical and empirical analysis in multiple aspects.

## 1. Introduction

Expanding classifiers to recognize continuously emerging classes is a fundamental and urgent challenge for machine learning systems, especially in deployed visual recognition and open-world scenarios (Zhou et al., 2025). While traditional methods such as model retraining offer potential solutions, the recurrent costs of data collection and annotation for each emerging class often render them inefficient (Wang et al., 2024). To overcome these data dependencies, Zero-Shot Learning (ZSL) is introduced to recognize unseen classes by transferring knowledge from seen classes (Xian et al., 2018). Recently, this paradigm has been extended to

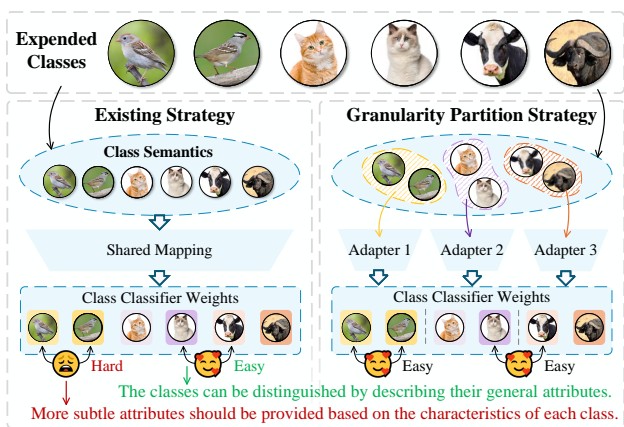

*Figure 1.* Comparison of weight generation mechanisms between the existing shared mapping and our granularity partition strategy.

image-free settings, employing a shared mechanism to map all class semantics to classifier weights via the same pattern (Christensen et al., 2023). This allows for the synthesis of weights from semantics, without accessing any images.

Although these methods eliminate the dependence on visual samples, real-world scenarios driven by continuous class expansion typically involve a mixture of visually distinct and similar classes, which poses a demand on discriminative capability at different levels of granularity across classes. However, existing methods rely on a shared mapping from semantics to weights, which struggles to meet this demand. Specifically, they tend to prioritize general attributes to distinguish distinct classes, while failing to adequately capture subtle attributes specific to individual classes for distinguishing similar classes[1], as depicted in Figure 1. Furthermore, with continuous class expansion, the accumulating general attributes suffer from severe homogenization and attribute overlap[2], while optimization interference driven by the gradient dominance of distinct classes intensifies, making it difficult to balance the generalization and discrimination.

To overcome this limitation of indiscriminate fitting caused by the shared mapping, a promising approach is to seek an

---

[1]School of Computer Science and Technology, University of Science and Technology of China, Hefei, China. Correspondence to: Huanhuan Chen <hchen@ustc.edu.cn>.

*Proceedings of the $43^{rd}$ International Conference on Machine Learning*, Seoul, South Korea. PMLR 306, 2026. Copyright 2026 by the author(s).

---

[1]Facial structure serves as a general attribute for distinguishing distinct (i.e., not easily confused) classes between cattle and cats, but it is not sufficient for easily confused classes (Sparrow species). Head patterns are the key subtle attribute for similar classes.

[2]More detailed theoretical analysis is shown in Appendix D.2.

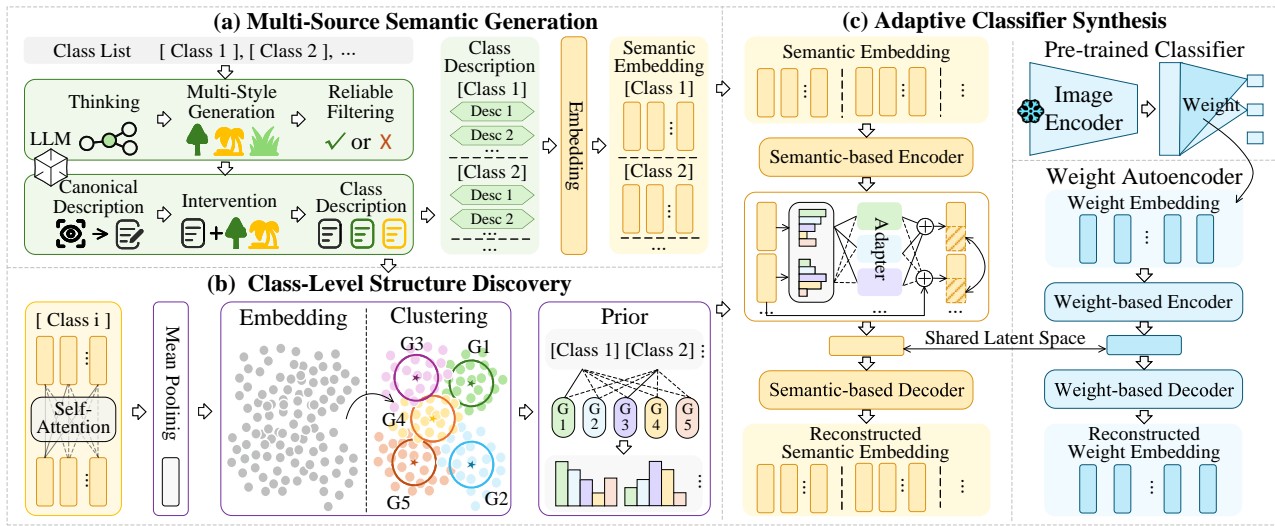

*Figure 2.* The overall architecture of the proposed framework. The figure details the data flow among the Multi-Source Semantic Generation (MSG) module, the Class-Level Structure Discovery (CSD) module, and the Adaptive Classifier Synthesis (ACS) module.

adaptive granularity partitioning strategy to balance distinct and similar classes. The general try is to utilize a hierarchical mechanism to decompose the original shared mapping into multiple subproblems, adaptively adjusting to the varying granular needs of different classes (Abdelsalam et al., 2021). By leveraging this capability, the learning objectives of distinct and similar classes can be separated, effectively blocking the dominant gradient propagation from distinct classes, thus enabling the model to capture subtle attributes while preventing confusion between irrelevant classes.

However, implementing this strategy is challenging. First, the model struggles to adaptively distinguish which attributes serve as effective discriminators, because general attributes are effective for distinct classes while becoming redundant information for similar classes. Therefore, indiscriminately mixing these attributes leads to an extremely high attribute density, where few subtle attributes are overwhelmed by abundant general attributes. Second, different confused class pairs rely on different subtle attributes, where attributes crucial for distinguishing one confused pair may be irrelevant noise for others. Therefore, fitting them in a shared parameter space prevents discriminative attributes needed for specific classes from being effectively refined. Finally, attributes focused on at different granularity levels need to be effectively fused into the weight space. This process requires guiding different subspaces to be optimized along different directions, while mitigating the decision bias that arises from focusing solely on subtle perspectives.

To address these challenges, a granularity-aware adaptive framework is proposed, as shown in Figure 2. First, a Multi-Source Semantic Generation (MSG) module is designed to generate robust semantics, which introduces controllable interventions to sever the interference of background noise

on attributes, enabling few subtle details to be preserved and enhanced. Next, a Class-Level Structure Discovery (CSD) module adaptively infers inter-class confusion relationships and partitions the high-density attribute space into different clusters, providing learnable structural parameters and separating distinct classes. Finally, an Adaptive Classifier Synthesis (ACS) module assigns adapters to each cluster and maps semantics into the weight space. By providing more parameter space, this module amplifies the representation of subtle attributes in the weight space and optimizes towards effective directions. The contributions are summarized as:

- This paper proposes a granularity-aware strategy that transforms the shared mapping into granular structures, separating demands for general and subtle attributes.

- An adaptive discrimination mechanism is introduced to capture subtle attributes in crowded spaces, balancing generalization breadth and discrimination precision.

- An automated pipeline is designed to adaptively perceive granularity demands with the continuous class expansion, maintaining robust decision boundaries.

## 2. Related Work

### 2.1. Classifier Expansion via Zero-Shot Learning

Zero-Shot Learning (ZSL) aims to recognize unseen classes by transferring knowledge from seen classes through auxiliary semantic information, such as attributes or text descriptions (Xian et al., 2017). Common ZSL methods mainly follow embedding-based paradigms, which align visual features and semantic representations within a shared latent space for compatibility measurement, as well as attribute-based models that infer labels via predicted attributes (Chen

et al., 2022; Liu et al., 2023). However, these traditional paradigms typically require accessing visual samples or expensive labeling costs when adapting to newly emerging classes, which renders them impractical for deployment in open-world scenarios (Tang et al., 2024).

To remove the dependency on unseen visual data, image-free ZSL has been proposed, where unseen class classifiers are synthesized directly from semantic information (Christensen et al., 2023). Early works generate unseen classifiers via linear combinations or semantic relations among seen classes (Mensink et al., 2014). Subsequent approaches learn explicit semantic-to-weight mappings using regression or regularization, while recent studies exploit generative modeling or large language models to enrich semantic representations (Xu et al., 2022; Lyu et al., 2025). However, these methods rely on the shared mapping, ignoring latent class structure and relative density between attributes, which prevents subtle attributes from being resolved. In contrast, our method addresses these limitations by introducing granularity-aware adaptation to handle general and subtle attributes separately.

### 2.2. Zero-Shot Adaptation over Granularity Partition

Adaptive learning serves as a fundamental paradigm to overcome the inherent rigidity of monolithic models, offering the flexibility to adjust according to varying input complexities (Atzmon & Chechik, 2019). To achieve this capability, adapter-based and Mixture of Experts (MoE) architectures have been widely applied to handle diverse data distributions (Yu et al., 2024a; Xin et al., 2024). Specifically, recent studies like CLIP-Adapter (Gao et al., 2024) have successfully applied to visual tasks, and the studies (Lu et al., 2024; Wang et al., 2025) have further evolved into unified adapters, which can dynamically adapt to different modalities or tasks by learning context-aware weight modulation. Concurrently, MoE models generalize this idea by introducing gating mechanisms to decompose objectives into subproblems assigned to specialized expert networks (Yu et al., 2024b). These methods have proven effective in scaling model capacity and resolving conflicts.

Despite their success in visual recognition and pre-training, extending them to image-free zero-shot classifier expansion is challenging. Typically, the adaptation process in vision tasks is driven by discriminative visual cues to guide expert selection (Li et al., 2025). However, the challenge in image-free ZSL shifts to inferring the processing strategy solely from semantics, as visual signals are unavailable. Moreover, while standard partitioning strategies can expand model capacity to handle data diversity, they generally operate under a uniform granularity assumption (Raposo et al., 2024). In contrast, our method explores the granularity level which discovers differences between distinct and similar classes, and introduces adapters to distinguish similar classes.

## 3. Method

This section first formalizes the task as learning a mapping from semantics to classifier weights, and then sequentially constructs three modules: Multi-Source Semantic Generation (MSG) to mitigate observational bias via causal intervention, Class-Level Structure Discovery (CSD) to identify coarse-grained structural divisions, and Adaptive Classifier Synthesis (ACS) to refine fine-grained decision boundaries.

### 3.1. Task Formulation

Let $S$ and $U$ denote the disjoint sets of seen and unseen classes, respectively, where $S \cap U = \emptyset$. A pre-trained classification model $\Phi : \mathcal{X} \to S$ is considered, which can be decomposed into a visual feature extractor $f$ and a classification weight matrix $W_S \in \mathbb{R}^{d_v \times |S|}$, where $\mathcal{X}$ represents the image space and $d_v$ represents the feature dimension. The objective of zero-shot classifier expansion is to expand the pre-trained classifier to recognize unseen classes $U$ without accessing any visual samples. Specifically, it involves learning a weight synthesis function $\Psi : T \to W$ that maps semantic descriptions $T$ to the weight space $W$.

The function $\Psi$ is optimized using the paired descriptors and weights of seen classes $\{(t_s, w_s)|s \in S\}$ during training. Then, the learned $\Psi$ synthesizes weights for unseen classes $W_U = \{\Psi(t_u)|u \in U\}$ during inference, which are finally injected into the original model, forming an expanded classifier with parameters $W_S \cup W_U$ to recognize test images. Unlike previous methods that $\Psi$ is considered as a shared mapping for all class semantics, which forces a compromise between coarse-grained separation and fine-grained discrimination, this paper aims to construct a granularity-aware adaptive $\Psi$ that explicitly models the latent semantic structure and focuses on core attributes for class discrimination.

### 3.2. Multi-Source Semantic Generation

To construct robust class semantics to prevent attributes from being disturbed by non-causal noise, the paper first filters out reliable attribute ontology to generate a class-level semantic canonicity and then performs controlled interventions on the descriptive styles and environments, explicitly exposing and mitigating observational bias. An automated and effective approach involves utilizing Large Language Models (LLMs) $\mathcal{M}$ and Chain-of-Thought (CoT) principles to generate a set of multi-source semantic descriptions. By stripping away spurious correlations, this ensures that subsequent modules can capture core attributes across different granularity levels.

**Canonical Anchoring with Reliable Attribute Ontology.** The semantic generation process starts with the construction of a universal attribute ontology, which can be described as a shared set of views $V = \{v_i\}_{i=1}^n$ to characterize all

classes $Y = S \cup U$. To mitigate the semantic drift caused by LLM hallucinations during automatic pipeline generation, a set of reliable attribute ontologies is required to strictly ground textual descriptions in visual reality. Specifically, for each class $y \in Y$ and attribute ontology $v_i$, the co-occurrence confidence $\hat{p}(v_i|y)$ is estimated by performing multiple Monte Carlo samplings of the LLMs' generative distribution[3], counting times that the attribute concept of class $y$ is assigned to the ontology $v_i$. Thus, the conditional probability of each class on each ontology is expressed as:

$$\hat{p}(y \mid v_i) = \frac{\hat{p}(v_i \mid y)\, p(y)}{\sum_{y' \in Y} \hat{p}(v_i \mid y')\, p(y')}, \quad (1)$$

where $p(y)$ is the prior probability of class $y$ (uniform).

Subsequently, the selection of ontologies is ranked based on transferability and discriminability (Ye et al., 2025b), both of which depend on the entropy $h(v_i)$ of the class distribution corresponding to the attribute ontology $v_i$:

$$h(v_i) = -\sum\nolimits_{y \in Y} \hat{p}(y \mid v_i) \log \hat{p}(y \mid v_i). \quad (2)$$

These two properties can be respectively represented as:

$$H_t(v_i) = \frac{h(v_i)}{\sum_{v_j \in V} h(v_j)}, \;\; H_d(v_i) = \frac{h_{\max} - h(v_i)}{h_{\max}}. \quad (3)$$

Here, $h_{\max}$ denotes the maximum entropy[4], i.e., $\log|Y|$. By jointly considering the confidence score and harmonic mean, which is calculated by the above properties, multiple optimal values are selected to form a set of reliable attribute ontologies $O_y \subset V$ for each class $y$.

To generate a canonical description $t_y$, serving as the stable foundation of class $y$ to objectively reflect visual facts, a meta-prompt strategy is introduced to concatenate attributes $A_y$ generated from $O_y$ into coherent semantics without applying specific linguistic styles or adding external information. Formally, the generation is defined as $t_y = \mathcal{M}(A_y, I)$, where $I$ represents the instruction for the fact-based composition (see Appendix C.1 for more details). The result $t_y$ encapsulates the invariant visual identity of class $y$, providing a standardized baseline for subsequent causal interventions.

**Interventional Semantic Expansion.** Once the canonical description $t_y$ has been fixed for each class, the paper treats it as a controllable starting point rather than the final representation, since it cannot robustly handle perturbations and is insufficient to support the semantic space. The semantic observation process can be formalized as a Structural Causal Model (SCM), where the generated description

$T$ is an outcome of intrinsic attributes $A$, confounded by the linguistic style $G$ and environment $E$. Since a single semantic represents only one biased realization of the distribution $P(T|A, G, E)$, interventions are performed on them to disentangle the class identity from spurious correlations, $P(T|A, \mathrm{do}(G), \mathrm{do}(E))$, where "do" denotes interventions[5].

Specifically, two intervention sets are defined, including a style set $G = \{g_a\}_{a=1}^{m_1}$ (e.g., scientific, distinct) to vary the linguistic modality, and an environment set $E = \{e_b\}_{b=1}^{m_2}$ (e.g., natural, wild) to simulate diverse visual contexts. Taking the canonical anchor as causal root, the intervened semantic is generated by prompting a rewrite of $t_y$ under specific intervention constraints $(g_a, e_b)$, denoted as:

$$\bar{t}_y^{a,b} \leftarrow \mathcal{M}(t_y \mid A = A_y, \mathrm{do}(G = g_a), \mathrm{do}(E = e_b)). \quad (4)$$

Iterating through all combinations in $G \times E$ yields a set of multi-source semantics $T_y = \{t_y\} \cup \{\bar{t}_y^{a,b}\}_{a,b}$ for class $y$, detailed in Appendix C.2. To interface with the subsequent modules, a pre-trained text encoder $\mathcal{E}(\cdot)$ is applied to embed the semantic $T_y$ into a vector $\mathbf{F}_y \in \mathbb{R}^{m \times d_e}$, expressed as:

$$\mathbf{F}_y = [\mathbf{f}_{y,1}, \bar{\mathbf{f}}_{y,2}, \dots, \bar{\mathbf{f}}_{y,m}]^T, \quad m = |G \times E| + 1, \quad (5)$$

where $d_e$ is the embedding dimension. This generation process effectively expands the semantic representation from a single point estimate into a set of robust semantic representations, covering the diverse manifold of potential observations while preserving the core causal mechanism.

### 3.3. Class-Level Structure Discovery

Traditional shared mapping functions typically treat the semantic space as a flat manifold, ignoring the granularity mismatch between distinct and similar classes. To adaptively uncover latent coarse-grained relationships among classes and provide structural guidance for fine-grained weight synthesis, this paper first fuses the multi-source semantics $\mathbf{F}_y$ into a stable representation $\hat{\mathbf{F}}_y$ and then discovers coarse-grained structural priors in the semantic space, providing a learnable partitioning for subsequent modules.

**Semantic Fusion.** The interventional expansion generates a diverse representation $\mathbf{F}_y$ for each class, containing varying linguistic structures and potential noise. To capture the mutual dependencies among descriptions and suppress outlier instances that deviate from the semantic consensus, a lightweight fusion module is employed, expressed as:

$$\hat{\mathbf{F}}_y = \mathrm{MP}\left(\mathrm{Softmax}\left(\frac{\mathbf{F}_y W_Q (\mathbf{F}_y W_K)^T}{\sqrt{d_a}}\right)(\mathbf{F}_y W_V)\right), \quad (6)$$

where $d_a$ is the attention dimension and $W_Q, W_K, W_V \in \mathbb{R}^{d_e \times d_a}$ are learnable matrices. "MP" represents the mean pooling, yielding a fused representation $\hat{\mathbf{F}}_y \in \mathbb{R}^{d_a}$.

---

[3]High-frequency recurrence is interpreted as factual robustness for initial filtering, quantifying the reliability of $v_i$ for class $y$.

[4]The discriminability is characterized as the relative reduction in class uncertainty before and after introducing the ontology.

[5]This paper adopts a structured nuisance-controlled formulation as a conceptual tool for exposing nuisance-induced variations.

**Coarse-grained Structural Prior.** The fused representation $\hat{\mathbf{F}}_y$ is first projected onto $\mathbf{F}'_y$ on the unit hypersphere $\mathbb{S}^{d_a-1}$ via L2 normalization. Spherical K-means, based on cosine similarity, is then applied to partition them into $K$ clusters[6]. However, this non-differentiable characteristic hinders end-to-end optimization. Therefore, the paper treats the obtained clustering structure as initialization and explicitly introduces a set of learnable cluster centers $\{\mu_k\}_{k=1}^K$ embedded within the semantic space. Specifically, class representations are generated by a K-component von Mises-Fisher (vMF) mixture model, with each cluster having its own center and shape. The overall probability density of the semantic representation $\mathbf{F}'_y$ can be represented as:

$$p(\mathbf{F}'_y) = \sum_{k=1}^K \pi_k C_{d_a}(\kappa_k) \exp(\kappa_k \mu_k^T \mathbf{F}'_y), \quad (7)$$

where $\kappa_k > 0$ is the concentration parameter, $\pi_k$ is the mixing coefficient, and $C_{d_a}(\kappa_k)$ is the normalization constant.

Since visual concepts often exhibit polysemous characteristics, hard assignment of clusters to downstream modules struggles to handle ambiguous class attribution at cluster boundaries. This paper adopts a soft routing strategy by calculating the posterior probability of class $y$ belonging to the semantic cluster $k$, serving as a routing signal $r_{y,k}$:

$$r_{y,k} = p(k|\mathbf{F}'_y) = \frac{\pi_k C_{d_a}(\kappa_k) \exp(\kappa_k \mu_k^T \mathbf{F}'_y)}{\sum_{j=1}^K \pi_j C_{d_a}(\kappa_j) \exp(\kappa_j \mu_j^T \mathbf{F}'_y)}. \quad (8)$$

The routing vector $\mathbf{r}_y = [r_{y,1}, r_{y,2}, \ldots, r_{y,k}]$ encodes the coarse-grained structural prior to subsequent adaptation.

### 3.4. Adaptive Classifier Synthesis with Invariance

The final classifier weights are synthesized by harmonizing the general knowledge with the discrimination of subtle attributes. This module formulates the shared semantic-to-weight mapping as a global backbone, refined with a set of lightweight residual adapters modulated by the structural prior $\mathbf{r}_y$. Building on this, the multi-source semantic representation $\mathbf{F}_y$ drives the mapping towards the intervention-stable core factors, focusing on the discriminative attributes of classes rather than superficial spurious variations.

**Fine-grained Adaptive Residual.** To resolve the tension between inter-class generalization and intra-class discrimination, the paper first trains a shared global mapping, consisting of a semantic autoencoder $(\mathcal{E}_\mathcal{T}, \mathcal{D}_\mathcal{T})$ and a weight autoencoder $(\mathcal{E}_\mathcal{W}, \mathcal{D}_\mathcal{W})$, to capture general visual patterns across all classes (Christensen et al., 2023). The training objective includes two autoencoder reconstruction losses and two cross-modal losses, detailed in Appendix C.3.

To refine decision boundaries within specific semantic clusters and ensure that the operating space is comparable[7], each cluster performs local tuning through a residual adapter $\mathcal{R}_k(\cdot)$, which can be designed as a lightweight bottleneck layer. Formally, the encoded latent representation $\mathbf{z}_{y,j}$ for the multi-source semantic $\bar{\mathbf{f}}_{y,j}$ of class $y$ is represented as:

$$\mathbf{z}_{y,j} = \sum_{k=1}^K r_{y,k} \left( \alpha \mathbf{h}_{y,j} + (1-\alpha)\mathcal{R}_k(\mathbf{h}_{y,j}) \right), \quad (9)$$

where $\mathbf{h}_{y,j}$ is the vector after global encoding, i.e., $\mathcal{E}_\mathcal{T}(\bar{\mathbf{f}}_{y,j})$, $\alpha$ is used to adjust the degree, and $r_{y,k}$ denotes the soft membership derived from Equation (8). This combination allows the generator to adapt to the collaboration of multiple branches and mitigate the inaccuracy of structural priors.

**Causal Invariance Constraint.** Although the vector $\mathbf{z}_{y,j}$ captures the mapping at granularity, the mapping itself may still contain spurious correlations. To satisfy the causal requirement that changing the observational noise should not alter the intrinsic identity of classes, a causal invariance constraint is imposed. Specifically, $\mathbf{z}_{y,j}$ is projected in two parts through mappings $\mathcal{P}_c$ and $\mathcal{P}_n$, yielding causal identity $\mathbf{z}_{y,j}^c$ and non-causal variation $\mathbf{z}_{y,j}^n$, which are then concatenated along the sequence dimension to form $\mathbf{Z}_{y,j} = [\mathbf{z}_{y,j}^c || \mathbf{z}_{y,j}^n]$. The paper designs a domain-swapping operation in the latent space, keeping the causal identity unchanged and swapping the non-causal part for the same class[8], which can be expressed as $\mathbf{Z}_{y,j\to j'} = [\mathbf{z}_{y,j}^c || \mathbf{z}_{y,j'}^n]$. Therefore, the decoder $\mathcal{D}_\mathcal{W}$ is forced to map these varying latent representations to a consistent functional neighborhood in the weight space by minimizing the causal invariance loss:

$$\mathcal{L}_{\text{inv}} = \sum_{y \in S} \mathbb{E}_{j \neq j'} \left( d(\mathcal{D}_\mathcal{W}(\mathbf{Z}_{y,j}), \mathcal{D}_\mathcal{W}(\mathbf{Z}_{y,j\to j'})) \right), \quad (10)$$

where $d(\cdot)$ denotes the cosine distance. This constraint penalizes sensitivity to non-causal linguistic or contextual variations, focusing solely on intrinsic visual attributes.

**Training and Classifier Synthesis.** By training on paired descriptors and weights in seen classes and descriptors only in unseen classes, the objective is to minimize the sum of the semantic, weight reconstruction, cross-modal, and invariance losses (Appendix C.4). During inference, the model takes the semantic $\mathbf{F}_u \in \mathbb{R}^{m \times d_e}$ as input and synthesizes unseen classifier weights $w_u$, which can be represented as:

$$w_u = \frac{1}{m} \sum_{j=1}^m \mathcal{D}_\mathcal{W}([\mathcal{P}_c(\mathcal{E}'_\mathcal{T}(\mathbf{F}_{u,j})) || \mathcal{P}_n(\mathcal{E}'_\mathcal{T}(\mathbf{F}_{u,j}))]), \quad (11)$$

where $\mathcal{E}'_\mathcal{T}(\cdot)$ represents the global encoder and adapter. The synthesized weights are ultimately injected into the original model $\Phi$ in the generalized ZSL (GZSL) to classify test images. The handling of new classes that do not have semantic vectors during training is discussed in Appendix C.5.

---

[6] We search over $\mathcal{K} = [\lfloor \sqrt{|Y|}/2 \rfloor, \lceil \sqrt{|Y|} \rceil]$ and choose the $K$ with the highest cosine-aware silhouette score, followed by only a small local adjustment that remains stable (Section 4.5).

[7] In contrast, assigning an independent encoder to each cluster hinders cross-cluster geometric consistency and parameter sharing.

[8] What would happen if the class retained its original semantics but its style or context were replaced by that of another instance?

*Table 1.* Comparative top-1 accuracy **T1** (%) and harmonic mean **H** of the proposed framework with other methods across three datasets.

| Dataset | | CUB | | | | AWA2 | | | | SUN | | | |
|---|---|---|---|---|---|---|---|---|---|---|---|---|---|
| Method | | T1 | U | S | H | T1 | U | S | H | T1 | U | S | H |
| **Expert-based** | ConSE | 41.3±0.2 | 0.4±0.0 | 87.9±0.2 | 0.8±0.1 | 45.0±0.6 | 2.9±0.3 | 96.2±0.1 | 5.7±0.5 | 43.8±0.6 | 0.1±0.0 | 48.7±1.2 | 0.2±0.1 |
| | COSTA | 33.3±0.2 | 0.0±0.0 | 87.9±0.2 | 0.0±0.0 | 45.9±0.6 | 0.0±0.0 | 96.3±0.1 | 0.0±0.0 | 18.9±0.5 | 0.0±0.0 | 52.3±0.1 | 0.0±0.0 |
| | SubReg | 47.6±1.7 | 0.5±0.3 | 87.9±0.2 | 1.0±0.6 | 53.5±2.6 | 0.0±0.0 | 96.3±0.1 | 0.0±0.0 | 48.8±0.7 | 0.0±0.0 | 52.3±0.1 | 0.0±0.0 |
| | wDAE | 49.5±3.3 | 0.9±0.4 | 87.4±0.1 | 1.8±0.9 | 52.9±4.5 | 0.0±0.0 | 96.1±0.1 | 0.1±0.1 | 49.8±0.4 | 0.0±0.0 | 50.8±0.5 | 0.1±0.1 |
| | VGSE | 45.9±0.8 | 38.2±0.8 | 60.9±3.1 | 46.9±0.6 | 53.8±0.5 | 37.5±1.1 | 83.0±2.3 | 51.6±0.7 | 43.8±0.5 | 41.8±0.5 | 10.6±1.8 | 16.8±2.4 |
| | ICIS | 60.3±1.0 | 47.6±1.6 | 70.8±1.0 | 56.9±0.9 | 62.9±1.9 | 33.5±4.4 | 93.5±2.0 | 49.3±4.4 | 51.7±0.7 | 41.7±2.0 | 28.6±2.9 | 33.9±1.3 |
| **LLM-generated** | ConSE | 45.2±0.6 | 0.4±0.1 | 87.9±0.2 | 0.8±0.1 | 50.4±0.6 | 2.4±0.2 | 96.2±0.1 | 4.7±0.4 | 40.2±0.6 | 0.1±0.1 | 49.7±0.2 | 0.2±0.1 |
| | COSTA | 40.3±0.4 | 0.0±0.0 | 87.9±0.2 | 0.0±0.0 | 57.1±0.4 | 0.0±0.0 | 96.3±0.1 | 0.0±0.0 | 30.4±1.1 | 0.0±0.0 | 52.3±0.1 | 0.0±0.0 |
| | SubReg | 59.6±1.3 | 1.2±0.2 | 87.9±0.2 | 2.4±0.4 | 63.1±0.4 | 0.3±0.0 | 96.2±0.1 | 0.7±0.0 | 52.8±3.9 | 1.3±0.3 | 52.2±0.1 | 2.5±0.7 |
| | wDAE | 57.6±0.7 | 2.5±0.4 | 87.6±0.2 | 4.9±0.8 | 66.2±0.7 | 1.1±0.1 | 96.1±0.2 | 2.1±0.2 | 56.3±1.1 | 1.6±0.4 | 51.0±0.3 | 3.1±0.7 |
| | VGSE | 49.1±0.7 | 39.6±1.8 | 61.6±3.1 | 48.1±0.5 | 60.2±1.6 | 42.1±1.4 | 83.2±2.3 | 55.9±1.4 | 42.7±0.3 | 40.2±0.5 | 11.2±1.8 | 17.5±2.3 |
| | ICIS | 62.1±1.7 | 46.9±1.3 | 73.5±0.7 | 57.2±1.0 | 64.5±3.2 | 37.0±3.1 | 92.5±1.5 | 52.7±3.0 | 55.8±1.4 | 27.2±4.3 | 49.2±2.0 | 35.1±2.9 |
| | +MSG | 63.1±1.4 | 50.2±1.2 | 72.5±0.8 | 59.4±0.9 | 62.6±2.4 | 42.8±2.8 | 91.1±1.5 | 58.2±2.0 | 57.2±1.2 | 27.8±3.9 | 42.6±1.9 | 33.6±2.5 |
| | CEMIL | 65.8±0.8 | 54.3±1.5 | 68.7±1.1 | 60.7±0.7 | 70.9±0.9 | 47.5±2.3 | 88.6±2.3 | 61.8±1.4 | 58.6±0.7 | 36.2±2.2 | 41.0±2.7 | 38.5±0.4 |
| | +MSG | 66.5±0.9 | 55.0±1.2 | 68.2±0.6 | 60.9±0.8 | 72.7±1.1 | 49.5±1.9 | 87.8±2.0 | 63.3±1.3 | 58.4±0.8 | 36.5±2.1 | 40.9±2.4 | 38.6±0.5 |
| | **Ours** | **71.8±0.6** | **59.6±1.2** | 65.8±0.9 | **62.5±0.6** | **77.2±1.3** | **57.3±2.1** | 86.2±1.8 | **68.8±1.2** | **63.3±0.9** | **43.0±1.5** | 39.5±1.3 | **41.2±0.7** |
| | ΔImp | 9.12% | 9.76% | - | 2.97% | 8.89% | 20.6% | - | 11.3% | 8.02% | 2.87% | - | 7.01% |

## 4. Experiments

This section first sets up the experiments and then evaluates the framework on three datasets, demonstrating superiority over state-of-the-art methods. In addition, ablation studies and further analyses are conducted, including robustness, generalization, and sensitivity. In the end, the continuous expansion capability of this framework is explored.

### 4.1. Experimental Setup

**Datasets.** The proposed framework is evaluated on three ZSL benchmarks, including fine-grained CUB (Wah et al., 2011) and coarse-grained domains represented by AWA2 (Xian et al., 2018) and SUN (Patterson et al., 2014). The division of classes into seen and unseen parts follows the partition protocol established by Xian et al. (2018). Specifically, CUB contains 200 bird species divided into 150 seen and 50 unseen classes, AWA2 is composed of 40 seen and 10 unseen animal classes, while SUN spans 717 scene classes, partitioned into 645 seen and 72 unseen classes. Crucially, adhering to the image-free ZSL setting, visual images of all classes are not accessible during the training phase.

**Evaluation Metrics.** The evaluation is performed in the ZSL and GZSL settings. For ZSL, the average per-class top-1 accuracy (**T1**) is provided, where the target labels are restricted to unseen classes. For GZSL, the top-1 accuracy is reported on samples of seen (**S**) and unseen (**U**) classes, where the inference search space includes both. To balance and measure how well the model overcomes the bias towards seen classes, the harmonic mean (**H**) is adopted as the main metric, calculated as $H = (2 \times S \times U)/(S + U)$. All results are averaged over six seeds to ensure statistical significance.

**Implementation Details.** The base classifier is initialized from the final layer of the ResNet101 model (He et al., 2016) trained on seen classes. The automated semantic pipeline mainly uses GPT-4o (Achiam et al., 2023) as the description generator and CLIP (Radford et al., 2021) as the embedding extractor, generating 30 multi-source descriptions for each class. The CSD module generates 10, 6, and 23 coarse-grained clusters for CUB, AWA2, and SUN, respectively, each with soft priors to correspond to different residual adapters. Structurally, the global encoders consist of a single linear layer coupled with a ReLU activation, while the decoders are instantiated as two-layer MLPs with a hidden dimension of 4096. The bottleneck dimension of adapters is set at 25% of the latent mapping dimension of the encoder, and the parameter $\alpha$ is set to 0.8. The framework is optimized using the Adam optimizer (Kinga et al., 2015) with a learning rate of $10^{-5}$, a predefined 1000 epochs, and an early stopping strategy. More details about time consumption and other settings are shown in Appendix E.1.

### 4.2. Comparison with State-of-the-Art Methods

**Baselines.** The comparative experiment incorporates a diverse spectrum of representative methods, including ConSE (Norouzi et al., 2014) and COSTA (Mensink et al., 2014), which rely on convex combinations of semantic embeddings and co-occurrence statistics, respectively; wDAE (Gidaris & Komodakis, 2019) and SubReg (Akyürek et al., 2021), which focus on refining classifier synthesis; as well as enhancement methods based on generative models or LLMs, VGSE (Xu et al., 2022), ICIS (Christensen et al., 2023), and CEMIL (Lyu et al., 2025). The specific method descriptions and applicability modifications are shown in Appendix E.2.

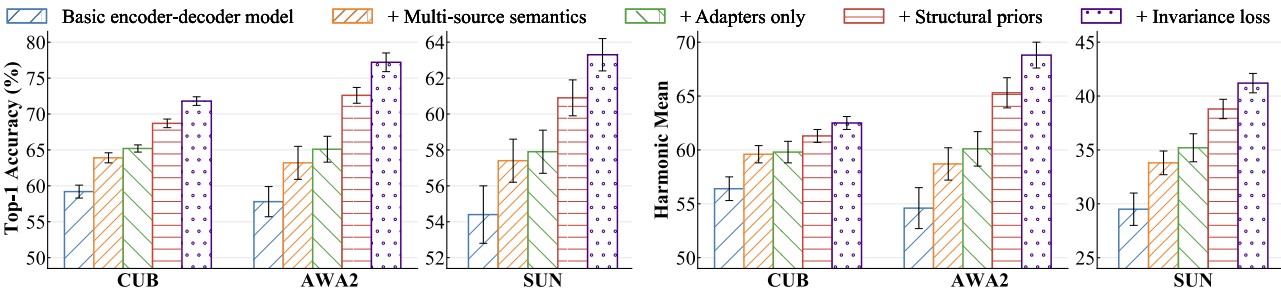

*Figure 3.* Ablation results on the core components of the proposed framework in terms of top-1 accuracy **T1** (%) and harmonic mean **H**.

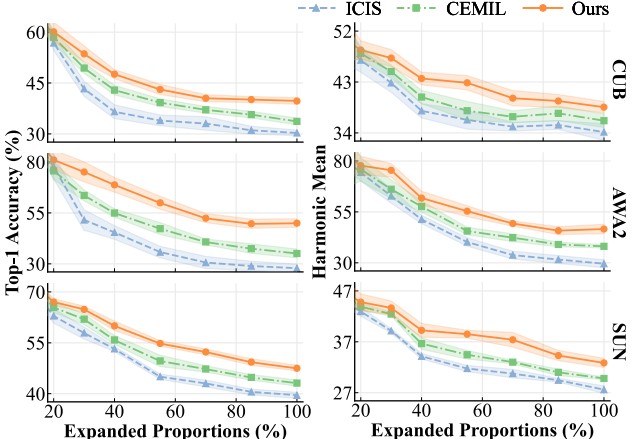

*Figure 4.* Comparison of our method with baselines across various expanded proportions of expanded classes at 20% seen classes in terms of top-1 accuracy **T1** (%) and harmonic mean **H**.

*Table 2.* **T1** of our method and baselines with Wiki2Vec attributes (Yamada et al., 2018) when shifting from ImageNet to the target.

| Target | wDAE | VGSE | ICIS | CEMIL | Ours |
|---------|------|------|------|-------|------|
| **CUB** | 3.7±0.7 | 7.3±0.0 | 4.2±0.5 | 15.5±1.1 | **20.2±0.7** |
| **AWA2** | 62.9±2.9 | 52.1±0.1 | 64.8±2.2 | 73.3±1.2 | **83.1±0.9** |
| **SUN** | 7.0±0.8 | 7.2±0.0 | 8.2±0.6 | 16.8±0.8 | **22.9±0.6** |
| **Average** | 24.5% | 22.2% | 25.7% | 35.2% | **42.1%** |

### 4.3. Ablation Study

To explore the incremental effectiveness and synergistic effects of the structural components, a comprehensive ablation study is organized, progressively transitioning from a basic dual encoder-decoder to the full framework. Specifically: 1) Utilize a basic dual encoder-decoder with a single semantic for each class. 2) Integrate the MSG module but apply mean pooling to the generated multi-source semantics. 3) Incorporate fine-grained adapters but without coarse-grained structural priors. 4) Introduce the CSD module to obtain coarse-grained structural constraints as routing priors. 5) Add the multi-source semantic influx into the ACS module with a causal invariance loss. The quantitative results of this progressive integration are detailed in Figure 3.

The integration of each component yields a consistent performance improvement, validating their necessity. Merely adding fine-grained adapters without structural guidance results in limited improvement, whereas a significant performance leap is observed upon the introduction of the CSD module. This indicates that this module provides effective routing priors capable of rationally assigning visually similar confusion classes to specific adapters, which is key to the framework's effectiveness and its ability to adapt to new classes, thereby mining subtle attributes within clusters and constructing accurate decision boundaries. In addition, compared to the single-semantic baseline, simply integrating multi-source semantics yields certain gains, but the synthesized classifiers are still contaminated by spurious correlations. By incorporating multi-source semantics and causal invariance loss, the framework successfully strips away non-causal noise, ensuring that the final classifiers possess class-discriminative characteristics. Further ablation studies of each module are shown in Appendix E.3.

**Comparable Scenarios.** All methods are evaluated under two different semantic scenarios. In the expert-based setting, all methods use expert-constructed attributes only. In the LLM-generated setting, our method uses the multi-source semantics produced by MSG, whereas the baselines use the mean-pooled multi-view descriptions from the study (Lyu et al., 2025). For a fairer comparison, we further evaluate two SOTA baselines using MSG as input. The top-1 accuracy and harmonic mean across two application scenarios and three datasets are shown in Table 1.

The experimental results show that the proposed method outperforms baselines in both scenarios. Specifically, by uncovering latent structural partitions, the weight synthesis process is decomposed into different adaptive directions for specific clusters, reducing the interference of general attributes on subtle ones. Meanwhile, adapters are employed to refine discriminative attributes among similar classes, thereby preserving the discriminative capacity lost that the shared mapping methods tend to lose. Notably, even with the same MSG-derived semantic, strong baselines still underperform our method, suggesting that the gains are not due to richer semantics alone. Through the introduction of these components, the adaptive mapping is established to balance generalization breadth and discrimination precision.

*Table 3.* Attribution analysis of the method's performance gains.

| Flag | CUB | | AWA2 | | SUN | |
|---|---|---|---|---|---|---|
| | **T1** | **H** | **T1** | **H** | **T1** | **H** |
| **A** | 59.2±0.9 | 56.4±1.1 | 57.8±2.2 | 54.6±2.0 | 54.4±1.6 | 29.5±1.5 |
| **B** | 68.2±0.7 | 61.0±0.7 | 70.5±2.1 | 63.8±1.9 | 59.5±1.2 | 37.9±1.1 |
| **C** | 63.1±0.7 | 59.4±0.8 | 62.6±2.3 | 58.2±1.5 | 57.2±1.4 | 33.6±1.2 |
| **D** | **71.8±0.6** | **62.5±0.6** | **77.2±1.3** | **68.8±1.2** | **63.3±0.9** | **41.2±0.7** |

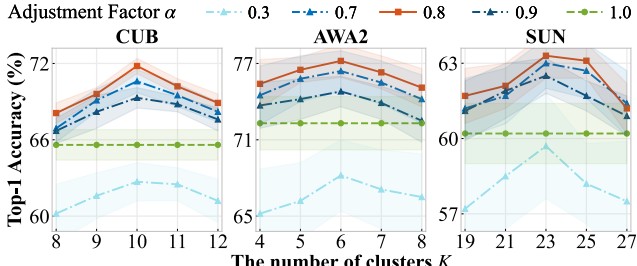

*Figure 5.* Sensitivity analysis on the effect of cluster number $K$ and adjustment factor $\alpha$ in terms of top-1 accuracy **T1** (%).

## 4.4. Further Empirical Analysis

**Gain Attribution Analysis.** To determine whether the source of attribution gains is a richer semantic foundation or granular-aware modeling, the paper conducts an orthogonal attribution study with a $2 \times 2$ design matrix, as reported in Table 3. Method A and B utilize single semantics combined with shared mapping and granularity modeling, respectively. Similarly, methods C and D employ multi-source semantics paired with shared mapping and granularity modeling. The semantic gain can be calculated as $((C-A)+(D-B))/2$, while the granularity-aware modeling gain can be defined as $((B-A)+(D-C))/2$. Through calculation, richer semantics alone bring an average gain of 3.8%, smaller than the 8.3% gain from granularity-aware modeling. This suggests that MSG mainly provides semantic coverage, while the main gains come from CSD and ACS, which turn semantics into a granularity-aware adaptive synthesis mechanism.

**Robustness Analysis in Data Efficiency.** To evaluate the robustness of our method under data-scarce scenarios, this experiment randomly samples subsets of the unseen classes, while keeping the proportion of seen classes at 20%. Performance trends with increasing proportions of unseen classes are shown in Figure 4. It can be observed that our method exhibits considerable robustness, which is reflected in the increasingly apparent performance disparity as the number of expanded classes increases. Notably, the SOTA baseline when extended to 50% of unseen classes, achieves approximate performance to our method when extended to 100% of classes. This empirical evidence further emphasizes the ability to mitigate interference in crowded semantic spaces and adaptively refine decision boundaries. More results for various seen class proportions are shown in Appendix E.5.

**Distribution Shift Generalization.** This experiment validates the generalizability of our method by shifting the data distribution. The model is trained using classes from ImageNet (Deng et al., 2009) as seen classes, and the learned semantic-to-weight mappings are transferred to three out-of-distribution datasets: AWA2, CUB, and SUN, where the classifier weights of unseen classes are then synthesized for evaluation. Table 2 shows that our method consistently outperforms baselines, suggesting its generalization against distribution shifts of various granularities and domains, which is attributed to the capture of core discriminative features.

## 4.5. Sensitivity Analysis

To investigate the impact of parameter selection on model performance, sensitivity analysis is conducted on two key parameters: the number of clusters $K$ and the adjustment factor $\alpha$ for the proportion of global and residual components. As reported in Figure 5, for the cluster number $K$, the peak performance indicates a balance between intra-cluster compactness and inter-cluster discrimination, while extremely small or large $K$ leads to either structural underfitting or group fragmentation. Since CSD provides soft priors and ACS does not rely on any single cluster in isolation, small changes in $K$ only lead to smooth routing adjustments rather than abrupt changes. Moreover, modest variations mainly affect the partition granularity rather than core neighborhood relations, allowing granularity-aware modeling to remain stable. Regarding the factor $\alpha$, the trend confirms that the adapters function as a complementary refinement, where excessive modulation risks overriding the original semantic manifold. Appendix E.6 shows more selections.

## 4.6. Continuous Class-Incremental Learning

To explore the capability of processing continuous class streams while mitigating forgetting, a generalized Class-Incremental Learning (CIL) (Zhou et al., 2024) scenario is constructed, where seen classes are considered as known sources of knowledge, while unseen classes are divided into five sequential and disjoint steps. After training on the seen classes, the model sequentially encounters batches of new unseen classes during the continuous expansion process, and the harmonic mean for each step is reported in Table 4.

The results show that our method consistently outperforms the baseline in all steps. Notably, as new steps arrive, the performance gap further widens, highlighting the superior stability of our method. This is because unseen classes are mapped into structurally compatible clusters rather than disrupting the previous manifold, which better preserves learned knowledge and minimizes the interference between old and new knowledge. Meanwhile, the causal mechanism effectively removes step-specific noise, ensuring that added classes are integrated based on invariant features. Further extended scenarios are discussed in Appendix E.9.

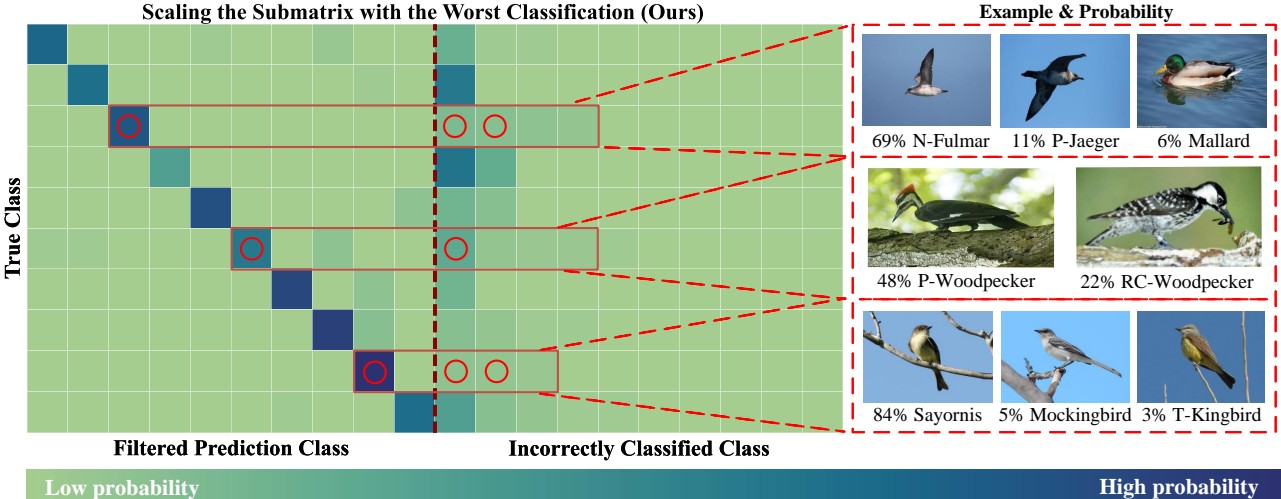

*Figure 6.* Visualization of the top-10 classes most likely to be misclassified under the ZSL setting. The left part illustrates the selected submatrices, while the right side displays specific class samples, which are similar to each other and easily mistaken for one another.

*Table 4.* Comparative harmonic mean **H** of our method with the SOTA baseline across five class-incremental learning tasks.

| Step | CUB | | AWA2 | | SUN | |
|---|---|---|---|---|---|---|
| | CEMIL | Ours | CEMIL | Ours | CEMIL | Ours |
| 1 | $69.6_{\pm 3.6}$ | $70.7_{\pm 2.9}$ | $80.9_{\pm 4.4}$ | $84.3_{\pm 4.1}$ | $49.2_{\pm 0.7}$ | $51.6_{\pm 1.0}$ |
| 2 | $65.4_{\pm 2.1}$ | $67.2_{\pm 3.1}$ | $74.2_{\pm 4.9}$ | $77.8_{\pm 3.2}$ | $45.3_{\pm 0.9}$ | $47.6_{\pm 0.7}$ |
| 3 | $62.9_{\pm 2.0}$ | $64.8_{\pm 2.3}$ | $68.2_{\pm 3.6}$ | $72.5_{\pm 2.0}$ | $41.6_{\pm 0.9}$ | $44.8_{\pm 0.6}$ |
| 4 | $60.8_{\pm 1.7}$ | $62.3_{\pm 1.5}$ | $63.4_{\pm 2.8}$ | $68.9_{\pm 1.8}$ | $39.6_{\pm 0.7}$ | $42.1_{\pm 0.7}$ |
| 5 | $59.1_{\pm 1.2}$ | $61.2_{\pm 1.4}$ | $59.7_{\pm 2.5}$ | $66.7_{\pm 2.1}$ | $38.2_{\pm 0.8}$ | $40.3_{\pm 0.5}$ |

### 4.7. Qualitative Results

To better understand how the proposed framework improves fine-grained discrimination, this qualitative analysis based on the confusion matrix is conducted in the ZSL setting, focusing on whether our method effectively reduces confusions between unseen classes difficult to distinguish rather than merely improving global metrics. Specifically, the top-10 classes that are most prone to misclassification are selected, which is described in detail in Appendix E.7. The relationship between confusion classes and their prediction proportions on the CUB dataset is shown in Figure 6.

In addition to the experiments mentioned above, Appendix E provides qualitative results, case studies, and further explorations of multiple modules, which validate the effectiveness of our framework from a more comprehensive evaluation.

## 5. Conclusion

This paper proposes a granularity-aware adaptive framework for zero-shot classifier expansion that overcomes the limitations of the shared semantic-to-weight mapping. By introducing multi-source semantic generation with controlled interventions and incorporating coarse-to-fine structural pri-

ors, the proposed method effectively decouples the learning objectives based on the granularity demands of distinct and similar classes, preventing gradient dominance between different classes from suppressing the subtle features needed for similar classes. Extensive experiments demonstrate consistent improvements and the trade-off between generalization and discrimination. Furthermore, this adaptive capability is crucial with continuous class expansion, allowing the system to remain stable even as semantic density increases.

## Acknowledgements

This research was supported in part by the National Nature Science Foundation of China (No. 62406302, 62137002, 62576327), in part by the Natural Science Foundation of Anhui province (No. 2408085QF195). This work was funded by the USTC Research Funds of the Double First-Class Initiative (Grant No. YD9110002085), Research Funds of Centre for Leading Medicine and Advanced Technologies of IHM (Grant No. 2025IHM01030).

## Impact Statement

This paper proposes a granularity-aware adaptive framework for image-free zero-shot classifier expansion, which synthesizes weights directly from semantic descriptions. By balancing general and subtle features, our method enhances discriminative stability in class expansion.

A primary positive impact of this paradigm is the inherent protection of data privacy. Since the model updates rely exclusively on semantic priors rather than visual samples, it eliminates the need to collect, store, or transmit sensitive user data during the expansion phase, making it highly suitable for privacy-sensitive applications.

Furthermore, our method reduces the reliance on large-scale annotated datasets and lowers the difficulty of deploying reliable recognition systems in data-scarce domains. However, since the reliance on semantic embeddings, the method may inherit biases present in the linguistic sources. Therefore, when applying this method, it is necessary to consider how to review and mitigate such biases in the semantic space.

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

This appendix provides supplementary materials for the submission, which is organized as follows:

- **A. Additional Related Works**    Discuss more related works and broader contexts including causal intervention in ZSL, zero-shot cross-modal alignment, and class-incremental learning.

- **B. Preliminaries**    Provide the preliminary knowledge of causal reasoning, including the fundamental concepts of structural causal models and causal interventions, as well as the adaptability to our proposed method.

- **C. Method Details and Extensions**    Present the framework details, including details of semantic generation, the composition of training loss, and the exploration of expanded scenarios.

- **D. Theoretical Analysis**    Offer a learnable derivation and comparative analysis of our method and shared mapping.

- **E. More Experimental Results and Analyses**    New additions include applicability analysis, confusion analysis, qualitative results, case studies of successes and failures, model parameter efficiency analysis, and further experiments on more datasets, comprehensive ablation studies, continuous class expansion, and more empirical analyses.

- **F. Limitations and Further Discussions**    Analyze the dependency on LLM generation quality, potential causes of failure, the comparison between intervention and counterfactual in this study, and the discussion of open-set learning.

- **G. LLM Prompt Design**    Provide LLM prompts for the multi-source semantic generation process.

## A. Additional Related Works

### A.1. Causal Intervention in Zero-Shot Learning

Causal intervention aims to reveal cause–effect relationships by actively intervening on variables and breaking spurious correlations induced by confounders (Pearl, 2009; Chen et al., 2026a). By shifting the modeling objective from statistical likelihood $P(Y|X)$ to causal probability $P(Y|\text{do}(X))$, intervention allows models to focus on invariant factors rather than environmental biases (Peters et al., 2017). Recently, causal intervention has been explored to address dataset bias and shortcut learning, including debiasing visual recognition (Zhang et al., 2023), domain generalization (Lv et al., 2022), and visual question answering (Liu et al., 2024), where it is applied to suppress context bias or disentangle causal visual factors from nuisance correlations (Tao et al., 2025). Appendix B shows preliminaries related to causality.

Causal intervention is also introduced to improve robustness by mitigating semantic and visual biases in ZSL task (Nilforoshan et al., 2023). Chen et al. (2024) explicitly models visual-semantic links and applies counterfactual intervention on attribute-related features to suppress spurious associations, enhancing the quality of semantic alignment. Ye et al. (2025a) demonstrates how diffusion augmentation with contrastive representation learning can reduce spurious visual-semantic correlations to improve unseen class recognition. However, they focus on either feature-level or correlation-regularization mechanisms and typically require access to visual samples or rely on fixed semantic representations, without directly incorporating interventions into classifier synthesis. Therefore, the intervention at the level of semantic composition and classifier expansion is introduced, enforcing invariance across multi-source semantics and latent structure.

### A.2. Zero-Shot Cross-Modal Alignment

Cross-modal alignment is the foundation of ZSL, aiming to establish a shared latent space between visual and semantic aspects (Cao et al., 2023; Zhang et al., 2025). Recent advancements have leveraged vision–language models to learning modality-invariant representations from web-scale image–text pairs, such as CLIP (Radford et al., 2021). Subsequent studies further improve alignment quality through prompt learning or adapter tuning to better adapt to downstream tasks (Zhou et al., 2024; Gao et al., 2024; Yuan et al., 2023). However, existing cross-modal alignment methods typically rely on global alignment objectives and treat semantic representations as homogeneous, making them sensitive to semantic bias and distribution shifts. Moreover, they focus on representation-level alignment rather than explicitly modeling how semantic information should be composed or intervened upon when transferring to unseen classes. These limitations are addressed through structure-aware and causally invariant semantic modeling to guide the alignment.

### A.3. Class-Incremental Learning

Class-Incremental Learning (CIL) aims to learn new classes continuously without forgetting previously acquired knowledge, reflecting a more realistic open-world scenario compared to standard ZSL (Zhou et al., 2024). Traditional methods rely

on replay buffers or knowledge distillation to mitigate catastrophic forgetting (Panos et al., 2023), while recent studies focus on prompt-based CIL by learning a pool of learnable prompts that query the frozen backbone to retrieve task-specific knowledge (Smith et al., 2023). They typically rely on visual data and focus on preserving previous knowledge, whereas classifier expansion aims to synthesize classifiers for unseen classes without any visual samples, which makes our framework less susceptible to catastrophic forgetting.

## B. Preliminaries

### B.1. Structural Causal Model

In standard supervised learning, deep learning models excel at learning statistical correlations from data, but these correlations are often spurious. Causal reasoning aims to go beyond correlation to discover the truly stable causal relationships between variables, using a Structural Causal Model (SCM) to formally describe the data generation process (Chen et al., 2026b; Ban et al., 2025), denoted as a tuple $\mathcal{B} = (\mathbf{U}, \mathbf{V}, \mathcal{F})$. Here, $\mathbf{U}$ denotes exogenous variables, $\mathbf{V}$ represents endogenous variables, and $\mathcal{F}$ is a set of structural equations. Each endogenous variable $V_i \in \mathbf{V}$ is determined by a structural equation:

$$V_i = f_i(\mathrm{Pa}(V_i), U_i), \tag{12}$$

where $\mathrm{Pa}(V_i)$ denotes the parents of $V_i$ in the causal graph, $U_i \in \mathbf{U}$ is an exogenous noise variable, and $f_i \in \mathcal{F}$. In this work, the SCM formulation is adopted as a conceptual tool to reason about how the multi-source class descriptions are generated and how semantic factors influence classifier generation, rather than to perform explicit causal discovery.

The causal relationships in the SCM are visualized as a Directed Acyclic Graph (DAG) $\mathcal{G}$. In ZSL, the conventional training objective models the observational probability $P(Y|X)$, which captures essentially mere statistical correlations. A common issue is the existence of a confounder $C$, such as the linguistic style $G$ or the observational environment $E$, which causally influences both $X$ and $Y$. This creates a back-door path $X \leftarrow C \rightarrow Y$, leading the model to learn spurious correlations rather than the true causal causality, e.g., associating the environment *snow* with the animal *wolf*.

### B.2. Causal intervention

To eliminate the effect of confounders and pursue the true causal effect of $X$ on $Y$, causal intervention is employed as the key tool for achieving this goal, modifying the data-generating process by actively setting a variable $X$ to a fixed value $x$ through the do($\cdot$) operator (Pearl & Mackenzie, 2018). Based on this, two different forms are distinguished. The first is observation $P(Y|X=x)$, which learns the correlation by describing what is the probability of $Y$ when we observe $X = x$; the second is intervention $P(Y|\mathrm{do}(X=x))$, which explores causality by describing what is the probability of $Y$ if we force $X = x$. Mathematically, performing do($X=x$) corresponds to a graph mutilation that deletes all incoming edges to variable $X$, effectively cutting off the back-door paths introduced by confounders. The definition can be represented as:[9]

$$P(Y|\mathrm{do}(X=x)) = \sum_u P(Y|X=x, U=u)\, P(U=u), \tag{13}$$

where $U$ summarizes the unobserved background factors. When there is a set of confounding variables $C$ that point to both $X$ and $Y$ ($X \leftarrow C \rightarrow Y$), the correlation between $X$ and $Y$ is spurious. The intervention do($X=x$) implies severing the $C \rightarrow X$ path. The back-door adjustment formula is used to calculate the intervention effect from observational data:

$$P(Y|\mathrm{do}(X=x)) = \sum_c P(Y|X=x, C=c)\, P(C=c). \tag{14}$$

The objective of the proposed framework shifts from optimizing the observational likelihood to optimizing the interventional probability $P(Y|\mathrm{do}(X))$. By simulating interventions (or enforcing causal invariance constraints), we force the model to rely on the invariant object semantics rather than unstable environmental biases, ensuring robust generalization to unseen categories. Notably, Equation (14) is designed to eliminate the interference of confounding factors, i.e., what kind of $Y$ is caused by $X$ itself. To achieve semantic expansion, this paper needs to control the attributes while intervening on confounding factors, which is consistent with the previous introduction. By intervening on the latent causal attribute $A$, the semantic generation process no longer depends on specific linguistic style $G$ and observational environment $E$, and multi-source semantics are generated through diversified sampling of style and environment.

---

[9]$P(Y|\mathrm{do}(X=x))$ is not equivalent to $P(Y|X=x)$. The latter merely filters samples but does not sever causal paths that produce $X$.

# C. Method Details and Extensions

This section provides details on the implementation of the framework, including the details for ontology and multi-source semantic generation (Appendix C.1, C.2, and Appendix G for prompts), the training loss of the global backbone and our framework (Appendix C.3 and C.4), and further exploration of the class expansion (Appendix C.5).

### C.1. Canonical Anchoring with Reliable Attribute Ontology

Current paradigms primarily utilize a single semantic to represent each class. However, such a single representation not only fails to cover the rich variations within that class but also easily confuses the intrinsic visual semantics with incidental linguistic styles and observational contexts, thereby introducing spurious correlations. To handle this reliance, the paper first introduces a rigorous mechanism to construct a set of reliable attribute ontologies for each class. By quantitatively evaluating the transferability and discriminability of semantic concepts, this module effectively filters out noise and non-visual correlations. These verified attributes are then aggregated via a meta-prompting strategy to generate a class-level canonical description, serving as a standardized baseline for the subsequent causal interventions. Consequently, this process migrates the hallucination bias inherent in unconstrained LLM generation, ensuring that the downstream causal intervention operates on a strictly visual-semantic manifold rather than spurious linguistic correlations[10].

**Universal Ontology Construction.** The semantic generation process begins with the construction of a universal attribute ontology, where the ontology is formally defined as a shared set of views $V = \{v_i\}_{i=1}^n$ designed to characterize all classes within the domain in this framework. Unlike unstructured text descriptions, these views act as a consistent coordinate system for visual discrimination. To acquire a universal candidate pool of these views, a domain-specific instruction is utilized to query LLMs, requiring the enumeration of unique visual perspectives rather than descriptions of specific instances. Similar to the study (Lyu et al., 2025), the generation of views consists of a coarse-to-fine-grained structure, using the prompt (1).

**Co-occurrence Estimation.** Even if the generation of non-visual views is avoided as much as possible during the process of universal ontology construction, the presence of LLM hallucinations makes more thorough ontology filtering necessary. To rigorously ground the generated ontology items in visual reality and compute the co-occurrence probability $\hat{p}(v_i|y)$ of view $v_i$ and class $y$ defined in Equation (1), a Monte Carlo sampling strategy based on the inherent randomness of LLMs is employed (Ledger & Mancinni, 2024), where random sampling noise during the decoding process may lead to false positives or false negatives in a single query. By aggregating responses over multiple independent queries, the sampling process effectively functions as a statistical stabilizer. True visual attributes exhibit high recurrence consistency across queries, whereas hallucinations or ambiguous associations tend to appear sporadically. Consequently, this transforms the task from a fragile binary check (i.e., whether an ontology $v_i$ is crucial for distinguishing class $y$) into a robust estimation of the posterior probability. This paper constructs the prompt (2) to generate attribute concepts multiple times and categorize them into their respective ontologies, where the categorization frequency is considered as a co-occurrence confidence $\hat{p}(v_i|y)$. Based on the sampling results, the confidence is estimated as $\hat{p}(v_i|y) = N_{\text{belong}}/N_{\text{total}}$, where $N_{\text{belong}}$ represents the number of attributes belonging to the ontology $v_i$ and $N_{\text{total}}$ denotes the total number of attributes across multiple queries.

**Reliable Attribute Ontology.** Not all ontologies are equally valuable for classification, where some may be too ubiquitous to provide distinction, while others may be too rare to support generalization. To filter out a set of reliable attribute ontologies, it is essential to balance transferability and discriminability (Shang et al., 2024; Ye et al., 2025b), where this trade-off is quantified using information metrics derived from the estimated co-occurrence probabilities $\hat{p}(v_i|y)$.

1. Transferability: Refers to the applicability scope of an ontology dimension. A dimension with high transferability is applicable to a wide range of seen and unseen classes, serving as a shared axis for knowledge transfer.

2. Discriminability: Refers to the partitioning capability of an ontology dimension. A dimension with high discriminability divides the domain into subsets, providing significant information gain for separating specific fine-grained groups.

Hallucinations in zero-shot generation typically manifest as non-visual abstractions or spurious correlations, which have a small estimated contribution $\hat{p}(v_i|y)$ in each class, initially filtered. The transferability $H_t(v_i)$ derived from Equation (3)

---

[10]The visual-semantic manifold refers to a shared latent space where semantic embeddings align with visual features. Restricting operations to it ensures that the synthesized weights reflect true visual differences between classes, rather than non-visual variations.

can filter random noise, which is because true attribute ontologies exhibit high consensus and applicability across multiple classes, while low-frequency noise is generated for one specific class due to random sampling. The latter can yield a sparse distribution $\hat{p}(y|v_i)$ and low entropy, leading to their rejection due to poor transferability scores. The discriminability $H_d(v_i)$ derived from Equation (3) can filter triviality. LLMs often generate trivial attributes by default that apply uniformly to all classes, but these provide zero information gain even if they are visually accurate. Since some hallucinations exhibit discriminative features due to randomness, the discriminability can be further optimized by introducing confidence scores:

$$H'_d(v_i) = \left( \frac{h_{\max} - h(v_i)}{h_{\max}} \right) \times \tanh \left( \sum\nolimits_{y \in Y} \hat{p}_{\text{raw}}(y, v_i) \right), \quad \text{where } \hat{p}_{\text{raw}}(y, v_i) = \hat{p}(v_i|y)\, p(y). \tag{15}$$

By optimizing the harmonic mean $H(v_i) = 2 \times H_t(v_i) \times H'_d(v_i)/(H_t(v_i) + H'_d(v_i))$ and also taking confidence scores into consideration, this framework effectively purifies the visual-semantic manifold, migrating hallucinatory noise and trivial redundancies. However, since the filtered ontologies are common to all classes and inconsistent with the specific visual features of each class, LLMs perform a self-consistency check to validate the existence of each ontology in reverse, ultimately obtaining a set of reliable attribute ontologies $O_y \subset V$ for each class. Example 1 illustrates the filtering process.

**Example 1.** *To demonstrate the pipeline, consider a simplified class set $Y = \{A, B, C, D\}$ and four candidate ontologies $V = \{v_1, v_2, v_3, v_4\}$, where the view $v_1$ is the non-visual noise, view $v_2$ is truly needed, view $v_3$ is a general perspective, and view $v_4$ focuses on highly distinctive features. The co-occurrence confidence $\hat{p}(v_i|y)$ generated by the LLM simulation and the conditional probability $\hat{p}(y|v_i)$ of each class on each ontology are shown in Table 5 and Table 6, respectively.*

*Table 5.* Co-occurrence confidence generated by LLM simulation.

| Class | $v_1$ | $v_2$ | $v_3$ | $v_4$ |
|-------|-------|-------|-------|-------|
| A | 0.05 | 0.70 | 0.25 | 0.00 |
| B | 0.00 | 0.10 | 0.20 | 0.70 |
| C | 0.00 | 0.70 | 0.30 | 0.00 |
| D | 0.05 | 0.30 | 0.60 | 0.05 |

*Table 6.* Conditional probability of each class on each ontology.

| View | A | B | C | D |
|------|------|------|------|------|
| $v_1$ | 0.50 | 0.00 | 0.00 | 0.50 |
| $v_2$ | 0.39 | 0.06 | 0.39 | 0.16 |
| $v_3$ | 0.19 | 0.15 | 0.22 | 0.44 |
| $v_4$ | 0.00 | 0.93 | 0.00 | 0.07 |

*The confidence scores in view $v_1$ are all low and can thus be initially filtered out. Then, the entropy of each ontology $v_i$ is calculated using Equation (2), resulting in $h(v_1) = 1.00$, $h(v_2) = 1.72$, $h(v_3) = 1.86$, and $h(v_4) = 0.35$. The transferability score corresponding to each ontology is required from Equation (3), resulting in $H_t(v_1) = 0.20$, $H_t(v_2) = 0.35$, $H_t(v_3) = 0.38$, and $H_t(v_4) = 0.07$. The discriminability score corresponding to each ontology is also required from Equation (3), where $h_{max} = \log 4 = 2$, resulting in $H_d(v_1) = 0.50$, $H_d(v_2) = 0.14$, $H_d(v_3) = 0.07$, and $H_d(v_4) = 0.82$. After introducing confidence scores, the discriminative score derived from Equation (15) is calculated, resulting in $H'_d(v_1) = 0.05$, $H'_d(v_2) = 0.13$, $H'_d(v_3) = 0.06$, and $H'_d(v_4) = 0.52$. The final score is calculated using the harmonic mean, resulting in $H(v_1) = 0.08$, $H(v_2) = 0.19$, $H(v_3) = 0.11$, and $H(v_4) = 0.13$. After sorting, the final order is $v_2 \prec v_4 \prec v_3 \prec v_1$. Notably, since the ontology $v_1$ in Table 5 has low confidence for all classes, its discriminative score calculated by Equation (15) is significantly reduced, which is not included in the final set of reliable attribute ontologies.*

**Canonical Description Generation.** Conditioned on the set of reliable attribute ontology $O_y$, a canonical description $t_y$ is synthesized to serve as the stable semantic root of the class $y$. To ensure that this description remains an objective reflection of visual facts rather than a source of hallucinated noise, LLMs are instructed to act as a neutral observer, introducing a meta-prompting strategy to concatenate attributes $A_y$ generated from $O_y$ into coherent semantics without adding external information or applying specific linguistic styles. The prompt (3) provides the generation of the canonical description.

### C.2. Interventional Semantic Expansion

The canonical description $t_y$ obtained in the previous stage serves as the invariant semantic root. However, visual expressions of class $y$ in real-world scenarios are confounded by generation styles $G$ (e.g., observational angles) and diverse environments $E$ (e.g., background), which causes the previous single-semantic representation of a class to be easily disturbed and to struggle to capture sufficient visual diversity. Therefore, to bridge the modality gap between the static text $t_y$ and dynamic visual instances, the framework employs an interventional expansion strategy for the multi-source semantic generation.

Formally, the generation of a descriptive sample $t$ is modeled as a structural causal model, where $t = \mathcal{F}(A, E, G)$. To eliminate spurious correlations (e.g., associating *Polar Bear* solely with *Ice*), intervention operations $do(G = g_a)$ and

$do(E = e_b)$ are performed on the generation process, where the style set $G = \{g_a\}_{a=1}^{m_1}$ contains stylistic modifiers and the environment set $E = \{e_b\}_{b=1}^{m_2}$ contains contextual modifiers. These two sets can be obtained from the prompt (4).

For each class $y$, the pair $(g_a, e_b)$ is randomly sampled from $G \times E$ and used to instruct LLMs in rewriting the canonical description $t_y$, where $t_y$ can be considered a product of the default descriptive style $g_0$ and environment $e_0$. Whenever an intervention is performed, the changes in these two sets occur in the way: $g_0 \to g_a$, $e_0 \to e_b$. Notably, LLMs are constrained to maintain the visual invariance of the canonical attributes $A_y$ generated from $O_y$ while adapting the non-visual context, generating expanded multi-source semantic descriptions by causal interventions as shown in the prompt (5).

By systematically sampling pairs $(g_a, e_b)$ and applying this pipeline, each single canonical anchor $t_y$ is expanded into a dense semantic set $T_y$, in which the signal of visual attributes $\{A_y\}_{y \in Y}$ remains constant, while the non-visual noise varies randomly. This structure minimizes the mutual information between the class label and spurious domain correlations, providing an ideal data distribution for the subsequent robust semantic-to-weight mapping.

### C.3. Global Semantic-to-Weight Alignment Backbone

Since training an autoencoder separately for each cluster makes causal spaces across clusters incomparable and inevitably leads to inaccurate partitions when the differences between classes are too large, the paper trains a global semantic-to-weight mapping to capture universal visual-semantic correlations shared across all classes. Specifically, the backbone consists of two parallel branches, that is, a semantic autoencoder $(\mathcal{E}_\mathcal{T}, \mathcal{D}_\mathcal{T})$ for the text modality and a weight autoencoder $(\mathcal{E}_\mathcal{W}, \mathcal{D}_\mathcal{W})$ for the visual classifier modality. The semantic branch takes the fused semantic representation $\hat{\mathbf{F}}_y \in \mathbb{R}^{d_a}$ derived in Equation (6) rather than the multi-source semantic representation $\mathbf{F}_y \in \mathbb{R}^{m \times d_e}$ as input, since here it is only used to train a global backbone. The weight branch takes the visual classifier weight $w_s \in \mathbb{R}^{d_v}$ from a pre-trained classification model as input. Both encoders map inputs to a shared latent space $\mathcal{Z}$ and achieve classification using reconstruction and cross-modal losses.

The global backbone is trained on the set of seen classes $S$ to minimize the discrepancy between modalities while preserving information. The objective function $\mathcal{L}_{\text{global}}$ comprises two reconstruction losses and two cross-modal losses, a total of four items, including semantic reconstruction, weight reconstruction, semantic-to-weight alignment, and weight-to-semantic reverse verification. The reconstruction loss $\mathcal{L}_{\text{rec}}$ ensures that latent features retain sufficient information to reconstruct original inputs for both modalities, which consists of semantic reconstruction $\mathcal{L}_{\mathcal{T} \to \mathcal{T}}$ and weight reconstruction $\mathcal{L}_{\mathcal{W} \to \mathcal{W}}$:

$$\mathcal{L}_{\text{rec}} = \mathcal{L}_{\mathcal{T} \to \mathcal{T}} + \mathcal{L}_{\mathcal{W} \to \mathcal{W}} = \sum_{y \in Y} d(\mathcal{D}_\mathcal{T}(\mathcal{E}_\mathcal{T}(\hat{\mathbf{F}}_y)), \hat{\mathbf{F}}_y) + \sum_{s \in S} d(\mathcal{D}_\mathcal{W}(\mathcal{E}_\mathcal{W}(w_s)), w_s), \tag{16}$$

where $d(\cdot, \cdot)$ denotes the cosine distance and semantic reconstruction loss can include unseen classes or only seen classes. The cross-modal alignment loss $\mathcal{L}_{\text{align}}$ enforces encoding from one modality and decoding into another to produce valid representations, consisting of semantic-to-weight alignment $\mathcal{L}_{\mathcal{T} \to \mathcal{W}}$ and weight-to-semantic verification $\mathcal{L}_{\mathcal{W} \to \mathcal{T}}$:

$$\mathcal{L}_{\text{align}} = \mathcal{L}_{\mathcal{T} \to \mathcal{W}} + \mathcal{L}_{\mathcal{W} \to \mathcal{T}} = \sum_{s \in S} d(\mathcal{D}_\mathcal{W}(\mathcal{E}_\mathcal{T}(\hat{\mathbf{F}}_s)), w_s) + \sum_{s \in S} d(\mathcal{D}_\mathcal{T}(\mathcal{E}_\mathcal{W}(w_s)), \hat{\mathbf{F}}_s), \tag{17}$$

where $\hat{\mathbf{F}}_s$ denotes the semantic representation corresponding to class $s$. The total loss for the global backbone is a sum:

$$\mathcal{L}_{\text{global}} = \mathcal{L}_{\text{rec}} + \mathcal{L}_{\text{align}} = \mathcal{L}_{\mathcal{T} \to \mathcal{T}} + \mathcal{L}_{\mathcal{W} \to \mathcal{W}} + \mathcal{L}_{\mathcal{T} \to \mathcal{W}} + \mathcal{L}_{\mathcal{W} \to \mathcal{T}}. \tag{18}$$

During inference for an unseen class $u \in U$, since $w_u$ is unavailable, the semantic-to-weight path is used to synthesize the classifier weight $w_u = \mathcal{D}_\mathcal{W}(\mathcal{E}_\mathcal{T}(\hat{\mathbf{F}}_u))$, where $\hat{\mathbf{F}}_u$ represents the class-level semantic representation corresponding to the unseen class $u$. Through the process described above, a learned semantic encoder $\mathcal{E}_\mathcal{T}$ is obtained, which serves as the global backbone in the ACS module. This paper adopts a strategy of freezing the global semantic encoder while keeping other encoders and decoders learnable, preventing the arbitrary distortion of the global semantic topology and the absorption of adapter-specific behaviors (Gao et al., 2024). Therefore, fine-grained adaptations also possess the ability to generalize to unseen classes while maintaining a practical compensation to coarse-grained global features.

### C.4. Granular Total Training Objective with Causal Invariance

The proposed framework is trained in an end-to-end fashion with the fixed semantic encoder using a composite objective function, utilizing the paired semantic descriptions and visual classifier weights $\{(t_s, w_s) | s \in S\}$ of seen classes and descriptors only $\{t_u | u \in U\}$ in unseen classes. The granular objective $\mathcal{L}_{\text{total}}$ is designed to enforce three critical properties: information preservation, modal alignment, and causal robustness, which formally correspond to the autoencoder reconstruction term $\mathcal{L}'_{\text{rec}}$, the cross-modal alignment term $\mathcal{L}'_{\text{align}}$, and the causal invariance term $\mathcal{L}'_{\text{inv}}$, respectively.

As detailed in the global backbone (Appendix C.3), the paper minimizes the reconstruction loss for both the multi-source semantic $\mathbf{F}_y \in \mathbb{R}^{m \times d_e}$ and the visual weight $w_s \in \mathbb{R}^{d_v}$ to prevent information collapse in the latent space, denoted as:

$$\mathcal{L}'_{\text{rec}} = \mathcal{L}'_{\mathcal{T} \to \mathcal{T}} + \mathcal{L}_{\mathcal{W} \to \mathcal{W}} = \sum_{y \in Y} d\left(\frac{1}{m} \sum_{j=1}^{m} \mathcal{D}_{\mathcal{T}}(\hat{\mathcal{E}}_{\mathcal{T}}(\mathbf{F}_{y,j})), \frac{1}{m} \sum_{j=1}^{m} \mathbf{F}_{y,j}\right) + \sum_{s \in S} d(\mathcal{D}_{\mathcal{W}}(\mathcal{E}_{\mathcal{W}}(w_s)), w_s), \quad (19)$$

where $\hat{\mathcal{E}}_{\mathcal{T}}(\mathbf{F}_{y,j}) = [\mathcal{P}_c(\mathcal{E}'_{\mathcal{T}}(\mathbf{F}_{y,j}))||\mathcal{P}_n(\mathcal{E}'_{\mathcal{T}}(\mathbf{F}_{y,j}))]$, derived from Equation (11), means that the learnable components in the ACS module, including the global encoder, adapters, and mappings explicit explicitly separating causal and non-causal elements, are simplified using $\hat{\mathcal{E}}_{\mathcal{T}}(\cdot)$. Furthermore, to meet the requirements of subsequent cross-modal alignment, this paper needs to either sum or concatenate the reconstructed multi-source semantic representations into a single long vector, so that it corresponds to the per-class weight. Although it appears as a single semantic representation, it is in fact formed by fusing multiple semantics from different linguistic styles and contextual variations.

To enable zero-shot transfer, the semantic and visual modalities must be aligned. The cross-modal alignment term $\mathcal{L}'_{\text{align}}$ includes the semantic-to-weight alignment $\mathcal{L}'_{\mathcal{T} \to \mathcal{W}}$ and the weight-to-semantic verification $\mathcal{L}'_{\mathcal{W} \to \mathcal{T}}$, represented as:

$$\mathcal{L}'_{\text{align}} = \mathcal{L}'_{\mathcal{T} \to \mathcal{W}} + \mathcal{L}'_{\mathcal{W} \to \mathcal{T}} = \sum_{s \in S} d\left(\frac{1}{m} \sum_{j=1}^{m} \mathcal{D}_{\mathcal{W}}(\hat{\mathcal{E}}_{\mathcal{T}}(\mathbf{F}_{s,j})), w_s\right) + \sum_{s \in S} d\left(\mathcal{D}_{\mathcal{T}}(\mathcal{E}_{\mathcal{W}}(w_s)), \frac{1}{m} \sum_{j=1}^{m} \mathbf{F}_{s,j}\right), \quad (20)$$

To mitigate spurious correlations, e.g., background bias or style bias, the paper implements the causal invariance constraint detailed in Equation (10), constructing an intervened latent pair by swapping non-causal components for a given class $y$. Since the semantic encoder is frozen but the other encoders and decoders are learnable, causal invariance can be enforced through latent decomposition and classifier synthesis without allowing the backbone to absorb adapter-specific behaviors, i.e., the adapter-modulated latent and decoder levels. A more general form of the causal invariance loss $\mathcal{L}'_{\text{inv}}$ is presented here, the purpose of which is to make the weight decoder yield consistent classifier weights despite this swap, represented as:

$$\mathcal{L}'_{\text{inv}} = \sum_{s \in S} \mathbb{E}_{j \neq j'} \left(d(\mathcal{D}_{\mathcal{W}}(\hat{\mathcal{E}}_{\mathcal{T}}(\mathbf{F}_{s,j})), \mathcal{D}_{\mathcal{W}}(\hat{\mathcal{E}}_{\mathcal{T}}(\mathbf{F}_{s,j}, \mathbf{F}_{s,j'})))\right), \quad (21)$$

where $\hat{\mathcal{E}}_{\mathcal{T}}(\mathbf{F}_{s,j}, \mathbf{F}_{s,j'}) = [\mathcal{P}_c(\mathcal{E}'_{\mathcal{T}}(\mathbf{F}_{y,j}))||\mathcal{P}_n(\mathcal{E}'_{\mathcal{T}}(\mathbf{F}_{y,j'}))]$ indicates that for the same class $s$, the causal component from the semantic $\mathbf{F}_{s,j}$ and the non-causal component from another semantic $\mathbf{F}_{s,j'}$ are concatenated. Since this loss only imposes constraints between multi-source semantic mappings and synthesis weights, and lacks any constraint with respect to the true weights $w_s$, jointly optimizing this loss $\mathcal{L}'_{\text{inv}}$ (or the loss $\mathcal{L}_{\text{inv}}$ in the main text, expressing the same meaning but differing in form) with the semantic-to-weight alignment loss $\mathcal{L}'_{\mathcal{T} \to \mathcal{W}}$ yields a robust mapping enriched with causal invariance.

The total training objective of the proposed framework can be represented as:

$$\mathcal{L}_{\text{total}} = \mathcal{L}'_{\text{rec}} + \mathcal{L}'_{\text{align}} + \mathcal{L}'_{\text{inv}} = \mathcal{L}'_{\mathcal{T} \to \mathcal{T}} + \mathcal{L}_{\mathcal{W} \to \mathcal{W}} + \mathcal{L}'_{\mathcal{T} \to \mathcal{W}} + \mathcal{L}'_{\mathcal{W} \to \mathcal{T}} + \mathcal{L}'_{\text{inv}}. \quad (22)$$

This joint optimization strategy, combined with the framework of the global backbone and learnable adapters, ensures that synthesized weights for unseen classes are not only visually discriminative but also causally robust against style variations.

### C.5. Continuous Incremental Expansion and Structural Evolution

To ensure the practicality and robustness of the proposed framework, this paper extends the weight synthesis part for unseen classes and briefly constructs three scenarios to progressively handle the continuous emergence of new classes in real-world environments. Before describing the three scenarios, the synthesis of the weights for new classes is briefly reviewed. When a new class $u \in U$ appears, the weight synthesis process begins with multi-source semantic generation, followed by obtaining the corresponding cluster using the CSD module and further deriving the routing prior $\mathbf{r}_u$. Subsequently, under the already trained network architecture, by inputting multiple semantic prototypes of this class, the learned features are forced to eliminate non-causal style differences under fine-grained partitioning, ultimately yielding robust classifier weights.

**Scenario 1: Transductive Setting.** In the transductive ZSL setting, the semantic descriptions of unseen classes $U$ are available during the training phase, even though their visual samples are not. This setting is also adopted in the method described in the main text, primarily reflected in the CSD module and semantic reconstruction loss $\mathcal{L}'_{\mathcal{T} \to \mathcal{T}}$. In this case, the paper performs class-level structure discovery (Section 3.3) on the union of all semantics $S \cup U$, which ensures that the learned prototypes $\{\mu_k\}$ and adapters $\{\mathcal{R}_k\}$ cover the entire semantic manifold from the origin. Therefore, the inference process is very straightforward, which is denoted in Equation (11), because each class naturally belongs to a learned cluster.

**Scenario 2: Inductive Setting with a Few New Classes.** In the strict inductive ZSL setting, unseen semantics are strictly unavailable during training, but their representations $\{\mathbf{F}_u\}_{u \in U}$ can be generated easily and intuitively during inference. In contrast, it is important to determine which cluster the unseen class $u$ belongs to and obtain the corresponding routing prior $\mathbf{r}_u$ to guide the application of adapters. To this end, this paper first calculates the maximum cosine similarity of a new class $u$ to the existing semantic cluster center $\{\mu_k\}_{k=1}^K$, which can be expressed as:

$$s_{\max}(u) = \max_{k=1,2,\ldots,K} \left(\mu_k^T \mathbf{F}_u'\right),$$
(23)

where $\mathbf{F}_u'$ is the representation obtained by semantic fusion, derived from Equation (6). Subsequently, a proximal new class refers to a class $u$ whose cosine similarity to the nearest learned cluster center exceeds a high confidence threshold $\tau$, i.e., $s_{\max} \geq \tau$. ZSL-based classifier expansion is established under this condition, which indicates that while the class identity is novel (e.g., *Leopard*), it shares significant semantic attributes with a known cluster (e.g., *Felines* includes *Tiger* and *Lion*). In this agreement, the soft routing mechanism derived from Equation (8) functions as a sharp selector and activates the corresponding adapters, generating refined weight $w_u$ that capture fine-grained visual characteristics specific to that cluster.

To ensure the comprehensiveness of the framework, consider the weight synthesis for a small number of semantically distant new classes, i.e., out-of-distribution (OOD) samples whose input classes are semantically distinct from all training clusters. In such cases, where the cosine similarity between the new class $u$ and the nearest learned cluster center exceeds the high confidence threshold $\tau$ ($s_{\max} < \tau$), enforcing adapter refinement based on inappropriate structural priors may degrade performance. This is because, unlike scenario 2 which relies on Equation (8) to calculate routing priors via *Softmax*, there is no match with any cluster in this scenario and calculating via *Softmax* forces the assignment of adapters and introduces noise. To handle this, the paper calculates the reliability score $\lambda(u)$ based on the distance to the nearest cluster, expressed as:

$$\lambda(u) = \text{Sigmoid}\left(\frac{s_{\max}(u) - \tau}{\gamma}\right),$$
(24)

where $\gamma$ is a temperature parameter controlling the transition sharpness and $\lambda(u) \to 0$ suppresses erroneous residuals from irrelevant adapters, ensuring baseline performance. The final weight synthesis, derived from Equation (9), is modified to:

$$\mathbf{z}_{y,j} = \sum_{k=1}^K r_{y,k} \left((1 - \lambda(u))\alpha \mathbf{h}_{y,j} + \lambda(u)(1 - \alpha)\mathcal{R}_k\left(\mathbf{h}_{y,j}\right)\right).$$
(25)

The model suppresses the potentially erroneous adapter residuals and safely falls back to the robust global backbone.

In this setting, when new classes appear only in the form of semantic descriptors, the evolution of the framework is restricted to structural reorganization, updating the routing prior $\mathbf{r}_u$ or the proportion of global encoder to adapters. In this case, performance improvements mainly arise from modeling the assignment relationship between semantics of unseen classes and the original clusters. However, in real-world scenarios, the static structural prior inevitably faces a dilemma between plasticity and stability, which means that a fixed structure preserves previous knowledge well but fails to dynamically represent more in-distribution or out-of-distribution semantics. Therefore, to facilitate a more in-depth discussion, this paper introduces the final scenario to extend the semantic structure and adapters while avoiding catastrophic forgetting.

**Scenario 3: Continuous Expansion.** The continuous expansion can be discussed from two perspectives: the splitting of existing clusters into new ones (internal fission) and the formation of significantly different clusters (outward branching).

This paper starts with the exploration of the first perspective, considering that the classes to be expanded can reliably depend on the original cluster partitioning to obtain their own prior information. However, as the number of classes increases, the original cluster partitioning becomes increasingly ambiguous, inevitably reducing the effectiveness of the structural prior. Therefore, the framework needs to detect structural changes and handle them accordingly. Since structural heterogeneity is difficult to evaluate using simple metrics, such as intra-cluster variance, this paper applies a hypothetical splitting strategy and evaluates the structural gain score after splitting to determine whether a real split is required. Specifically, the hypothetical split is periodically performed on each cluster $k$ with the semantic set $Y_k$, where the semantic set is filtered by the condition that the routing vector $\mathbf{r}_y$ for each class $y$ has its maximum value on $k$-component, including the semantics of both the original and new classes[11]. This splitting process is achieved by applying the vMF mixture model to divide $Y_k$ into two

---

[11]When discussing the classifier expansion (Class Incremental Learning, CIL), there is usually an implicit assumption that the new classes are not only for testing, but that a small number of samples of these new classes will also be collected over time.

subclusters with centers $\mu_{k_1}$ and $\mu_{k_2}$ based on the Maximum A Posteriori (MAP) probability[12]. The paper then evaluates the validity of this split using the structural gain score $\mathcal{G}(k)$, derived from the silhouette coefficient adapted for cosine geometry. For each sample $c \in Y_k$, the cosine distance to centers $\mu_{k_1}$ and $\mu_{k_2}$ is calculated, denoted as cohesion $a(c)$ and separation $b(c)$, respectively. The global gain $\mathcal{G}(k)$ of cluster $k$ is the average of silhouette scores $s(c)$ over all samples, expressed as:

$$\mathcal{G}(k) = \frac{1}{|Y_k|} \sum_{c \in Y_k} s(c), \quad \text{where } s(c) = \frac{b(c) - a(c)}{\max(a(c), b(c))}. \tag{26}$$

If $\mathcal{G}(k)$ exceeds a threshold $\mathcal{G}_0$, it suggests that the data is better modeled as two separate clusters than as one, triggering a real division. On the contrary, a low score indicates that the cluster is unimodal, and forcing a split would be harmful. After determining that structural evolution is necessary, the acquisition of routing priors also changes accordingly, depending on a new set of parameters $\{(\mu_{k_j}, \kappa_{k_j}, \pi_{k_j})\}_{j=1}^{2}$. Taking the transformation of the original cluster $k$ into clusters $k_1$ and $k_2$ as an example, the posterior probability that class $c$ belongs to the new cluster $k_1$ can be expressed as:

$$r'_{c,k_1} = \frac{\pi_{k_1} C_{d_a}(\kappa_{k_1}) \exp(\kappa_{k_1} \mu_{k_1}^T \mathbf{F}'_c)}{\sum_{j=1}^{k-1} \pi_j C_{d_a}(\kappa_j) \exp(\kappa_j \mu_j^T \mathbf{F}'_c) + \sum_{j=1}^{2} \pi_{k_j} C_{d_a}(\kappa_{k_j}) \exp(\kappa_{k_j} \mu_{k_j}^T \mathbf{F}'_c) + \sum_{j=k+1}^{K} \pi_j C_{d_a}(\kappa_j) \exp(\kappa_j \mu_j^T \mathbf{F}'_c)}. \tag{27}$$

By updating the routing vectors $\mathbf{r}'$ for all classes using the above equation, the newly added adapters $\mathcal{R}_{k_1}(\cdot)$ and $\mathcal{R}_{k_2}(\cdot)$ in the ACS module are initialized with the parameters of their parent adapter $\mathcal{R}_k(\cdot)$ and participate in subsequent training.

The second perspective is based on a scenario where many classes with semantic representations significantly different from clusters formed by the original seen classes. As the framework operates continuously, a large number of distant new classes have accumulated, indicating that the semantic distribution has grown new branches and is not covered by the original $K$ clusters. However, relying solely on the global backbone in Equation (25) is suboptimal for long-term performance. Therefore, it is necessary for the framework to undergo structural evolution to meet the demands of continuous expansion[11], which can be achieved by applying a monitoring and evolution strategy. Specifically, this strategy continuously monitors the density of incoming inference requests and starts with calculating the out-of-distribution ratio $\rho$, denoted as:

$$\rho = \frac{\sum_{u \in U_s} \mathbb{I}[s_{\max}(u) < \tau]}{|U_s|}, \tag{28}$$

where $U_s$ is the buffer of recent new classes and $\mathbb{I}(\cdot)$ represents the indicator function. When $\rho$ exceeds a critical threshold $\rho_0$, it signals that the current structural prior is obsolete and the framework need to be switched from inference to evolution. Subsequently, to accommodate the new semantic branches, the estimation of vMF mixture model on the updated semantic pool $S \cup U_s$ is performed, resulting in an expanded set of cluster centers $\{\mu'_k\}_{k=1}^{K'}$, where $K' > K$. By comparing the evolution of the clusters before and after, the new adapters inherit parameters directly from the old adapters for stable clusters, while for newly formed clusters, the framework defines their neighboring nodes and initializes new adapters through warm-starting[13]. This facilitates rapid adaptation to the new semantic branch while preventing catastrophic forgetting.

For continual learning, the structural-evolution hyperparameters $\{\tau, \mathcal{G}_0, \rho_0\}$ can also be defined by statistical significance rather than dataset-specific heuristics, improving reproducibility and reducing tuning burden. $\tau$ measures whether a new class deviates significantly from the current structure. After training, we compute the cosine similarity between each seen class and the center of its assigned cluster, and define $\tau$ using a lower-tail statistic of this distribution like the 10th percentile. $\mathcal{G}_0$ determines whether a cluster should be split internally. We use the silhouette scores of seen classes under the initial clustering as a reference and define $\mathcal{G}_0$ using an upper-tail statistic of this distribution like the 75th percentile. $\rho_0$ controls the difficulty of triggering structural evolution. To balance the computational cost of more frequent updates against performance, we set $\rho_0 = 0.1$ on all datasets, meaning evolution is triggered when 10% of a new batch is not well covered.

**Parameter-Efficient Adapter Variant**    In continuous class expansion, the number of adapters increases with the number of clusters, leading to a growing number of trainable parameters and low training efficiency. While standard MLPs or the bottleneck layer can achieve the purpose of refining decision boundaries, they still introduce a large number of parameters, that is, $K \times d_z^2$ for MLPs and $K \times 2 \times d_z \times d_b$ for the bottleneck layer, where $d_z$ is the dimension after global encoding

---

[12]This paper sets the number of divisions to 2, and subsequent periodic observations can evolve into a recursive binary fission process.

[13]The parameters of new adapters are obtained by adding Gaussian noise to the parameters of adapters corresponding to the nearest cluster. This process is expressed as $\theta' \leftarrow \theta + \mathcal{N}(0, \sigma^2 I)$, where $\theta'$ and $\theta$ denote the parameters of new and original adapters, respectively.

$\mathcal{E}_{\mathcal{T}}(\cdot)$ and $d_b$ is the bottleneck layer dimension. To support potential structural evolution and ensure the sustainability of the proposed framework, a variant is introduced where each adapter $\mathcal{R}_k(\cdot)$ is instantiated as a low-rank adapter (Hu et al., 2022), which not only improves parameter efficiency but also imposes a geometric regularizer on the adaptation process.

Let $\mathbf{h} = \mathcal{E}_{\mathcal{T}}(\bar{\mathbf{f}})$ denote the feature projected by the global backbone, where $\bar{\mathbf{f}}$ is the multi-source semantic representation. Instead of learning a full-rank mapping, the $k$-th expert $\mathcal{R}_k(\cdot)$ is constrained to update $\mathbf{h}$ within a low-dimensional subspace. This is parameterized by two low-rank matrices $A_k \in \mathbb{R}^{d_z \times d_r}$ and $B_k \in \mathbb{R}^{d_z \times d_r}$, where the rank $d_r \ll d_z$. The residual mapping is defined as a linear low-rank transformation, which can be expressed as:

$$\mathcal{R}_k(\mathbf{h}) = B_k A_k^T \mathbf{h} \tag{29}$$

Substituting this into the fine-grained adaptive combination derived from Equation (9), the representation $\mathbf{z}_{y,j}$ becomes:

$$\mathbf{z}_{y,j} = \sum_{k=1}^{K} r_{y,k} \left( \alpha \mathbf{h}_{y,j} + (1-\alpha) B_k A_k^T \mathbf{h}_{y,j} \right) = \left( \alpha I_{d_z} + (1-\alpha) \sum_{k=1}^{K} r_{y,k} B_k A_k^T \right) \mathbf{h}_{y,j}, \tag{30}$$

This formulation offers the advantage of subspace constraints over full-rank adapters beyond mere compression. The update $\Delta\mathbf{h} = \mathcal{R}_k(\mathbf{h}) = B_k A_k^T \mathbf{h}$ is strictly confined to the column space of $B_k$, which ensures that the global semantic topology preserved by the identity term $I_{d_z}$ is not arbitrarily distorted. The cluster-specific knowledge is encoded only in these few principal directions, while the shared global structure remains intact in the orthogonal complement. Furthermore, for unseen classes whose routing priors $\mathbf{r}$ may be uncertain, the low-rank constraint can prevent synthesized weights from drifting into invalid regions of the manifold, acting as a strong regularizer against overfitting to specific characteristics of seen classes.

The number of trainable parameters for low-rank adapters is $K \times 2 \times d_z \times d_r$, which is the minimum required since $d_r \ll d_b < d_z$. According to the experimental details provided in Appendix E.1, consider a high-dimensional setting with the globally encoded dimension $d_z = 2048$, the bottleneck dimension $d_b = 512$, and the rank $d_r = 8$. Through calculation, the number of parameters for an adapter is a, b, and c under the MLP, bottleneck layer, and low-rank settings, respectively. By calculation, the number of parameters of a single adapter under the standard MLPs, bottleneck layer, and low-rank settings are approximately 4.2M, 2.1M, and 33K, respectively. As the system evolves to incorporate $K$ distinct experts, the storage overhead for the first two settings will be significantly greater, whereas the third setting requires less. This extreme parameter efficiency is what makes the constructed continuous expansion scenario computationally feasible.

## D. Theoretical Analysis

### D.1. Differentiability Analysis and End-to-End Gradient Propagation

The proposed framework integrates the MSG, CSD, and ACS modules into a unified pipeline, where the MSG module is employed for semantic generation and remains frozen during the process, while the latter two modules participate in end-to-end gradient propagation. Traditional clustering algorithms typically employ a hard assignment strategy, where the assignment of a sample to a cluster is determined by an $\mathrm{argmax}$ operation that rigidly selects the index of the nearest cluster center, that is, $k^* = \mathrm{argmax}_k \cos(\mathbf{F}'_y, \mu_k)$. This operation functions as a discrete step function, meaning its derivative is zero almost everywhere. Consequently, small perturbations in the cluster centers $\mu_k$ do not change the discrete assignment index $k^*$, blocking the gradient flow effectively. This discontinuity prevents the semantic structure from being optimized according to the final classification objective, making it difficult to handle the partitioning of cluster boundary points and the rectification of clustering errors, especially in continuous class expansion. To achieve end-to-end learning, a continuous differentiable bridge between the CSD and ACS modules is established through soft structural prior assignment.

**Forward Calculation from Semantic to Optimization Objective.** For each class $y \in Y$, the multi-source semantic representation $\mathbf{F}_y \in \mathbb{R}^{m \times d_e}$ is obtained through the MSG module and the pre-trained text encoder $\mathcal{E}(\cdot)$. The process begins by aggregating these features into a stable representation $\hat{\mathbf{F}}_y \in \mathbb{R}^{d_a}$ through Equation (6). Subsequently, this fused representation $\hat{\mathbf{F}}_y$ are normalized to $\mathbf{F}'_y$ and modeled drawn from a K-component mixture of vMF distribution, denoted as:

$$p(\mathbf{F}'_y) = \sum_{k=1}^{K} \pi_k f\left(\mathbf{F}'_y \mid \mu_k, \kappa_k\right), \text{ where } f\left(\mathbf{F}'_y \mid \mu_k, \kappa_k\right) = C_{d_a}(\kappa_k) \exp(\kappa_k \mu_k^T \mathbf{F}'_y) \text{ and } \|\mu_k\|_2 = 1. \tag{31}$$

By utilizing this modeling approach, the soft routing mechanism can be driven to generate the structural prior $r_{y,k}$ for assigning class $y$ to the $k$-th structural cluster derived from Equation (8).

Parallel to routing, the raw semantic representation $\mathbf{f}_{y,j}$ are encoded into the latent space through the global encoder, that is, $\mathbf{h}_{y,j} = \mathcal{E}_{\mathcal{T}}(\bar{\mathbf{f}}_{y,j})$. The core synthesis leverages the routing priors $\mathbf{r}_y$ to modulate the contribution of adapters $\{\mathcal{R}_k\}_{k=1}^K$, yielding latent representation $\mathbf{z}_{y,j}$ of the weighted residual combination derived from Equation (9). To maintain the invariance of core features under noise conditions, the latent vector $\mathbf{z}_{y,j}$ is explicitly split into ausal identity $\mathbf{z}_{y,j}^c$ and non-causal variation $\mathbf{z}_{y,j}^n$ through projections $\mathcal{P}_c$ and $\mathcal{P}_n$. The final synthesized classifier weight $\hat{w}_{y,j}$ is generated by the weight decoder $\mathcal{D}_{\mathcal{W}}$, that is, $\hat{w}_{y,j} = \mathcal{D}_{\mathcal{W}}([\mathbf{z}_{y,j}^c || \mathbf{z}_{y,j}^n])$. The forward process concludes with the total granular loss $\mathcal{L}_{\text{total}}$ derived from Equation (22), which aggregates the reconstruction, cross-modal alignment, and invariance loss.

**End-to-End Gradient Backpropagation.** Following the forward computation of the total loss $\mathcal{L}_{\text{total}}$, the chain rule is applied to derive the end-to-end gradient backpropagation. Based on the composition of the total loss, where $\mathcal{L}_{\mathcal{W} \to \mathcal{W}}$ and $\mathcal{L}'_{\mathcal{W} \to \mathcal{T}}$ are differentiable according to the study (Christensen et al., 2023), and losses $\mathcal{L}'_{\mathcal{T} \to \mathcal{T}}$, $\mathcal{L}'_{\mathcal{T} \to \mathcal{W}}$, and $\mathcal{L}'_{\text{inv}}$ essentially only require one of them to be differentiable, this paper takes the loss $\mathcal{L}'_{\text{inv}}$ as an example to demonstrate differentiability.

**Proposition D.1** (**Differentiability**). *Assuming that the L2 normalization and the denominator of the cosine distance are $\epsilon$-stabilized to avoid zero-norm boundaries, the loss $\mathcal{L}'_{\text{inv}}$ is differentiable with respect to the parameters in the CSD and ACS modules. Furthermore, there is also no gradient detachment between modules, that is, gradients can backpropagate from the latent space through the soft routing mechanism to structural parameters $\{\mu_k, \kappa_k, \pi_k\}$ and representations $\{\mathbf{F}_y\}$ from MSG.*

*Proof.* **Related to Invariance Constraint.** Consider a triplet $(y, j, j')$ in Equation (10), this paper defines:

$$a := \mathcal{D}_{\mathcal{W}}(\mathbf{Z}_{y,j}), \quad b := \mathcal{D}_{\mathcal{W}}(\mathbf{Z}_{y,j \to j'}), \quad \ell := d(a, b). \tag{32}$$

Since $\mathcal{L}'_{\text{inv}}$ is a sum or expectation of $\ell$, it suffices to derive gradients of $\ell$. According to the chain rule, the partial derivatives with respect to $\mathbf{Z}_{y,j}$ and $\mathbf{Z}_{y,j \to j'}$ can be respectively represented as:

$$\frac{\partial \ell}{\partial \mathbf{Z}_{y,j}} = J_{\mathcal{D}_{\mathcal{W}}}(\mathbf{Z}_{y,j})^T \frac{\partial \ell}{\partial a}, \quad \frac{\partial \ell}{\partial \mathbf{Z}_{y,j \to j'}} = J_{\mathcal{D}_{\mathcal{W}}}(\mathbf{Z}_{y,j \to j'})^T \frac{\partial \ell}{\partial b}, \tag{33}$$

where $J_{\mathcal{D}_{\mathcal{W}}}(\mathbf{Z}_{y,j})^T$ denotes the transpose of the Jacobian matrix of the function $\mathcal{D}_{\mathcal{W}}$ at input $\mathbf{Z}_{y,j}$, describing the local geometric rate of change of $\mathcal{D}_{\mathcal{W}}$. Since $\mathcal{D}_{\mathcal{W}}$ is composed of multi-layer MLPs and ReLU activation functions, it is differentiable almost everywhere. Consequently, the Jacobian $J_{\mathcal{D}_{\mathcal{W}}}$ exists and is bounded, ensuring numerical stability.

Since the original branch $\mathbf{Z}_{y,j} = [\mathbf{z}_{y,j}^c || \mathbf{z}_{y,j}^n]$ and the swapped branch $\mathbf{Z}_{y,j \to j'} = [\mathbf{z}_{y,j}^c || \mathbf{z}_{y,j'}^n]$, the concatenation becomes a linear indexing operation, and the gradients are distributed according to the slices, represented as:

$$\frac{\partial \ell}{\partial \mathbf{z}_{y,j}^c}\bigg|_{\text{orig}} = \left(\frac{\partial \ell}{\partial \mathbf{Z}_{y,j}}\right)_c, \quad \frac{\partial \ell}{\partial \mathbf{z}_{y,j}^n}\bigg|_{\text{orig}} = \left(\frac{\partial \ell}{\partial \mathbf{Z}_{y,j}}\right)_n, \quad \frac{\partial \ell}{\partial \mathbf{z}_{y,j}^c}\bigg|_{\text{swap}} = \left(\frac{\partial \ell}{\partial \mathbf{Z}_{y,j \to j'}}\right)_c, \quad \frac{\partial \ell}{\partial \mathbf{z}_{y,j'}^n}\bigg|_{\text{swap}} = \left(\frac{\partial \ell}{\partial \mathbf{Z}_{y,j \to j'}}\right)_n. \tag{34}$$

According to the multi-variable chain rule, the total gradient is equal to the sum of the gradients along all paths. Therefore, the same variable $\mathbf{z}_{y,j}^c$ receives and sums the gradients from two paths, which can be expressed as:

$$\frac{\partial \ell}{\partial \mathbf{z}_{y,j}^c} = \frac{\partial \ell}{\partial \mathbf{z}_{y,j}^c}\bigg|_{\text{orig}} + \frac{\partial \ell}{\partial \mathbf{z}_{y,j}^c}\bigg|_{\text{swap}}, \quad \frac{\partial \ell}{\partial \mathbf{z}_{y,j}^n} = \frac{\partial \ell}{\partial \mathbf{z}_{y,j}^n}\bigg|_{\text{orig}} + \frac{\partial \ell}{\partial \mathbf{z}_{y,j'}^n}\bigg|_{\text{swap}}. \tag{35}$$

Notably, only the gradients involved in the triplet $(y, j, j')$ are shown here. In fact, any $\mathbf{z}_{y,j}^c$ will accumulate gradients from its own original branch and all branches that use it as swapping noise, and $\mathbf{z}_{y,j}^n$ also represents the accumulated gradient. Therefore, the swapping operation does not introduce non-differentiable points and does not detach gradients. According to the mappings $\mathcal{P}_c$ and $\mathcal{P}_n$, $\mathbf{z}_{y,j}$ is decomposed into causal identity $\mathbf{z}_{y,j}^c$ and non-causal variation $\mathbf{z}_{y,j}^n$. Let $\mathbf{z}_{y,j}^c = \mathcal{P}_c(\mathbf{z}_{y,j})$ and $\mathbf{z}_{y,j}^n = \mathcal{P}_n(\mathbf{z}_{y,j})$, then the partial derivative of the loss $\ell$ with respect to the vector $\mathbf{z}_{y,j}$ can be expressed as:

$$\frac{\partial \ell}{\partial \mathbf{z}_{y,j}} = J_{\mathcal{P}_c}(\mathbf{z}_{y,j})^T \frac{\partial \ell}{\partial \mathbf{z}_{y,j}^c} + J_{\mathcal{P}_n}(\mathbf{z}_{y,j})^T \frac{\partial \ell}{\partial \mathbf{z}_{y,j}^n}. \tag{36}$$

**Related to Adaptive Residual.** Next, the gradient is backpropagated to adapters. Equation (9) can be rewritten as:

$$\mathbf{z}_{y,j} = \sum_{k=1}^K r_{y,k} u_{y,j,k}, \quad \text{where } u_{y,j,k} = \alpha \mathbf{h}_{y,j} + (1 - \alpha) \mathcal{R}_k(\mathbf{h}_{y,j}). \tag{37}$$

Since $\mathbf{z}_{y,j}$ is linear in $r_{y,k}$, we can obtain:

$$\frac{\partial \ell}{\partial r_{y,k}} = \left\langle \frac{\partial \ell}{\partial \mathbf{z}_{y,j}}, \frac{\partial \mathbf{z}_{y,j}}{\partial r_{y,k}} \right\rangle = \left\langle \frac{\partial \ell}{\partial \mathbf{z}_{y,j}}, u_{y,j,k} \right\rangle = \left( \frac{\partial \ell}{\partial \mathbf{z}_{y,j}} \right)^T u_{y,j,k}, \tag{38}$$

where $\langle \cdot, \cdot \rangle$ represents the dot product since the result on the left side of the equation is a scalar. This equation shows that as long as $\partial \ell / \partial \mathbf{z}_{y,j} \neq \mathbf{0}$, there exists $\partial \ell / \partial r_{y,k} \neq 0$, thereby the supervision of ACS will directly update the routing probability of CSD, and **there will be no detachment between modules**. Simultaneously, gradients to adapter parameters follow from:

$$\frac{\partial \ell}{\partial u_{y,j,k}} = r_{y,k} \frac{\partial \ell}{\partial \mathbf{z}_{y,j}}, \quad \frac{\partial \ell}{\partial \theta_k} = (1 - \alpha) \left( \frac{\partial \ell}{\partial u_{y,j,k}} \right)^T \frac{\partial \mathcal{R}_k(\mathbf{h}_{y,j})}{\partial \theta_k}. \tag{39}$$

**Related to Structural Prior.** We first obtain the logit of the mixed vMF model according to Equation (31):

$$\ell^{\mathrm{v}}_{y,k} = \log \left( \pi_k C_{d_a}(\kappa_k) \exp(\kappa_k \mu_k^T \mathbf{F}'_y) \right) = \log \pi_k + \log C_{d_a}(\kappa_k) + \kappa_k \mu_k^T \mathbf{F}'_y. \tag{40}$$

Then, Equation (8) is equivalent to $r_{y,k} = \mathrm{Softmax}(\ell^{\mathrm{v}}_{y,1}, \ell^{\mathrm{v}}_{y,2}, \cdots, \ell^{\mathrm{v}}_{y,K})_k$. The softmax backward rule yields:

$$\frac{\partial \ell}{\partial \ell^{\mathrm{v}}_{y,k}} = r_{y,k} \left( \frac{\partial \ell}{\partial r_{y,k}} - \sum\nolimits_{j=1}^{K} r_{y,j} \frac{\partial \ell}{\partial r_{y,j}} \right). \tag{41}$$

Furthermore, the partial derivatives with respect to $\{\mu_k, \kappa_k, \pi_k\}$ in the vMF mixture model are respectively expressed as:

$$\frac{\partial \ell}{\partial \mu_k} = \frac{\partial \ell}{\partial \ell^{\mathrm{v}}_{y,k}} \kappa_k \mathbf{F}'_y, \quad \frac{\partial \ell}{\partial \kappa_k} = \frac{\partial \ell}{\partial \ell^{\mathrm{v}}_{y,k}} \left( \mu_k^T \mathbf{F}'_y + \frac{\partial}{\partial \kappa_k} \log C_{d_a}(\kappa_k) \right), \quad \frac{\partial \ell}{\partial \pi_k} = \frac{1}{\pi_k} \frac{\partial \ell}{\partial \ell^{\mathrm{v}}_{y,k}}. \tag{42}$$

In addition, the gradient with respect to the semantic representation $\mathbf{F}'_y$ can be expressed as:

$$\frac{\partial \ell}{\partial \mathbf{F}'_y} = \sum\nolimits_{k=1}^{K} \frac{\partial \ell}{\partial \ell^{\mathrm{v}}_{y,k}} \kappa_k \mu_k. \tag{43}$$

**Related to Semantic Fusion.** Since $\mathbf{F}'_y$ is the representation obtained by projecting the fused representation $\hat{\mathbf{F}}_y$ onto the unit hypersphere $\mathbb{S}^{d_a-1}$ via L2 normalization, with $\mathbf{F}'_y = \hat{\mathbf{F}}_y / \|\hat{\mathbf{F}}_y\|$ and $\rho = \|\hat{\mathbf{F}}_y\|$, the Jacobian of L2 normalization is:

$$\frac{\partial \mathbf{F}'_y}{\partial \hat{\mathbf{F}}_y} = \frac{1}{\rho} \left( I - \frac{\hat{\mathbf{F}}_y \hat{\mathbf{F}}_y^T}{\rho^2} \right). \tag{44}$$

Therefore, the gradient with respect to the semantic representation $\hat{\mathbf{F}}_y$ can be expressed as:

$$\frac{\partial \ell}{\partial \hat{\mathbf{F}}_y} = \left( \frac{\partial \mathbf{F}'_y}{\partial \hat{\mathbf{F}}_y} \right)^T \frac{\partial \ell}{\partial \mathbf{F}'_y} = \frac{1}{\rho} \left( I - \frac{\hat{\mathbf{F}}_y \hat{\mathbf{F}}_y^T}{\rho^2} \right) \frac{\partial \ell}{\partial \mathbf{F}'_y}. \tag{45}$$

Since $\hat{\mathbf{F}}_y$ is produced by the attention fusion in Equation (6), its gradients with respect to the fusion parameters and the input $\mathbf{F}_y$ are obtained through standard attention backpropagation, thus **completing the backpropagation from $\mathcal{L}'_{\mathrm{inv}}$ to the fusion and routing parameters**. In general, the loss $\mathcal{L}'_{\mathrm{inv}}$ is differentiable with respect to the key parameters in CSD and ACS modules, and there is no gradient detachment between modules, which proves the end-to-end gradient propagation. $\square$

### D.2. Optimization Interference of the Shared Semantic-to-Weight Mapping

**Existing Limitations in the Shared Mapping.** Traditional image-free ZSL methods rely on a shared semantic-to-weight mapping $\Psi : T \to W$. Specifically, the loss $\mathcal{L}_{\mathcal{T} \to \mathcal{W}}$ between the semantic representation $\hat{\mathbf{F}}_s$ and synthesized weights $\hat{w}_s$ is minimized by training a mapping $\hat{w}_s = \Psi(\hat{\mathbf{F}}_s)$ on seen classes $S$, that is, $\mathcal{L}_{\mathcal{T} \to \mathcal{W}} = \sum_{s \in S} d(\Psi(\hat{\mathbf{F}}_s), w_s)$, where $d(a, b) = 1 - a^T b / (\|a\| \|b\|)$. The paper will explain differences in how the shared mapping fits distinct and similar classes.

**Proposition D.2** (**Optimization Interference**). *Consider a shared semantic-to-weight mapping $\hat{w}_s = \Psi(\hat{\mathbf{F}}_s)$ trained on seen classes $S$ by minimizing cosine distance. The optimization of $\Psi$ is dominated by gradient components that are globally consistent across classes (typically corresponding to general features), while local and class-specific gradient components (typically required for subtle features among similar classes) exhibit stronger conflict when aggregated over $S$.*

*Proof.* This paper first transforms the shared semantic-to-weight mapping into an equivalent form, where we consider the original optimization objective to be minimizing the cosine distance, that is, $d(\hat{w}_s, w_s) = 1 - \cos(\hat{w}_s, w_s)$. To facilitate the analysis of gradient properties, the normalized directions are first defined as:

$$\tilde{w}_s = \frac{\Psi(\hat{\mathbf{F}}_s)}{\|\Psi(\hat{\mathbf{F}}_s)\|_2} = \frac{\hat{w}_s}{\|\hat{w}_s\|_2}, \quad \tilde{v}_s = \frac{w_s}{\|w_s\|_2}. \tag{46}$$

Consider the squared Euclidean distance between the two points on a unit sphere:

$$\|\tilde{w}_s - \tilde{v}_s\|_2^2 = (\tilde{w}_s - \tilde{v}_s)^T(\tilde{w}_s - \tilde{v}_s) = \|\tilde{w}_s\|_2^2 + \|\tilde{v}_s\|_2^2 - 2\tilde{w}_s^T\tilde{v}_s. \tag{47}$$

Since both $\tilde{w}_s$ and $\tilde{v}_s$ are located on the unit sphere, we have $\|\tilde{w}_s\|_2^2 = 1$ and $\|\tilde{v}_s\|_2^2 = 1$, as well as the inner product is equal to the cosine similarity, that is, $\tilde{w}_s^T\tilde{v}_s = \cos(\hat{w}_s, w_s)$. Substituting this into the above equation, we obtain the equation:

$$\|\tilde{w}_s - \tilde{v}_s\|_2^2 = 2 - 2\cos(\hat{w}_s, w_s) = 2 \cdot d(\hat{w}_s, w_s). \tag{48}$$

This indicates that minimizing cosine distance is minimizing a squared distance between directions on the unit sphere. Therefore, which directions are learned or neglected can be translated into an analysis of directional variations on the sphere.

To obtain an analytical closed-form solution, we assume the shared mapping $\Psi$ is approximately linear in a local neighborhood (Jacot et al., 2018). Let $\Psi$ be a linear transformation matrix $A \in \mathbb{R}^{d_w \times d_t}$, then $\Psi(\hat{\mathbf{F}}_s) = A\hat{\mathbf{F}}_s$, where $d_w$ and $d_t$ denote the dimension of weights and semantic representations, respectively. Given that strategies such as weight decay and early stopping induce smoothness in the mapping during training, we consider a standard local approximation around the convergence point. Then, the optimization problem can be formulated as a Ridge Regression, expressed as:

$$\min_{\Psi} \mathcal{L}_{\mathcal{T}\to\mathcal{W}} \iff \min_{\Psi} \mathcal{L}_{\text{MSE}} = \min_{\Psi} \sum_{s \in S} \|\tilde{w}_s - \tilde{v}_s\|_2^2 \overset{\approx}{\iff} \min_{W} \mathcal{L}_{\text{Ridge}} = \sum_{s \in S} \|A\hat{\mathbf{F}}_s - \bar{v}_s\|_2^2 + \lambda\|A\|_F^2, \tag{49}$$

where $\bar{v}_s$ is the projection of the target vector $\tilde{v}_s$ onto the tangent space and $\lambda$ is the regularization coefficient.

We stack all training samples $s \in S$ into matrix form, obtaining the input semantic matrix $F' = [\hat{\mathbf{F}}_1, \ldots, \hat{\mathbf{F}}_{|S|}] \in \mathbb{R}^{d_t \times |S|}$ and the target weight matrix $W' = [\bar{v}_1, \ldots, \bar{v}_{|S|}] \in \mathbb{R}^{d_w \times |S|}$. By taking the derivative of $\mathcal{L}_{\text{Ridge}}$ with respect to $A$ and equating it to zero, we obtain the closed-form solution for the optimal solution $A^*$. This process can be expressed as:

$$\frac{\partial \mathcal{L}_{\text{Ridge}}}{\partial A} = 2(AF' - W')(F')^T + 2\lambda A = 0 \implies A^* = W'(F')^T(F'(F')^T + \lambda I)^{-1}. \tag{50}$$

To analyze the structure of matrix $F'(F')^T$, we perform eigendecomposition:

$$F'(F')^T = U\Lambda U^T = \sum_{i=1}^{d_t} \sigma_i^2 u_i u_i^T, \quad \text{where } \sigma_1 \geq \sigma_2 \geq \cdots \geq \sigma_{d_t} \geq 0. \tag{51}$$

Here, $\sigma_i^2$ represents the eigenvalue and larger eigenvalues correspond to the directions where the variation is most significant. $u_i$ denotes the $i$-th principal direction, which determines the general and subtle features based on the value $\sigma_i$. We project the closed-form solution $A^*$ onto the $i$-th principal direction $u_i$ to observe the effect of $A^*$ on the basis vector $u_i$:

$$A^* u_i = W'(F')^T(F'(F')^T + \lambda I)^{-1} u_i = W'(F')^T U(\Lambda + \lambda I)^{-1} U^T u_i = \frac{1}{\sigma_i^2 + \lambda} W'(F')^T u_i. \tag{52}$$

Since $F'$ contains the value $\sigma_i$ by performing a singular value decomposition of $F'$, the magnitude of the term $(F')^T u_i$ is proportional to $\sigma_i$. Finally, we evaluate the predictive capability of the model. Given that the input data $F'$ inherently possesses a magnitude of $\sigma_i$ along the direction $u_i$, which is determined by the data variance, we examine the effect of $A^*$ acting on $F'$ to measure the effective gain. This process can be expressed as:

$$A^* F' = W'(F')^T(F'(F')^T + \lambda I)^{-1} F' \implies G_{\text{gain}}(\sigma_i) \propto \sigma_i \cdot \frac{1}{\sigma_i^2 + \lambda} \cdot \sigma_i = \frac{\sigma_i^2}{\sigma_i^2 + \lambda}. \tag{53}$$

According to the gain $G_{\text{gain}}$, the learning capability of the mapping matrix $A^*$ for the $i$-th semantic principal direction $u_i$ depends on the relative magnitude of the signal strength $\sigma_i^2$ and the regularization strength $\lambda$. For the leading principal components $u_i$ where $i$ is small, the corresponding eigenvalues are much larger than the regularization coefficient, i.e., $\sigma_i^2 \gg \lambda$. Consequently, $G_{\text{gain}} \approx \sigma_i^2/\sigma_i^2 \to 1$, which indicates that the optimizer preserves the information along these directions almost without loss. In contrast, for the lower-ranked tail components $u_j$ where $j$ is large, the eigenvalues are overwhelmed by the regularization, i.e., $\sigma_j^2 \ll \lambda$. As a result, $G_{\text{gain}} \approx \sigma_j^2/\lambda \to 0$, implying that these directions are ignored.

*Remark* D.3 (Extension to Non-linear Deep Networks). While the derivation above assumes a linear mapping for tractability, the conclusion holds for deep non-linear networks. According to the Neural Tangent Kernel (NTK) theory (Jacot et al., 2018), the optimization dynamics of wide neural networks can be approximated by a linear model in the tangent space. Furthermore, the empirical study (Rahaman et al., 2019) has confirmed that deep networks exhibit a universal spectral bias, preferentially learning low-frequency functions while struggling with high-frequency components. Therefore, the linear analysis presented here serves as a simplified approach to highlight the mismatch problem in mixed-granularity scenarios.

For the general features of distinct classes, which are consistently present across the majority of samples, the semantic projections exhibit significant magnitudes, yielding large values $\sigma_i^2$ and gain $G_{\text{gain}}$. During training, the gradients of these features superimpose coherently, thereby dominating the optimization process. In contrast, for similar classes, semantic projections along the directions of subtle features are meaningful only within specific local neighborhoods, while remaining close to zero or exhibiting a random noise distribution for the most classes. When gradients are aggregated over the entire set $S$, the directions required for these subtle features are often inconsistent across classes, which results in extremely small eigenvalues $\sigma_j^2$ and $G_{\text{gain}} \to 0$ by Equation (53). Therefore, optimization interference occurs between the two parts. $\qquad\square$

**Shared Mapping under High Class Density.** With the continuous expansion of classes, the increasing density of both distinct and similar classes causes traditional shared mappings to exacerbate feature interference between them. This paper will analyze two limitations with previous methods in this scenario and explain how our proposed method addresses them.

**Proposition D.4 (General Feature Overlap).** *As the class density increases within a bounded semantic space, the shared mapping $\Psi$ inevitably amplifies the coarseness of general features, leading to irreversible feature overlap for distinct classes.*

*Proof.* Let $F' = [\hat{\mathbf{F}}_1, \ldots, \hat{\mathbf{F}}_{|S|}] \in \mathbb{R}^{d_t \times |S|}$ represent the semantic matrix. From the closed-form ridge solution in Equation (50), the spectral gain $G_{\text{gain}}$ along direction $u_i$ is given by $G_{\text{gain}}(\sigma_i) = \sigma_i^2/(\sigma_i^2 + \lambda)$. Define the set of effectively preserved directions as $\mathcal{I}_\lambda = \{i : \sigma_i^2 \geq c\lambda\}$ and let $r = |\mathcal{I}_\lambda|$ be the effective rank. The learned mapping $A^*$ primarily relies on the $r$-dimensional subspace spanned by $U_{\mathcal{I}_\lambda} = [u_i]_{i \in \mathcal{I}_\lambda}$. As class density $|S|$ increases, the semantic space becomes crowded. For any class $s$, its synthesized weight direction $\tilde{w}_s = \hat{w}_s/\|\hat{w}_s\|_2$ is compressed into this $r$-dimensional general subspace because the subtle features, i.e., lower-ranked tail components, are filtered out by $G_{\text{gain}} \to 0$.

To quantify the impact of increasing class density, consider $N := |S|$ class prototypes on the unit hypersphere $\mathbb{S}^{r-1}$, where more directional points $\{\tilde{w}_s\}_{s \in S}$ need to be accommodated. We use the geometric concept of sphere packing to describe this degree of crowding (Fazeli et al., 2015), that is, if the angular distance between any two class weights $(\tilde{w}_s, \tilde{w}_{s'})$ is at least $\theta$, the total volume $\mathcal{V}(\cdot)$ of their non-overlapping spherical caps cannot exceed the sphere's total volume[14]:

$$\sum_{i=1}^{N} \mathcal{V}(\text{cap}_i(\theta/2)) = N \cdot \mathcal{V}(\text{cap}(\theta/2)) \leq \mathcal{V}(\mathbb{S}^{r-1}). \tag{54}$$

As $N$ increases and $r$ is locked, the volume of spherical cap for each class shrinks, which can be expressed as an integral:

$$\mathcal{V}(\text{cap}(\theta/2)) = \mathcal{V}(\mathbb{S}^{r-2}) \int_0^{\theta/2} (\sin t)^{r-2} \, dt. \tag{55}$$

When $\theta$ is very small, we have $\sin t \approx t$ and:

$$\int_0^{\theta/2} (\sin t)^{r-2} \, dt \approx \int_0^{\theta/2} t^{r-2} \, dt = 2^{1-r} \frac{\theta^{r-1}}{r-1} \implies \mathcal{V}(\text{cap}(\theta/2)) \approx 2^{1-r} \frac{\mathcal{V}(\mathbb{S}^{r-2})}{r-1} \theta^{r-1}. \tag{56}$$

---

[14] In an $r$-dimensional space, the feature direction of each class can be represented as a point on the hypersphere $\mathbb{S}^{r-1}$. To distinguish between two classes, the certain distance of angular separation $\theta$ is required. If we define a spherical cap with an angular radius of $\theta/2$ around the point of each class, these caps must be non-overlapping to ensure separability.

Therefore, the volume of a spherical cap scales as $C_r \theta^{r-1}$, which implies the minimum separation vanishes at a rate:

$$N \cdot \mathcal{V}(\mathrm{cap}(\theta/2)) \lesssim N \cdot C_r \theta^{r-1} \leq \mathcal{V}(\mathbb{S}^{r-1}) \implies \theta^{r-1} \lesssim \frac{\mathcal{V}(\mathbb{S}^{r-1})}{N \cdot C_r} \lesssim \frac{1}{N} \implies \theta \lesssim (\frac{1}{N})^{1/(r-1)}, \tag{57}$$

where "$\lesssim$" means that we ignore constants that depend only on $r$. This geometric limit implies that as $N$ increases and $r$ is locked, the inter-class angular distance $\theta \to 0$, causing severe feature overlap. The general features, which originally provided sufficient cues for distinct classes, become coarse as they are over-shared by too many classes. $\square$

**Proposition D.5** (**Intensification of Interference**). *As the class density increases, the optimization of the shared mapping* $\Psi$ *is increasingly dominated by gradients of distinct classes, which masks the subtle feature gradients of similar classes.*

*Proof.* The total gradient of the shared mapping parameters can be decomposed into general and subtle parts of each class:

$$\nabla \mathcal{L}_{\mathcal{T} \to \mathcal{W}} = \sum_{s \in S} \nabla \ell_s = \sum_{s \in S} \left( g_s^{\mathrm{gen}} + g_s^{\mathrm{sub}} \right), \tag{58}$$

where $g_s^{\mathrm{gen}}$ and $g_s^{\mathrm{sub}}$ denote gradient directions driven by general and subtle features, respectively. To facilitate the analysis, when the density of mixed classes is high, we assume that the general gradient components are consistent, while the subtle gradient components are diverse and locally centered, and different classes are assumed to be uncorrelated or weakly correlated. Therefore, we calculate the sum of the general gradient components $\|G_{\mathrm{gen}}\|$ for all classes, expressed as:

$$\|G_{\mathrm{gen}}\|^2 = \| \sum_{s \in S} g_s^{\mathrm{gen}} \|^2 = \left( \sum_{s \in S} \|g_s^{\mathrm{gen}}\|^2 + \sum_{s \neq s'} \langle g_s^{\mathrm{gen}}, g_{s'}^{\mathrm{gen}} \rangle \right) \propto O(|S|^2) \implies \|G_{\mathrm{gen}}\| \propto O(|S|). \tag{59}$$

In contrast, the sum of the subtle gradient components $\|G_{\mathrm{sub}}\|$ across all classes can be calculated through expectation:

$$\mathbb{E}\|G_{\mathrm{sub}}\|^2 = \mathbb{E}\| \sum_{s \in S} g_s^{\mathrm{sub}} \|^2 = \left( \sum_{s \in S} \mathbb{E}\|g_s^{\mathrm{sub}}\|^2 + \sum_{s \neq s'} \mathbb{E} \langle g_s^{\mathrm{sub}}, g_{s'}^{\mathrm{sub}} \rangle \right) \propto O(|S|) \implies \mathbb{E}\|G_{\mathrm{sub}}\| \propto O(\sqrt{|S|}). \tag{60}$$

The relative strength between signals decays as $\|G_{\mathrm{gen}}\|/\mathbb{E}\|G_{\mathrm{sub}}\| \propto \sqrt{|S|}/|S|$. As the density of classes increases, the ratio approaches zero, which indicates that the total gradient is dominated by the general component, masking the subtle part. $\square$

**Granularity-Aware Adaptation.** To avoid the shared mapping being dominated by general gradient component in mixed-class scenarios, especially at high densities, the framework overcomes the limitations of a shared mapping by transitioning to a granularity-aware adaptive strategy, which mitigates feature overlap and optimization interference through structural decoupling. With this approach, the mapping is no longer a fixed single channel, but rather adapts to different classes.

To counter the sphere packing bound which forces feature overlap under high class density, the CSD module performs a granularity-based partitioning of the semantic manifold. By decomposing the global $S$ classes into $K$ local clusters, we alleviate geometric constraints on the minimum angular distance $\theta$. When class density is high, internal splitting or outward expansion inevitably occurs as discussed in Appendix C.5. In this space, the packing density is determined by the number of samples within each local cluster rather than global expansion, which can be addressed by adaptively increasing $K$.

From an optimization perspective, the ACS module mitigates the gradient dominance by introducing cluster-specific residual adapters $\mathcal{R}_k$, where adaptive priors provide different confidence levels for each class to selectively enhance fine-grained discrimination, decoupling the learning of general and subtle features. The frozen semantic encoder preserves the generalizable components $G_{\mathrm{gen}}$, while the adapters are exclusively optimized to capture the class-specific gradients $G_{\mathrm{sub}}$ within each granularity level. In the view of Ridge Regression, this adaptive prior can be interpreted as increasing the effective gain of fine-grained directions, moving them from the $G_{\mathrm{gain}} \to 0$ ($\sigma_j^2 \ll \lambda$) regime into a learnable range. Moreover, subtle gradients from different similar clusters no longer cancel each other in the shared update, mitigating the gradient-dominance interference under high density and enabling a better balance between generalization and discrimination.

## E. More Experimental Results and Analyses

This section provides a comprehensive supplementary analysis to further validate the effectiveness and robustness of the proposed framework. The experimental setup is first elaborated, including dataset statistics, component replaceability, and implementation details (Appendix E.1). Subsequently, the evaluation is extended to additional datasets (Appendix E.2),

and fine-grained internal module ablation studies are conducted to verify component efficacy (Appendix E.3). To explore the intrinsic properties of the model, this paper conducts in-depth investigations and analyses, including semantic quality applicability, robustness, and parameter sensitivity analysis (Appendix E.4, E.5, and E.6). In addition to these quantitative results, qualitative visualizations are provided to intuitively illustrate the underlying mechanisms and effectiveness (Appendix E.7), accompanied by specific case studies of success and failure instances (Appendix E.8). Finally, the framework is further explored within a class-incremental learning scenario to validate its plasticity (Appendix E.9).

### E.1. Experimental setup

**Diverse Datasets.**   To comprehensively evaluate the effectiveness and robustness of the proposed framework, most experiments are conducted on three benchmark datasets, CUB, AWA2, and SUN, covering various granularities and domains. CUB focuses on bird species, which contains 11,788 images spanning 200 fine-grained classes. AWA2 is a coarse-grained dataset containing 37,322 images of 50 animal classes. SUN is a large-scale scene recognition dataset comprising 14,340 images from 717 scene classes. To further verify effectiveness and generalization, two additional datasets are incorporated, aPY (Farhadi et al., 2009) and FLO (Nilsback & Zisserman, 2008). aPY combines images from Pascal VOC and Yahoo, which contains 15,339 images across 32 coarse-grained classes. The dataset is divided into 20 seen classes and 12 unseen classes (Reed et al., 2016), known as diverse object scales and background clutter. FLO, consisting of 8,189 images of 102 flower classes, is a fine-grained dataset where there are 82 seen classes and 20 unseen classes (Xian et al., 2018), posing a challenge due to small inter-class visual variations. In addition, the large-scale ImageNet (Deng et al., 2009) dataset covering 1,000 object classes is utilized as the source domain (seen classes to train the model) for the distribution shift generalization.

**Replaceable LLMs and Embedding Models.**   The generation of multi-source semantics is not limited to a specific LLM or embedding model, and Appendix E.4 demonstrates the versatility and broad applicability of our framework, where GPT-4o and CLIP are selected as the optimal LLM and embedding model for other experiments, respectively. As a dependency tool for class description generation, LLMs are divided into two groups: API-query models and open-source models, where the former includes GPT-3.5-turbo, GPT-4o-mini, Gemini-2.5-flash, and GPT-4o (Achiam et al., 2023), and the latter includes LLaMA-3.1 (8B) (Touvron et al., 2023) and Qwen-2.5 (7B) (Bai et al., 2023).

The frozen embedding models that map text descriptions to representation vectors are also interchangeable, including two standard embedding models (CLIP and SBERT) and two open-source LLMs (LLaMA and Qwen). CLIP obtains a joint embedding space for images and attributes by contrastive learning, pulling the embeddings of matching image-text pairs closer in space while pushing mismatched pairs apart. SBERT is designed to produce meaningful sentence embeddings. For the generative backbones LLaMA and Qwen, to leverage their representational capability, the paper computes high-quality text embeddings by applying mean pooling to the output hidden states of the last transformer layer.

**Flexible Source of Class Attributes.**   Most baseline methods rely on class attributes, which are primarily derived from experts and can also be implemented using Wiki2Vec (Yamada et al., 2018), and ConceptNet (Speer & Lowry-Duda, 2017). Wiki2Vec is used to generate word and entity embeddings from the entire corpus of Wikipedia. ConceptNet Numberbatch is a knowledge-enhanced embedding that integrates ConceptNet commonsense relations with distributional semantics learned from large text corpora. The standard attributes of expert-constructed are 312, 85, and 102 dimensions for CUB, AWA2, and SUN, respectively, while the embeddings from others both have 300 dimensions.

**Implementation Details.**   In the stage of multi-source semantic generation, to ensure time efficiency and meet the limitation on the number of context tokens, this paper adopts a parallel query strategy, setting each batch to include five classes. On average, a full LLM inference over the entire dataset takes about 8, 15, and 25 minutes for AWA2, CUB, and SUN, respectively. The network architecture starts with obtaining the weights of the base classifier, which is trained for 100 epochs on the seen classes using the cross-entropy loss function and the Adam optimizer with a learning rate of $10^{-4}$, and then used as input to the subsequent weight-synthesis branch. Subsequently, the adapter is made up of two-layer MLPs coupled with a ReLU activation function. The latent space dimensions obtained after applying the causal invariance mechanism are 2048, 2048, and 4096 for CUB, AWA2, and SUN, respectively, which are the same as the latent space dimensions of the weight branch. The training process consists of 1000 epochs in all datasets and employs an early stopping strategy, where all initial weights in linear layers follow a Gaussian distribution with a mean of 0 and a standard deviation of 0.02. The Adam optimizer is used with $\beta_1 = 0.9$ and $\beta_2 = 0.999$. The batch sizes are set to 16, 10, and 24 for CUB, AWA2, and SUN, respectively. All experiments are conducted on a 32-core AMD Ryzen 9 7950X CPU @ 4.5GHz and an NVIDIA GeForce RTX 4090 GPU. The training times are approximately 46, 38, and 123 seconds for CUB, AWA2, and SUN, respectively.

*Table 7.* Comparative top-1 accuracy **T1** (%) and harmonic mean **H** of our framework with other methods on aPY and FLO datasets.

| Dataset | | aPY | | FLO | |
|---|---|---|---|---|---|
| Method | | T1 | H | T1 | H |
| Expert-based | ConSE | 24.7±0.3 | 4.7±0.1 | 30.9±1.0 | 0.6±0.1 |
| | COSTA | 37.3±0.1 | 0.0±0.0 | 36.6±0.8 | 0.0±0.0 |
| | SubReg | 33.9±3.3 | 0.0±0.0 | 38.1±5.4 | 0.1±0.3 |
| | wDAE | 32.8±3.2 | 0.1±0.1 | 38.9±6.1 | 0.0±0.0 |
| | VGSE | 37.8±0.6 | 22.4±0.9 | 36.0±2.0 | 34.9±2.0 |
| | ICIS | 34.5±2.1 | 18.5±2.0 | 48.6±0.8 | 46.0±1.0 |
| LLM-generated | ConSE | 40.5±0.3 | 4.7±0.1 | 35.7±0.3 | 0.5±0.1 |
| | COSTA | 48.4±0.4 | 0.0±0.0 | 47.1±0.5 | 0.0±0.0 |
| | SubReg | 46.3±2.3 | 0.1±0.1 | 48.6±4.5 | 1.2±0.6 |
| | wDAE | 45.0±1.6 | 0.1±0.0 | 50.9±5.4 | 0.0±0.0 |
| | VGSE | 44.8±0.4 | 26.6±1.5 | 43.6±1.1 | 37.9±2.2 |
| | ICIS | 44.6±2.3 | 19.2±1.8 | 51.9±1.3 | 45.4±1.5 |
| | CEMIL | 48.6±2.5 | 23.9±1.6 | 54.3±1.3 | 50.4±1.2 |
| | **Ours** | **51.2±1.6** | **32.7±1.5** | **61.8±0.9** | **55.1±1.2** |
| | ΔImp | 5.35% | 22.9% | 13.8% | 11.0% |

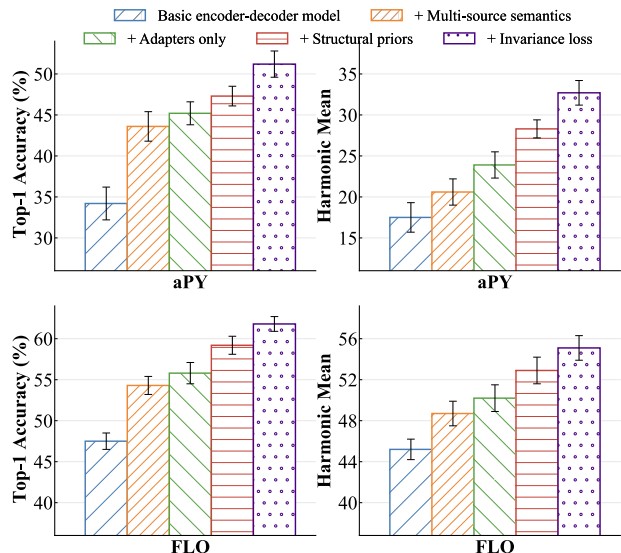

*Figure 7.* Ablation studies on the core components in terms of top-1 accuracy (%) and harmonic mean on aPY and FLO datasets.

### E.2. Extended Comparisons and Analyses on Additional Datasets

**Baseline Descriptions and Applicability Modifications.** The baseline methods are described in detail here and their applicability is illustrated in the image-free setting, as mentioned in Section 4.2. ConSE (Norouzi et al., 2014) maps images into a semantic embedding space by computing a convex combination of the semantic vectors of seen classes, weighted by the posterior probabilities predicted by a pre-trained classifier. COSTA (Mensink et al., 2014) estimates classifiers for unseen labels by computing a weighted combination of existing classifiers, where the weights are derived directly from the (normalized) co-occurrence statistics of visual concepts in image data. They are the same as the previous implementations.

For wDAE (Gidaris & Komodakis, 2019), it is trained using an episodic procedure to mimic the few-shot task at the test time. For each training episode, the model is trained with a reconstruction loss and a cross-entropy loss. To adapt the image-free ZSL task, the initial weight is provided by applying MLPs as a predictor and the cross-entropy loss is discarded. For SubReg (Akyürek et al., 2021), referring to the modified method in the study (Christensen et al., 2023), the subspace regularizers are adapted as an auxiliary loss function to classifiers during training. Specifically, for each predicted unseen classifier $w_u$, the projection in the subspace of seen classifiers is calculated using a projection matrix $P_s$. The loss is defined as the distance between the original predicted weight and its projection, i.e., $\mathcal{L} = \sum_{u \in U} d(\mathbf{w}_u, P_s^T \mathbf{w}_u)$.

For VGSE (Xu et al., 2022), semantic embeddings are learned by analyzing seen classes, and the semantic embeddings of unseen classes are obtained by embedding similarity, mainly including two strategies: weighted average (WAvg) and similarity matrix optimization (SMO). This framework uses the SMO strategy to learn a similarity mapping $r$ between an unseen class and all seen classes, generating the attributes of unseen classes. For ICIS (Christensen et al., 2023), it is the ZSL task that does not require images, consisting of two encoders and decoders, and synthesizes classifier weights for unseen classes by learning the semantic-to-weight mapping. CEMIL (Lyu et al., 2025) leverages LLMs to autonomously generate multi-view class descriptions, which are refined through contextual attention and ultimately mapped to classifier weights.

**Results and Analyses.** To fully substantiate the effectiveness of the proposed framework, comparative evaluations are extended to the aPY and FLO datasets, covering a range from coarse-grained to fine-grained visual domains. As evidenced by the quantitative results presented in Table 7, the proposed method consistently demonstrates superior performance compared to baselines across both benchmarks. Furthermore, to discover the source of these performance gains, an in-depth ablation analysis of the core components is conducted, as illustrated in Figure 7. The experimental trends once again highlight the necessity of adaptive granularity partitioning and intervention, and the effectiveness of synergy between modules.

*Table 8.* Ablation study on the three core modules (MSG, CSD, and ACS) in terms of top-1 accuracy **T1** (%) and harmonic mean **H**. The first column in the table denotes the three core modules (MSG, CSD, and ACS), and the second column indicates the finer-grained components within each module, which are derived from the acronyms of the subtitles (except for "LOSS") in Section 3.

| Dataset | | | CUB | | AWA2 | | SUN | |
|---|---|---|---|---|---|---|---|---|
| | | Internal Module Ablation | T1 | H | T1 | H | T1 | H |
| - | - | **Our full framework** | **71.8±0.6** | **62.5±0.6** | **77.2±1.3** | **68.8±1.2** | **63.3±0.9** | **41.2±0.7** |
| MSG Module / CA | | 1) w/o reliable attribute ontology | 69.6±0.7 | 61.3±0.7 | 72.9±1.8 | 65.3±1.4 | 60.3±1.1 | 38.2±0.9 |
| | | 2) w/o transferability filtering | 70.4±0.6 | 61.8±0.6 | 74.3±1.7 | 67.1±1.5 | 62.3±0.8 | 40.3±0.9 |
| | | 3) w/o discriminability filtering | 70.5±0.5 | 61.8±0.6 | 75.8±1.5 | 68.1±1.0 | 61.9±1.3 | 40.0±1.0 |
| MSG Module / ISE | | 4) w/o interventional expansion | 68.2±0.7 | 61.0±0.7 | 70.5±2.1 | 63.8±1.9 | 59.5±1.2 | 37.9±1.1 |
| | | 5) only generic prompt ensemble | 69.4±0.8 | 61.6±0.7 | 71.9±1.9 | 65.1±1.7 | 60.7±0.8 | 39.1±1.0 |
| | | 6) w/o style intervention | 70.2±0.6 | 62.1±0.5 | 75.1±1.5 | 67.9±1.5 | 62.5±0.9 | 40.8±0.9 |
| | | 7) w/o context intervention | 69.8±0.7 | 61.5±0.5 | 73.8±1.6 | 66.3±1.5 | 61.2±1.0 | 39.3±0.9 |
| CSD Module / SF | | 1) w/o semantic fusion (remove) | 69.5±0.8 | 61.0±0.8 | 73.9±1.6 | 66.8±1.3 | 60.7±1.3 | 39.1±0.8 |
| | | 2) w/o semantic fusion (replace) | 70.8±0.5 | 62.2±0.6 | 76.4±1.3 | 67.2±1.2 | 62.3±1.1 | 40.2±0.5 |
| | | 3) w/o attention refinement | 71.1±0.4 | 62.3±0.6 | 75.5±1.5 | 67.0±1.4 | 61.8±1.0 | 40.2±0.6 |
| CSD Module / CSP | | 4) w/o soft routing | 66.5±1.2 | 60.2±1.1 | 68.9±2.1 | 62.1±1.6 | 58.2±1.5 | 35.9±1.3 |
| | | 5) w/o learnable parameters | 68.6±0.9 | 60.7±1.0 | 72.6±1.5 | 65.1±1.4 | 60.5±1.2 | 38.6±1.1 |
| ACS Module / FAR | | 1) w/o fine-grained adaptive residuals | 65.6±1.2 | 59.8±1.2 | 72.3±2.1 | 63.9±1.8 | 60.2±1.2 | 37.8±1.0 |
| | | 2) w/o frozen parameters of global backbone | 67.2±1.1 | 60.3±1.1 | 70.9±1.8 | 62.8±1.7 | 58.6±1.4 | 35.6±1.3 |
| ACS Module / LOSS | | 3) w/o cosine distance | 63.2±1.4 | 2.7±0.4 | 66.7±2.3 | 16.8±1.8 | 58.8±1.2 | 0.0±0.0 |
| | | 4) w/o weight-to-semantic $\mathcal{L}'_{\mathcal{W}\to\mathcal{T}}$ | 69.6±0.7 | 61.3±0.7 | 74.6±1.7 | 66.8±1.5 | 61.3±1.2 | 40.1±1.0 |
| | | 5) w/o unseen class semantics for training | 71.2±0.8 | 62.3±0.6 | 76.2±1.5 | 68.3±1.4 | 62.8±1.0 | 41.0±0.9 |

## E.3. Internal Module Ablation Studies

To further validate the necessity and specific contributions of each component, this section presents a detailed investigation into the internal components of the three core modules. In contrast to the transition from the base encoder-decoder to the full framework shown in Section 4.3, since the details within modules are difficult to clearly explain using a step-by-step evolution, each ablation study is conducted starting from the full framework, removing or replacing one module at a time to individually analyze its contribution. The ablation results within the three modules are shown in Table 8.

**Internal Ablation of the MSG Module.** To evaluate the contribution of each component within the MSG module, detailed ablation studies are conducted while keeping the subsequent CSD and ACS modules in optimal configurations. The exploration is twofold, focusing on the reliability of the attribute ontology and the necessity of decoupled causal interventions. First, to validate the filtering mechanism of the reliable attribute ontology, three variants are constructed. 1) Utilize the raw universal ontology without filtering. 2) Remove the transferability filtering mechanism. 3) Remove the discriminability filtering mechanism. Second, to verify that performance gains originate from the explicit disentanglement of spurious correlations rather than mere data augmentation, the process of interventional expansion is ablated into three parts. 4) Utilize the single canonical description only. 5) Remove the intervention based on linguistic styles, i.e., keep context interventions only. 6) Remove the intervention based on environmental contexts, i.e., keep style interventions only.

Quantitative results demonstrate that the configuration of the MSG module yields the most robust performance across all metrics and datasets. Regarding the ontology, the raw universal ontology introduces significant noise and hallucinations. Filtering solely by transferability effectively removes random sampling noise but retains attributes that lack classification ability, whereas filtering solely by discriminability captures distinctive features but fails to eliminate non-visual hallucinations and capture transferable common knowledge. The combination of both metrics achieves the necessary balance for visually grounded descriptions. From the perspective of causal intervention, relying solely on the canonical description suffers from the visual diversity and overfitting to specific patterns. The style and context intervention improve robustness against textual variations and mitigate the bias between background and object features, respectively. The superior performance of the full model confirms that visually identifying a class requires robustness against both the style and context. Notably, generic prompt ensembling increases description diversity, but does not identify nuisance factors or constrain them as irrelevant to class identity, thus introducing extra semantic variance rather than separating class-intrinsic information. The impact of semantic quality generated under different LLMs and embedding models on experimental results is shown in Appendix E.4.

**Internal Ablation of the CSD Module.** To validate the effectiveness of each component within the CSD module, which bridges the semantic space and classifier synthesis, ablation studies are performed on semantic fusion and structural routing. For fusion strategies, two variants are compared to validate the necessity of the proposed fusion. 1) Remove the fusion component and keep the raw multi-source input directly utilized for structural discovery without aggregation. 2) Replace the fusion component with generated canonical descriptions of the MSG module. 3) Remove the attention refinement and just use mean pooling for multi-source embeddings. For the structure mechanism, the investigation compares: 4) the soft routing based on vMF posterior probabilities against a hard assignment strategy using "argmax". Additionally, the structure is changed by: 5) replacing the learnable vMF mixture model with a non-parametric spherical K-means.

The results in Table 8 show that the proposed framework yields the best structural priors. For semantic fusion, using the raw multi-source input leads to fragmented clusters due to unaligned stylistic or contextual variations, while the mean pooling method blends intervention with facts, disturbing the core signal. Our method based on attention refinement outperforms both in suppressing outlier instances and capturing semantic dependencies, while representing semantic consensus and visual manifold integrity in contrast to single canonical descriptions for each class. For structural assignment, the hard approach causes performance degradation for classes at cluster boundaries by forcing reliance on a single adapter, while the soft approach enables a weighted collaboration of adapters, effectively handling semantic ambiguity. Furthermore, learnable model parameters allow cluster centers to evolve during training, providing a dynamic fit to the semantic-to-weight mapping.

**Internal Ablation of the ACS Module.** This study investigates the framework effectiveness and training objectives of the ACS module using a two-fold ablation strategy, that is, the parameter optimization and loss components. Specifically, the former refers to the learning and training strategy related to the global backbone. 1) Remove fine-grained adaptive residuals and only retain the global backbone. 2) Use the global backbone for full fine-tuning instead of freezing parameters of the semantic encoder. The latter focuses on the analysis of each component of the loss function derived from Equation (22)[15]. 3) Replace the cosine distance using L2 distance. 4) Remove the loss of weight-to-semantic verification $\mathcal{L}'_{\mathcal{W}\to\mathcal{T}}$. 5) Remove the unseen class parts contained in the semantic reconstruction term $\mathcal{L}'_{\mathcal{T}\to\mathcal{T}}$ derived from Equation (19) during training.

Empirical evidence highlights the critical role of the applied training strategy and loss function. In the architectural comparison, the variant of retaining only global backbone fails to capture fine-grained distinctions due to its rigid mapping, whereas fine-tuning the semantic encoder makes the existence of the adapter redundant for the final decision, evident in the performance comparable to the method without adapters. The proposed method successfully navigates this trade-off by only utilizing the frozen semantic encoder as a stable anchor while allowing adapters to capture local nuances. In terms of losses, the distance metric aims to reconstruct and align from an angular perspective, reducing the model's bias towards seen classes, thereby greatly improving the harmonic mean compared to the L2 distance. In addition, the introduction of the weight-to-semantic loss bridges the modality gap, forcing the establishment of generative verification from the weight space to the semantic space. Notably, the performance generated by semantic reconstruction with or without unseen classes during training is basically the same, which indirectly suggests that our method is effective in semantic reconstruction, that is, whether or not unseen classes are reconstructed, the generated latent semantic representations are similar.

By absolving the internal of three modules, it can be observed that the interventional expansion, soft routing, and fine-grained adapters have a significant impact on the performance of the framework. In addition, the reliable attribute ontology and learnable clustering centers, among other components, suggest their respective contributions in the mixed classification.

**Analysis of Training Strategies.** In addition to the above ablation studies with accuracy, this paper also statistically compares the training efficiency of whether or not the global backbone parameters are completely frozen. The number of trainable parameters and training time are reported in Table 9. Freezing or fine-tuning all parameters is detrimental to model performance to some extent, which is mainly because the former cannot make causal invariance constraints effective, while the latter absorbs the design function of adapters. Consequently, our framework utilizes the frozen encoder to make adapter residuals effective, while other fine-tuning ensures that causal properties are reflected in the weight mapping and synthesis.

*Table 9.* Comparative evaluation of training strategies.

| Dataset | Strategy | Param | Time | T1 |
|---------|----------|-------|------|------|
| **CUB** | Frozen | 21M | 18s | 69.3 |
| | Ours | 53M | 46s | **71.8** |
| | Fine-tuned | 54M | 49s | 67.2 |
| **AWA2** | Frozen | 13M | 14s | 74.8 |
| | Ours | 44M | 38s | **77.2** |
| | Fine-tuned | 45M | 40s | 70.9 |

---

[15]Here, the reconstruction $\mathcal{L}'_{rec}$ and the semantic-to-weight alignment $\mathcal{L}'_{\mathcal{T}\to\mathcal{W}}$ are not included for ablation. This is because the former is the information preservation for the basic encoder-decoder, and the latter is the key to the weight synthesis, both of which are necessary.

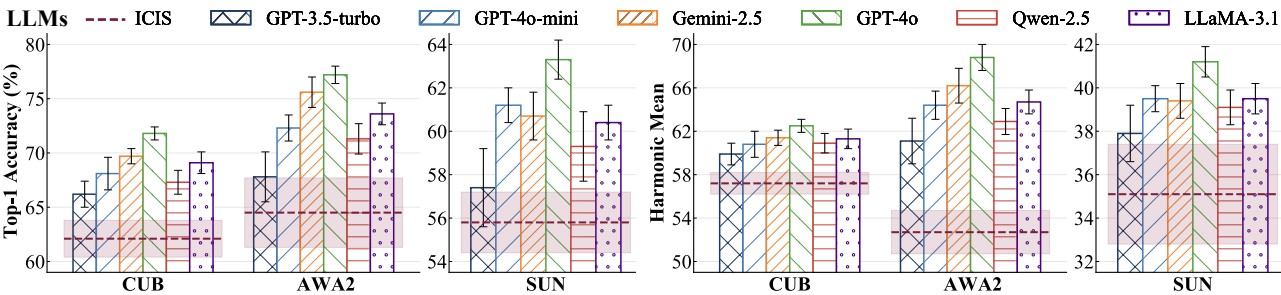

*(a)* Comparison of our method with ICIS across different LLMs while keeping CLIP fixed as the embedding model.

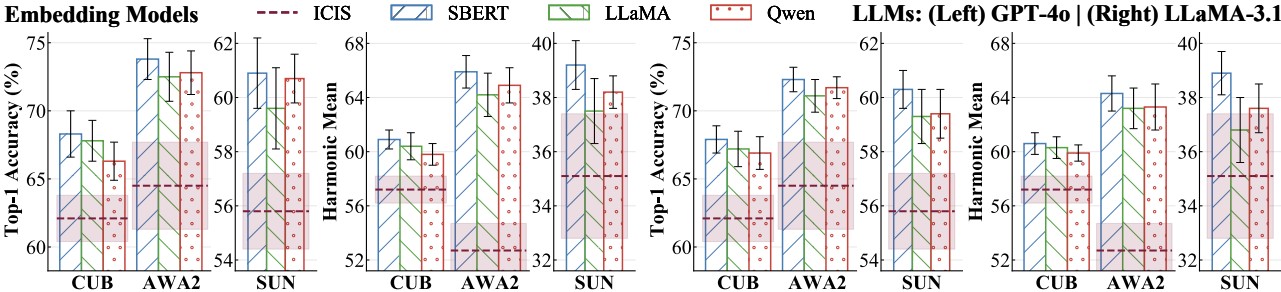

*(b)* Comparison of our method with ICIS across different embedding models while keeping either GPT-4o (the two subplots on the left, top-1 accuracy and harmonic mean) or LLaMA-3.1 (the two subplots on the right, top-1 accuracy and harmonic mean) as the base LLM.

*Figure 8.* Comparative top-1 accuracy **T1** (%) and harmonic mean **H** of the proposed framework under the replaceable options for different LLM generators and embedding encoders, conducted by fixing either the embedding model or the LLM and varying the other component.

## E.4. Semantic Applicability Analysis

To explore the applicability and dissect the determinants of semantic quality, a comprehensive semantic relevance analysis is conducted from two factors: the prompting strategy for querying LLMs and the text encoder for embedding descriptions.

**Prompting Strategies for Semantic Quality.** A core contribution of this paper is moving beyond raw class descriptions to canonical and intervened semantics. To quantify the gain from these strategies, three additional prompting strategies are evaluated. 1) Use only class labels for prompting. 2) Use simple prompts to instruct LLMs. 3) Instruct LLMs with CoT principles (ours). The comparable top-1 accuracy (**T1**) and harmonic mean (**H**) are shown in Table 10. It can be observed that both label-based and simple prompting strategies

*Table 10.* Comparative evaluation of prompting strategies.

| Prompting Strategy | CUB | | AWA2 | | SUN | |
|---|---|---|---|---|---|---|
| | **T1** | **H** | **T1** | **H** | **T1** | **H** |
| Class labels | 60.7 | 57.8 | 63.2 | 59.3 | 54.8 | 35.1 |
| Simple prompts | 68.5 | 60.3 | 72.2 | 65.8 | 60.9 | 38.6 |
| **CoT prompts** | **71.8** | **62.5** | **77.2** | **68.8** | **63.3** | **41.2** |

lead to performance degradation, primarily due to extreme semantic sparsity and the inadequacy of the provided knowledge to support the semantic-to-weight mapping. The CoT-based prompting method forces the model to focus on observable details through step-by-step inference, thereby obtaining more accurate knowledge that is truly helpful for the task.

**Impact of Various LLM Generators and Embedding Encoders.** The quality of the initial semantic pool depends not only on the different prompting strategies mentioned above, but also on the generative capability of LLMs and the representational power of the text encoder. This experiment tests the performance of the proposed framework by swapping these replaceable components, which are derived from Appendix E.1. The experimental results are shown in Figure 8, where Figure 8a illustrates the impact of different LLMs on the experimental results when using CLIP as the embedding model, and Figure 8b explores the impact of different embedding models when the LLM is fixed. It can be observed that while stronger LLMs like GPT-4o yield slight improvements in the semantic richness of the descriptions, our framework maintains robust performance even with weaker API-query models (e.g., GPT-3.5-turbo) and open-source models (e.g., LLaMA-3.1). This indicates that the CSD module effectively mitigates the noise from weaker generators by discovering latent consensus. Similarly, while CLIP provides the best visual-semantic alignment, the framework remains functional with the pure text encoder SBERT and generative LLMs (i.e., LLaMA and Qwen), proving its applicability across different feature spaces.

*Table 11.* Comparison of our framework with ICIS and CEMIL across the proportion of expanded classes in terms of top-1 accuracy (%) and harmonic mean. The row "SeenPro" indicates the two proportions of seen classes in this experiment, and the second column denotes the proportion of classes to be expanded. The **bold** text indicates the optimal results at different proportions of expanded classes.

| Dataset | | CUB | | | | | | SUN | | | | | |
|---|---|---|---|---|---|---|---|---|---|---|---|---|---|
| SeenPro | | 10% | | | 30% | | | 10% | | | 30% | | |
| Method | | ICIS | CEMIL | Ours | ICIS | CEMIL | Ours | ICIS | CEMIL | Ours | ICIS | CEMIL | Ours |
| **Top-1 Accuracy** | 20% | 37.2±2.2 | 38.4±1.2 | **39.3±1.9** | 60.4±1.8 | 61.2±0.9 | **63.0±1.8** | 55.6±2.0 | 56.8±2.9 | **58.6±3.0** | 65.9±2.5 | 67.0±1.3 | **68.8±1.2** |
| | 30% | 29.6±1.6 | 31.1±2.2 | **36.3±1.2** | 50.1±2.7 | 53.3±1.4 | **58.3±1.0** | 47.1±2.6 | 51.9±1.7 | **54.9±1.5** | 61.7±1.3 | 63.7±1.5 | **66.5±0.9** |
| | 40% | 26.8±2.1 | 28.8±0.9 | **34.8±1.3** | 42.4±2.9 | 48.7±1.7 | **54.2±2.5** | 42.5±1.7 | 47.3±0.8 | **51.6±1.2** | 56.1±1.7 | 59.4±1.7 | **63.8±1.1** |
| | 55% | 24.3±1.2 | 26.3±1.1 | **30.0±1.1** | 38.8±1.7 | 44.0±1.9 | **50.7±1.7** | 37.2±1.0 | 41.6±0.4 | **45.3±1.3** | 48.6±1.1 | 52.2±0.7 | **57.7±1.1** |
| | 70% | 23.3±1.4 | 25.2±1.2 | **28.9±1.1** | 38.0±2.0 | 42.8±1.8 | **49.2±1.0** | 34.4±0.9 | 38.6±0.5 | **43.2±0.8** | 47.5±1.1 | 51.3±0.8 | **56.2±1.2** |
| | 85% | 23.1±0.5 | 25.0±0.9 | **27.3±0.5** | 36.7±1.5 | 41.2±1.6 | **47.3±1.5** | 32.3±0.9 | 35.5±0.3 | **40.7±0.6** | 45.3±0.9 | 48.9±0.5 | **53.3±1.0** |
| | 100% | 20.8±0.7 | 22.9±0.6 | **26.4±0.7** | 35.7±0.7 | 40.1±1.6 | **45.8±0.9** | 31.2±1.0 | 34.7±0.8 | **41.2±0.7** | 44.2±0.8 | 47.0±0.8 | **51.8±0.9** |
| **Harmonic Mean** | 20% | 48.2±2.9 | 49.5±1.1 | **50.3±1.8** | 52.0±1.2 | 53.9±0.4 | **54.9±1.6** | 37.2±2.3 | 37.0±1.2 | **39.1±1.7** | 42.8±1.3 | 44.5±1.2 | **44.8±1.0** |
| | 30% | 32.4±1.4 | 39.6±2.4 | **44.6±1.6** | 49.4±2.1 | 51.1±0.2 | **53.2±1.6** | 32.9±2.4 | 34.1±1.2 | **36.7±0.7** | 40.5±1.1 | 41.6±0.6 | **43.3±1.1** |
| | 40% | 30.1±2.0 | 34.3±1.1 | **40.7±1.5** | 44.7±1.4 | 46.9±0.9 | **50.6±1.3** | 31.1±1.9 | 32.2±1.4 | **35.2±0.8** | 37.2±1.4 | 38.5±0.7 | **40.4±1.0** |
| | 55% | 23.2±1.1 | 26.1±1.5 | **34.1±0.8** | 42.5±1.8 | 44.2±1.0 | **48.7±1.0** | 24.6±1.1 | 27.5±1.0 | **31.1±1.0** | 34.9±0.9 | 35.6±0.7 | **37.1±0.6** |
| | 70% | 22.7±0.6 | 24.9±0.8 | **31.9±0.9** | 41.2±0.9 | 42.9±0.8 | **46.3±1.1** | 22.9±1.2 | 26.1±0.7 | **29.8±0.9** | 33.8±0.9 | 34.7±0.3 | **36.9±0.9** |
| | 85% | 21.4±1.2 | 23.4±0.9 | **27.8±0.7** | 41.0±0.7 | 41.4±0.6 | **44.3±1.0** | 21.8±0.8 | 25.5±1.0 | **28.9±0.6** | 32.8±0.7 | 33.8±0.6 | **35.1±0.8** |
| | 100% | 20.2±1.0 | 22.7±0.9 | **26.3±0.9** | 39.6±0.7 | 41.1±0.8 | **43.5±0.7** | 20.2±1.0 | 24.2±0.7 | **28.5±0.7** | 31.1±0.8 | 32.4±0.4 | **34.0±0.7** |

## E.5. Further Robustness Analysis

To complement the standard evaluations in the main text, this supplemental analysis measures the capability to expand classes under conditions of data scarcity, that is, the proportion of identifiable unseen concepts relative to the limited source data, to determine whether the structural advantages of the framework remain when the knowledge source is shrunk while the generalization target is expanded. Specifically, the sparsity of seen classes is reduced to 10% and 30%, while the scale of the target unseen classes is also adjusted. Table 11 show the performance comparison with ICIS and CEMIL, suggesting that the proposed framework exhibits a higher expansion threshold than competing methods. This data efficiency, reflected in achieving performance comparable to the baseline trained with more data, while using less training data, suggests that the intervention successfully disentangles essential features from environmental noise. By focusing on these invariant properties, the model maintains high discriminability and supports broad class expansion, validating its applicability in the real world.

In addition to the generalization analysis presented in Section 4.4, this paper also uses ConceptNet (derived from Appendix E.1) as an additional knowledge source for baselines, besides Wiki2Vec. Table 12 shows the top-1 accuracy **T1** (%) on the three datasets, which suggests that our method consistently outperforms baselines, effectively preserving structural consistency and capturing core discriminative features.

*Table 12.* **T1** of our method and baselines with ConceptNet attributes when shifting from ImageNet to the target.

| Target | wDAE | VGSE | ICIS | CEMIL | Ours |
|---|---|---|---|---|---|
| CUB | 6.4±0.9 | 11.1±0.1 | 7.5±0.6 | 16.3±1.2 | **22.9±0.9** |
| AWA2 | 62.7±2.2 | 64.7±0.1 | 62.0±3.1 | 72.1±1.6 | **76.5±1.2** |
| SUN | 11.4±0.8 | 11.3±0.0 | 13.4±0.8 | 18.5±1.2 | **24.3±0.7** |

## E.6. Further Sensitivity Analysis

The bottleneck dimension $d_b$ within the residual adapters determines the capacity to learn fine-grained residual features without disrupting the global knowledge of the partially frozen backbone. To evaluate its impact, the dimension $d_b$ is varied across the set $\{d_z/2^n\}_{n=0}^6$, where $d_z$ is the dimension after global encoding $\mathcal{E}_{\mathcal{T}}(\cdot)$. As shown in Table 13, performance steadily improves as $d_b$ increases from $n = 6$ to $n = 2$, attributing to the enhanced capacity for capturing fine-grained residuals. However, further increasing $d_b$ beyond $d_z/4$ leads to potential overfitting because the excessive parameterization distorts the well-structured space and impairs the model's ability to generalize to unseen classes. Consequently, $d_b = d_z/4$ is

*Table 13.* Comparative top-1 accuracy and harmonic mean of our framework across various bottleneck dimensions. "Dim" denotes the dimension of $d_z/2^n$, where $n \in \{0, 1, \ldots, 6\}$.

| Dim (n) | | 0 | 1 | 2 | 3 | 4 | 5 | 6 |
|---|---|---|---|---|---|---|---|---|
| CUB | T1 | 69.8 | 70.9 | **71.8** | 70.2 | 69.1 | 67.8 | 66.7 |
| | H | 61.7 | 62.2 | **62.5** | 61.8 | 61.1 | 60.5 | 60.2 |
| AWA2 | T1 | 73.9 | 75.2 | **77.2** | 74.8 | 73.6 | 73.1 | 72.9 |
| | H | 66.1 | 67.9 | **68.8** | 67.2 | 66.4 | 65.2 | 64.6 |
| SUN | T1 | 60.9 | 61.8 | **63.3** | 62.5 | 61.3 | 60.4 | 60.1 |
| | H | 39.2 | 39.9 | **41.2** | 40.1 | 39.5 | 38.2 | 37.6 |

adopted as the default setting, representing an optimal trade-off between model capacity and generalization efficiency.

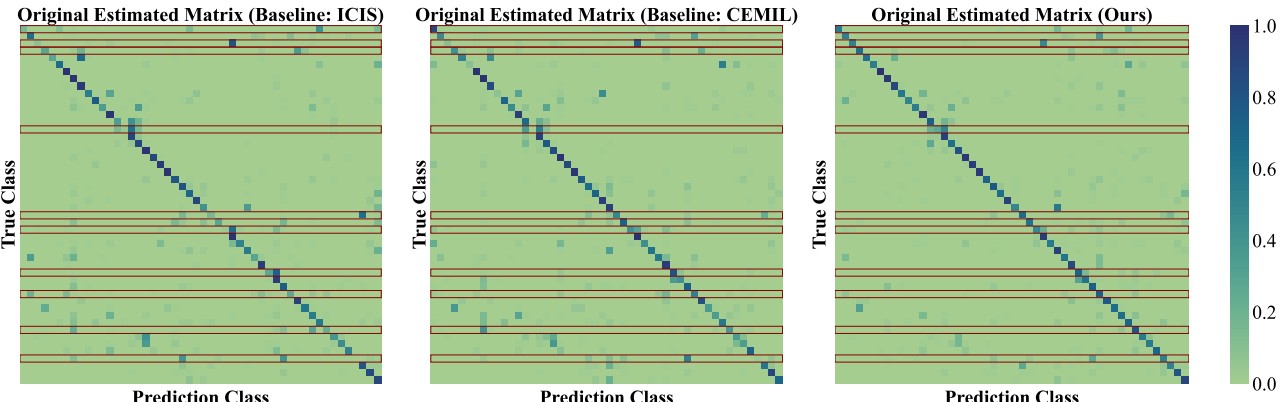

*Figure 9.* Confusion matrix visualization of the predicted distribution of test images on the fine-grained CUB dataset. The red box indicates the top-10 classes (this does not represent confused classes here) that are most likely to be misclassified in subsequent selections.

### E.7. Qualitative Results

To provide an intuitive understanding of the internal mechanisms and discriminative capabilities of the framework, classification behaviors based on confusion matrix and visualizations of clustering and visual features are presented as follows.

**Confusion Matrix and Analysis.**   While overall accuracy and harmonic mean reflect average performance, they do not reveal where classification errors occur, especially among visually similar fine-grained classes. To better understand how the proposed framework improves fine-grained discrimination, this qualitative analysis based on the confusion matrix is conducted in the ZSL setting, focusing on whether our method effectively reduces confusions between unseen classes difficult to distinguish rather than merely improving global metrics. For clarity, this paper first visualizes the full confusion matrix (shown in Figure 9) and then identifies a subset of the most challenging classes based on the baseline ICIS[16]. Specifically, in the prediction distribution matrix of the baseline, the sum of all the off-diagonal elements in the corresponding row for each unseen class is calculated to measure the overall score of classification for that class. Classes are then ranked according to the score, and the top-10 classes that are most prone to misclassification are selected. Based on this, submatrices are extracted corresponding to the same set of classes from the complete confusion matrices of the baselines ICIS and CEMIL, and the method proposed in this paper for comparative analysis. To preserve the original error proportion relationships, the extracted submatrices are not performed for further normalization. The results on the CUB dataset are shown in Figure 10.

Figure 9 shows the overall classification performance of the different methods, mainly reflected in the concentration of elements on the diagonal and the off-diagonal errors. Compared to the baselines, our method significantly suppresses these off-diagonal errors while continuously enhancing the diagonal elements, thereby improving the classification accuracy for each class. Figure 10 further confirms this point, but it does not fully illustrate the refinement of the decision boundaries for class discrimination achieved by our method. Therefore, this paper provides a substantive definition of confused classes.

**Definition E.1** (**Confused Classes**). Basic visual features can be captured, but misclassification often occurs on shared visual features. A generalized set of confused classes $C$, determined by a confidence threshold $\theta$, can be defined as follows:

$$C = \{c \,|\, \underbrace{((\hat{c} \neq c) \wedge (\Delta p < -\theta))}_{\text{A serious misclassification}} \vee \underbrace{(|\Delta p| \leq \theta)\}}_{\text{Confusion}}, \quad \text{where } c \in Y \text{ and } \Delta p = p(c|x) - \max_{y \in Y, y \neq c} p(y|x). \quad (61)$$

Among them, $Y$ represents the set of classes, $\hat{c}$ represents the predicted class denoted as $\hat{c} = \arg\max_{y \in Y} p(y|x)$, $x$ represents a sample from a true class $c$, and $p(y|x)$ represents the probability distribution on all possible classes.

To further highlight the effectiveness of our method at the classification boundaries, this paper presents a confusion analysis based on Definition E.1, where serious misclassifications have already been considered above and only the confusion is considered here. According to $|\Delta p| \leq \theta$ in Equation (61), it is divided into positive and negative confusion, where $\theta$ is set to 20%[17]. Using baseline CEMIL as the benchmark for confused class statistics, comparative results are shown in Figure 11.

---

[16]The confusion matrix is obtained by statistically analyzing the prediction distribution of images to approximate the error probability.

[17]Here, positive class confusion and negative class confusion refer to $0 \leq \Delta p \leq \theta$ and $-\theta \leq \Delta p < 0$, respectively.

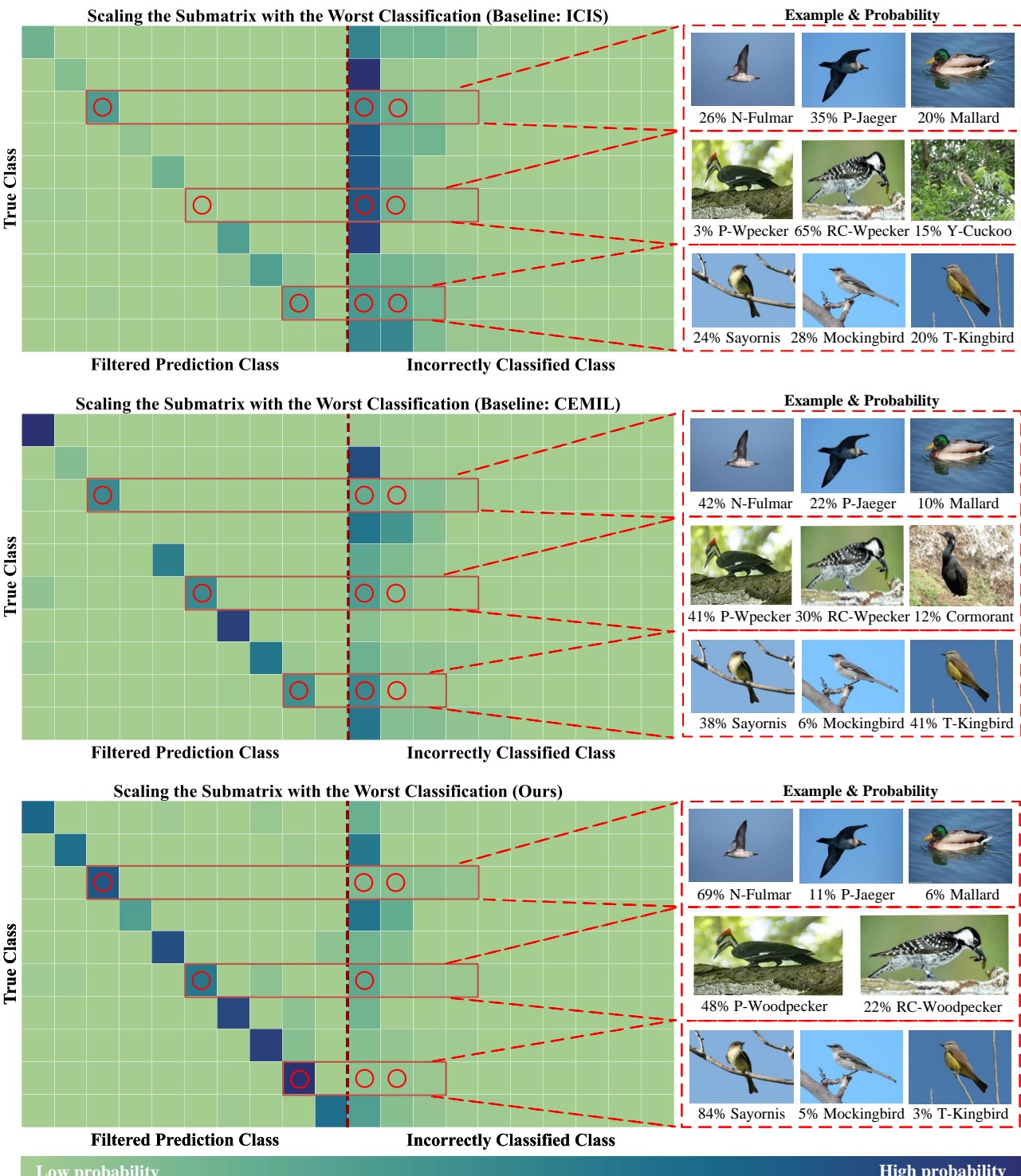

*Figure 10.* Detailed visualization of the top-10 classes most likely to be misclassified under the ZSL setting. The left part illustrates the selected submatrices, comprising the top-10 classes and scores assigned to other incorrect classes. The right side displays specific class samples, which are similar to each other and easily mistaken for one another (including misclassification and confused classes).

It can be observed that there is a certain similarity between the true class and confused classes, and they are highly susceptible to interference from background factors. The proposed method, compared to the baseline CEMIL, not only improves the accuracy of the target class but also suppresses interference concentrated in known visually similar confused classes, which

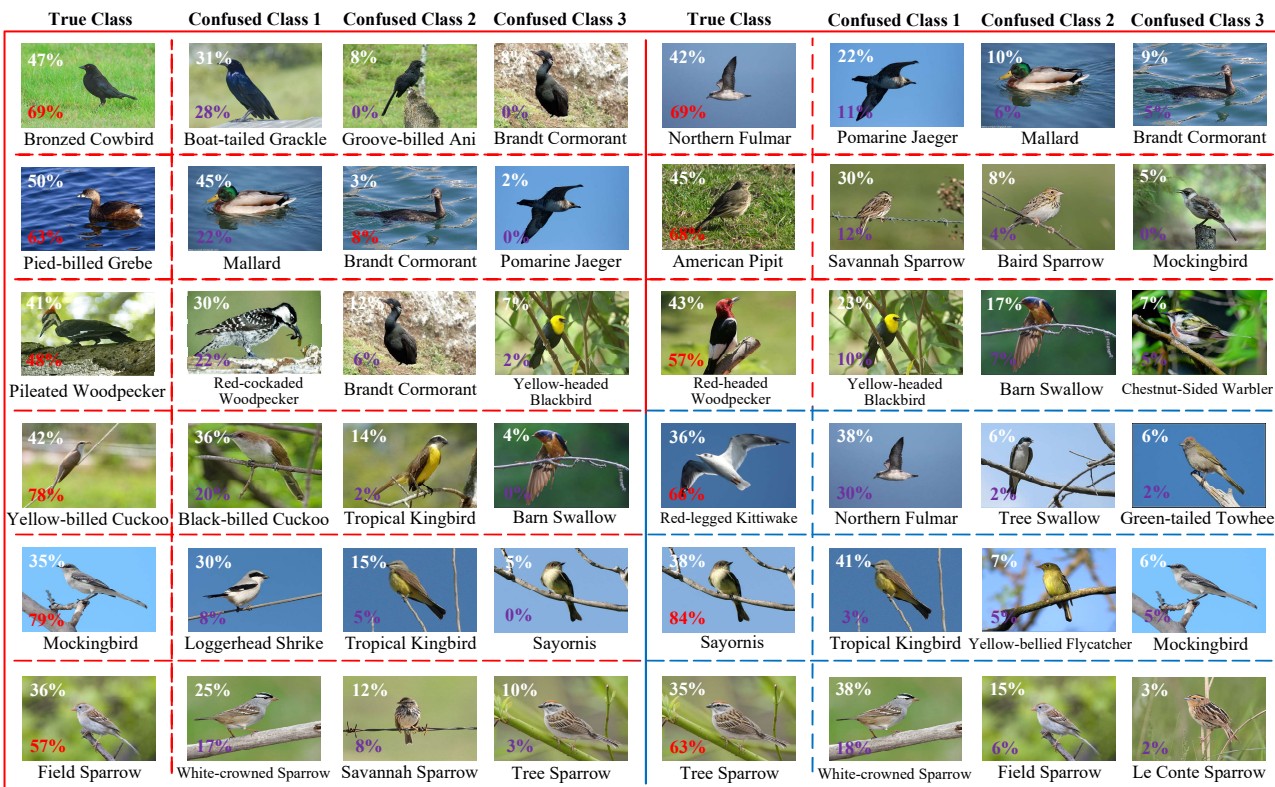

*Figure 11.* Confusion analysis between our method and CEMIL, where the three most representative confused classes are listed for each class. The red and purple values in the figure indicate the improvement and decline of our method compared to the baseline (the white values). The red border represents the positive confusion, while the blue border denotes the negative confusion.

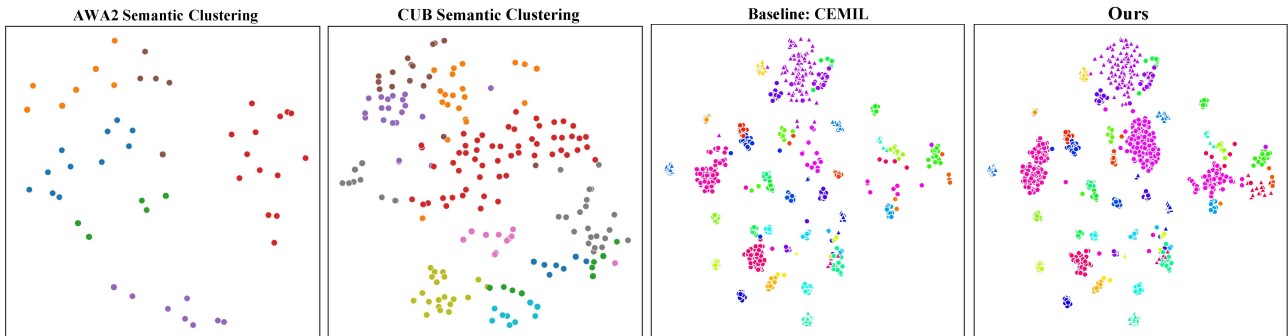

*Figure 12.* Visualizations with t-SNE embeddings for semantic clustering (left two columns) and visual features of samples (right two columns) in GZSL, where the former is shown on the AWA2 and CUB, while the latter compares with the baseline CEMIL on the CUB.

suggests that the structure-aware routing mechanism and causal invariance constraint effectively guide the synthesized classifiers to focus on invariant and discriminative visual features. Therefore, the model achieves clearer class separation and more stable decision boundaries, rather than relying on the shared mapping and superficial semantic correlations. Overall, this analysis provides qualitative evidence that the performance gains of our method stem from meaningful reductions in fine-grained confusions, corroborating the improvements observed in quantitative evaluations.

**t-SNE Visualization of Semantic Clustering and Visual Features.** To validate the effectiveness of the proposed framework in refining the semantic and feature space, t-SNE (Maaten & Hinton, 2008) is employed to visualize semantic clustering and visual features. Specifically, this paper applies t-SNE to the partitioned coarse-grained semantic clusters, followed by L2 normalization, and the results are shown on the left side of Figure 12. To validate how our method refines classification boundaries, the right two columns show the feature visualizations of correctly predicted image samples under different methods, where the prediction accuracy at the class boundaries indirectly reflects the clarity of these boundaries.

*Table 14.* Partial failure mode analysis derived from non-visual symmetry constraints on the CUB dataset.

| True Class | Confused Class 1 | Confused Class 2 | Descriptions |
|---|---|---|---|
| 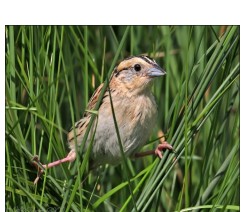 | 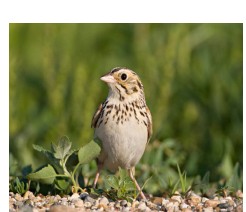 |  | (True Class: **27%**) **Henslow Sparrow** Small and flat-headed sparrow with greenish-yellow hue, dark streaks, and short tail. (Confused Class 1: **50%**) **Le Conte Sparrow** Tiny sparrow with buffy-yellow face, streaky sides and back, a pale belly, and flat forehead. (Confused Class 2: **23%**) **Baird Sparrow** Small sparrow with distinctive streaked back and a pale, buffy face with fine streaked breast. |

It can be observed that the classes are not randomly assigned, but instead exhibit clear biological and semantic patterns. Specifically, visually similar species, e.g., *Laysan Albatross* and *White Pelican*, are spontaneously routed to the same adapter cluster, focusing on shared attributes such as "webbed feet" and "large wingspans". Conversely, distinct species like *Sparrows* and *Woodpeckers* are aggregated into a separate cluster. This confirms that the framework effectively decomposes the complex fine-grained task into manageable subproblems based on the intrinsic semantic hierarchy, validating the effectiveness of the proposed framework without requiring explicit hierarchical supervision. Regarding the distribution of correctly predicted sample features, our method shows tight clustering within classes and a higher number of correct samples between classes, which indirectly demonstrates its discriminative ability for confused classes.

### E.8. Case Study

To further dissect the behavior of our model beyond statistical metrics, a qualitative case study is conducted to visualize specific instances of successful recognition and typical failure modes. As illustrated in Figure 11 and Table 14, representative samples are selected to analyze the reasoning mechanism under challenging conditions.

**Analysis of Success Patterns.** To illustrate how our framework achieves robust recognition, the classification of *Sayornis* shown in Figure 11 is analyzed, comparing our method against the baseline CEMIL. The performance leap of our method is not attributed to a single module but stems from the synergistic collaboration between the structure awareness and causal intervention. The primary cause of the baseline's failure is that these are visually and semantically similar classes, whose subtle attributes are overwhelmed by general attributes. The secondary cause is the reliance on spurious environmental cues. As shown in the sample images, both *Sayornis* and *Tropical Kingbird* share almost all attributes except for subtle differences in plumage color, and appear in similar contexts, e.g., "perched on branch". By introducing adapters and causal invariance, the model focuses on subtle discriminative attributes and ignores non-causal background factors.

Although causal intervention ensures the model looks at the right object, this is insufficient because the visual details of the classes themselves are not being considered. Therefore, the introduction of adapters focuses on the capturing fine-grained details, such as the distinct grayish-brown upperparts of *Sayornis* versus the bright yellow plumage of *Tropical Kingbird*, which might be overlooked when using only the shared mapping. The success is a result of this correction, that is, the causal mechanism pulls the projection away from the background-dominated manifold, and the adapter pushes it precisely into the correct fine-grained subspace, effectively separating *Sayornis* from its confused neighbors.

**Analysis of Failure Modes.** Despite our framework effectively mitigates confusion caused by spurious contextual correlations, it encounters limitations when handling asymmetric visual containment between classes. This failure mode can be defined as a scenario where a dominant class *absorbs* the predictions of a subordinate class, leading to one-way confusion, a significant example of which is provided in Table 14. When the input image is a *Henslow Sparrow*, the model exhibits low confidence in the true class (27%) and misclassifies it as *Le Conte Sparrow* (50%), with *Baird Sparrow* also taking a share (23%), which suggests that the true identity is drowned out by the confused classes. Conversely, when the true class is a *Baird Sparrow* or *Le Conte Sparrow*, the model correctly identifies them with high confidence. This asymmetry suggests that *Le Conte Sparrow* serves as a visual super-set relative to *Henslow Sparrow*, where the later is visually characterized by features, e.g., specific streaks, which are essentially a subset or a less distinct differences of the features found in the former. Exploring approaches such as introducing negative feature learning or inter-class intervention to highlight distinguishing features between classes are promising directions, which are discussed in detail in Appendix F.

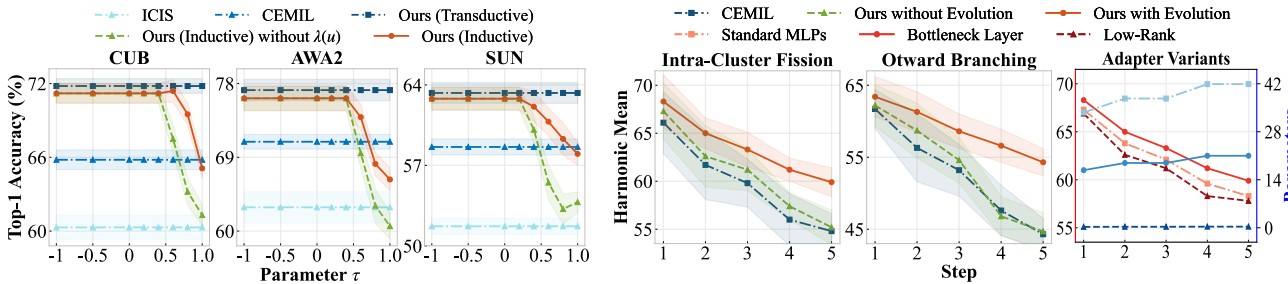

*(a)* Comparison of method effectiveness with baselines (ICIS and CEMIL) under the transductive and inductive scenarios.

*(b)* Comparison of effectiveness with CEMIL, and the harmonic mean (red) and efficiency (blue) of adapter variants under continuous scenarios.

*Figure 13.* Validation of the effectiveness of our framework across three different scenarios and adapter variants for incremental scenarios.

## E.9. Continuous Class Expansion in Extended Scenarios

To validate the practicality and robustness of the framework in real-world environments, this experiment extends the evaluation to the three scenarios constructed in Appendix C.5, that is, transductive, inductive, and continuous expansion.

**Transductive and Inductive Scenarios.** The adaptability of our model under different data availability assumptions is first evaluated. In the scenario of transductive settings, the semantic descriptions of unseen classes are accessible during training and the CSD module is performed on the union of all semantics $S \cup U$, ensuring that the learned clusters cover the entire manifold. As expected, the model achieves the highest performance because the routing priors are globally optimized, consistent with the performance of the proposed framework in previous experiments. In the strict inductive setting, unseen semantics are unavailable during training, requiring to distinguish between proximal new classes (similar to existing clusters, $s_{max} \geq \tau$) and distant OOD classes ($s_{max} < \tau$). Experiments are conducted on unseen classes across each dataset to validate the robustness of our framework, which can be achieved by changing the confidence threshold $\tau$ to indirectly control the semantic distance between unseen classes and existing clusters[18]. Figure 13a formally shows the two scenarios and the assignment method described above, suggesting that for semantically distant classes, the standard adapter refinement degrades performance compared to the global backbone. However, by incorporating $\lambda(u)$ into the weight synthesis derived from Equation (25), the model successfully suppresses erroneous residuals from irrelevant adapters and safely reverts to the global backbone. This demonstrates that the framework can handle OOD semantics and ensure no excessive degradation.

**Continuous Evolution.** Preliminary experiments on class incremental learning have been conducted in Section 4.6 of the main text, based on the scenario that the clusters learned by the seen classes are sufficient to cover the semantic space of the unseen classes. However, a complete framework needs to continuously evolve through incremental learning and be able to handle with intra-cluster fission and outward branching. Therefore, two distinct experimental strategies are constructed based on the semantic topology of the datasets, where novel concepts may either be fine-grained variations of existing knowledge or entirely new semantic structures. Specifically, based on the semantic clustering properties shown in the previous experiments, the datasets are re-partitioned here to simulate two experimental strategies. Taking CUB as an example, the first strategy involves selecting half of semantic classes as seen classes (still maintaining the original clustering properties) and half as unseen classes for clusters containing a large number of classes, while randomly selecting a small number of classes from the other clusters as unseen classes, maintaining a total of 120 seen classes and 80 unseen classes. Similar to the previous settings, the unseen classes are divided into five sequential and disjoint tasks, and the experimental results are shown in Figure 13b using the bottleneck layer. The second strategy treats all classes within some clusters as unseen classes, again maintaining the setting of 120 seen classes and 80 unseen classes. The experimental results are reported in Figure 13b. The results for both strategies demonstrate the effectiveness of our framework in addressing structural changes that may occur during incremental class learning, further illustrating its robustness and adaptability.

**Efficiency of Adapter Variants in Incremental Scenarios.** To balance accuracy and training efficiency in the incremental scenario (scenario 3), this experiment further evaluates the performance of three adapter variants: the standard MLPs, the bottleneck layer, and the low-rank adapter, focusing on their trade-offs in accuracy and parameter efficiency. Figure 13b

---

[18]The parameter $\tau$ is determined in practical applications based on the maximum distance between the cluster center and the sample points belonging to that cluster across all existing clusters. Here, it is treated as a variable to simulate relative distance scenarios.

illustrate the changes in harmonic mean and cumulative storage cost as the number of clusters increases under the strategy of the intra-cluster fission. It can be observed that although the low-rank adapter does not perform as well as the standard MLPs and bottleneck layer, it still provides a positive contribution to the selection of discriminative attributes. For MLPs or bottleneck layer, the computational cost increases significantly after each fission, leading to excessive memory usage and slower inference in long-term continuous learning. In contrast, the growth curve for low-rank adapters is almost flat, allowing the system to accommodate more new semantic clusters and making it a viable solution for continuous expansion.

## F. Limitations and Further Discussions

**Dependency on LLM Knowledge Quality.** A primary limitation of the proposed framework lies in its inherent reliance on the pre-trained knowledge base of LLMs. While the introduction of the intervention effectively filters observational noise, the ultimate discriminative quality of the semantics is bounded by the factuality, coverage, and granularity of the generated knowledge, especially for long-tail classes whose canonical attributes are not represented in general LLM priors. For example, LLMs may generate generic or repetitive descriptions for scientifically obscure concepts or extremely rare species, leading to less discriminability. In such cases, errors introduced at the semantic generation stage can propagate through the MSG, CSD, and ACS module, leading to structurally plausible yet inaccurate semantics, and finally biased synthesized classifiers. A promising mitigation approach is to augment MSG with Retrieval-Augmented Generation (RAG) over domain-specific knowledge bases and automatic verification to reduce hallucinations and improve long-tail reliability.

**Conflict between Static Semantics and Dynamic Visuals.** The multi-source semantics generated in this paper are primarily intended to explicitly expose noise factors unrelated to the class itself, and then eliminate noise through invariance in the ACS module, thus ensuring the robustness and generalization of the model. However, a gap still exists between these static semantic descriptions and dynamic visual instances. LLM-generated texts typically describe objects in their typical states and cannot cover extreme pose variations, severe occlusions, or atypical viewpoints that may appear in real images. Since most semantic anchors are view-independent, exploring view-dependent semantic generation is meaningful, where auxiliary visual priors condition the prompting to produce context-specific descriptions, making the semantic anchors better aligned with the actual visual distribution. Furthermore, instance-level text-image pairs also represent a potential solution.

**Comparison of Intervention and Counterfactual.** In causal inference, intervention and counterfactual reasoning reside at different cognitive levels. Interventions, typically defined by the do-operator, focus on evaluating the average effect on the overall distribution by actively changing a specific variable in the system. In the semantic generation discussed in this paper, interventions are used to sever spurious correlations between environmental noise and class labels, thereby extracting the semantic core that remains invariant across multiple contexts. In contrast, counterfactual reasoning involves a higher level of instance-level retrospection, not only considering what would happen if a variable were changed, but also exploring how the outcome would have evolved if a specific condition had been different given the observed facts. While counterfactual reasoning has been explored in previous ZSL studies at the feature or correlation level (Wang et al., 2026), our focus is to incorporate interventions directly into semantic generation and weight refinement, so that the synthesized classifiers are guided by intervention-stable causal cores rather than superficial linguistic or contextual variations.

Although the intervention effectively mitigates misclassifications caused by style or context bias, it still proves insufficient in addressing the problem of asymmetrical visual containment derived from Appendix E.8. This failure indicates that global interventions aimed at eliminating noise interference alone are insufficient to resolve the nested dilemma within the fine-grained feature space. Introducing counterfactual reasoning will allow the model to perform attribute stripping simulations, that is, for a specific sample, it can deduce how the bird's classification in the semantic manifold would shift if it lacked this particular subtle stripe, which is a meaningful work for the future. This instance-based counterfactual boundary exploration, by simulating subtle perturbations of visual features, can force the refinement of decision boundaries, thus providing a more targeted approach than simple intervention for distinguishing visually nested classes.

**Generalization to Open-world Scenarios.** Although the proposed framework performs well in classifier expansion for predefined unseen classes, open-set recognition remains an open challenge, that is, test samples may belong to classes outside both seen and synthesized unseen sets. In current zero-shot classifier expansion, due to the characteristics of the datasets, there are transferable features between seen and unseen classes, meaning the classes are not extremely different, and each test instance should be mapped to the known semantic manifold. Although we have discussed countermeasures for the forced misclassification of out-of-distribution (OOD) samples in Appendix C.5 and conducted simulated experiments in

Appendix E.9, the model still requires rigorous testing in open-set and incremental learning scenarios. Therefore, exploring classifier expansion in open-world settings constitutes a meaningful and valuable direction for future work.

# G. LLM Prompt Design

---

**Prompt for Universal Ontology Construction (1)**

**Research Domain:** You are an expert with years of research in visual observation and classification for [domain], with a focus on analyzing and distinguishing the visual characteristics of the classes from images.
**Task:** The [domain] includes the following classes: [class_set]. Identify and list [view_num] distinct visual perspectives that are commonly used to scientifically distinguish different classes using a step-by-step reasoning.
**Reasoning:** First, consider the visual differences between various subclasses in this domain and determine which visual features vary the most (e.g., the predator–prey contrast). Second, convert specific features into generalizable perspectives (e.g., instead of 'red feathers', please use 'fur color'). Third, remove subjective (e.g., beautiful), non-visual (e.g., loud call), or overly generic attributes (e.g., has object).
**Output:** Output the final verified perspectives, where the output format should follow this structure: [Coarse-grained dimension]-[Fine-grained view] (e.g., [Physical traits]-[Body shape]).

---

**Prompt for Attribute Generation and Categorization (2)**

**Research Domain:** You are an expert with years of research in visual observation and classification for [domain], with a focus on analyzing and distinguishing the visual characteristics of the classes from images.
**Task:** The [domain] in this batch includes the following classes: [class_set_batch]. Generate [att_num] refined visual attributes for each class and categorize them into a specific ontology, where the set of ontologies includes [ontology].
**Reasoning:** First, retrieve the visual details for each class in this batch. Second, consider the unique visual characteristics of these classes and think about what makes each class look different from the others. Then, within the scope of the ontology set, generate [att_num] precise and visually descriptive attribute phrases (e.g., red feathers), avoiding abstract or non-visual terms (e.g., rare). Finally, strictly map each generated attribute to its most likely corresponding ontology, ensuring that each attribute is associated with exactly one ontology.
**Output:** Return the result as a JSON object where keys are class names, containing a dictionary of ontologies and their attribute lists: {Class_1: {Ontology_1: [Attribute_1, Attribute_2, ... ], Ontology_2: ... }, Class_2: ... }.

---

**Prompt for Canonical Description Generation (3)**

**Research Domain:** You are an expert with years of research in visual observation and classification for [domain], with a focus on analyzing and distinguishing the visual characteristics of the classes from images.
**Task:** Generate a single, coherent, and visually relevant description for each class [class_set].
**Reasoning:** First, the class descriptions are constrained to the set of perspectives [reliable_ontologies], generating attribute concepts for each perspective within each class. Second, using purely descriptive language, convert the generated attributes into fluent sentences without incorporating any stylistic elements or contextual information. Next, review the generated descriptions, removing any subjective adjectives and references to background, ensuring the tone is purely observational. Finally, ensure descriptions are concise and strictly based on the defined perspective.
**Output:** Return the final description and follow the structure: {Class_1: Description of Class_1, Class_2: ... }.

---

**Prompt for Intervention Dictionary Construction (4)**

**Research Domain:** You are an expert with years of research in visual observation and classification for [domain], with a focus on analyzing and distinguishing diverse visual characteristics of the classes from images.
**Task:** Create two modifier lists, linguistic styles and contextual variations, to expand diversity of class descriptions.
**Reasoning:** First, based on the [class_set] within this [domain], preliminarily determine the possible linguistic description styles and environmental contexts. Second, for the style factors, identify [style_high_num] distinct high-level domains; for the context factors, identify [context_high_num] distinct high-level domains. Then, for each domain, brainstorm [low_num] specific and mutually exclusive scenarios. Avoid synonyms. Finally, conduct a diversity review to ensure a balance between common and rare scenarios.
**Output:** Provide results in this structure: {Styles: {High_1: [Low_1, Low_2, ... ], High_2: ... }, Contexts: ... }.

**Prompt for Interventional Semantic Generation (5)**

**Research Domain:** You are an expert with years of research in visual observation and classification for [domain], with a focus on distinguishing and describing diverse visual characteristics of the classes from images.
**Task:** Under environment context [context_set], use language style [style_set] to describe the [class].
**Reasoning:** First, all description generation is based on [canonicity], which includes the visual attributes and is generated in the default style and environment. Second, analyze the rendering properties of styles and the physical properties of environments. Then, based on the [canonicity], the [class] is placed within an [context] and described according to the specified [style]. Finally, the generated description is checked against the [canonicity] to ensure that inherent visual attributes remain unchanged, with any varies being solely due to the style and environment.
**Output:** Return the response in this structure: {Style_1: {Context_1: Description_1, Context_2: ... }, Style_2: ... }.

