# OpenReview forum: "Granularity-Aware Adaptive Classifier Expansion via Zero-Shot Learning"
_ICML.cc/2026/Conference — ICML 2026 regular_

### Official Review · Reviewer_q8pY · 2026-03-02

**Soundness:** 3
**Presentation:** 3
**Significance:** 3
**Originality:** 3
**Overall Recommendation:** 4
**Confidence:** 3

**Summary:**

This paper addresses the challenge of image-free zero-shot classifier expansion, where classifiers for novel classes are synthesized purely from semantic information without relying on visual exemplars. Authors argue that employing a single, shared semantic-to-weight mapping struggles when the expansion process mixes visually distinct and highly similar classes, particularly in continuous arrival scenarios. To solve this, the authors introduce a three-step pipeline. First, MSG uses LLMs to generate and filter semantic descriptions. Next, CSD clusters the classes based on how easily they are confused. Finally, ACS trains a separate adapter for each of these clusters rather than relying on one global mapping. The method is evaluated across standard zero-shot learning benchmarks and a continuous class-incremental expansion setting, demonstrating consistent performance gains and improved stability.

**Compliance With Llm Reviewing Policy:**

Affirmed.

**Final Justification:**

The rebuttal adequately addresses my main concerns. In particular, the authors now clarify the intended role of the SCM and make the intervention claim more concrete. The rebuttal also strengthens the robustness claim of CSD with meaningful stability evidence, including ARI/NMI across seeds and incremental steps, reassignment rates, and discussion of when structural updates occur. My remaining concerns are mainly about overall system complexity and broader practical trade-offs, which I view as paper-level limitations rather than unresolved rebuttal-level issues.

**Key Questions For Authors:**

1) For each baseline evaluated in the main tables, what exact semantic inputs were used (e.g., expert attributes vs. ConceptNet vs. your LLM descriptions)? Can you provide results where the strongest baselines are fed the exact same semantic embeddings produced by your MSG module?

2) What concrete nuisance factors are explicitly targeted by the intervention step? Can you provide a controlled ablation demonstrating that the intervention formally removes specific noise modes, rather than just acting as a generic prompt-ensemble variance reducer?

3) How sensitive is the overall accuracy to the number of clusters and the chosen clustering hyperparameters? Furthermore, how stable are the discovered partitions across continuous expansion steps?

**Limitations:**

While the impact statement appropriately notes that relying on semantic embeddings can inherit linguistic biases, authors could explicitly discuss the framework's dependence on specific semantic sources (LLM API variability, prompt drift) and acknowledge how disparities in semantic input richness currently complicate direct comparisons with legacy baselines.

**Strengths And Weaknesses:**

Strengths:
- The granularity-aware formulation targets a highly relevant and realistic failure mode in ZSL. Authors demonstrate that distinct versus similar classes impose conflicting optimization demands, and why a monolithic mapping allows gradients from easy/distinct classes to dominate.
- The overall pipeline is sound and interpretable at the module level. Furthermore, the presentation is strengthened by implementation details and the appendix provides useful operational transparency regarding LLM/embedding model interchangeability, inference token costs, and training runtimes.
- Beyond the standard datasets, the paper explicitly evaluates a continuous class-incremental protocol. Observation that performance gaps widen favorably as more classes are added strongly supports the authors' claims regarding stability under continuous expansion, closely aligning with practical deployment needs.

Weaknesses:
- The paper uses causal terminology to describe its semantic denoising process. However, it is not clear what formal structural causal model assumptions justify interpreting these steps as true causal interventions rather than simply sophisticated prompt-ensembling and filtering.
- Although the paper evaluates an LLM-generated semantic setting for both the proposed method and baselines, the proposed pipeline appears to produce richer and more curated semantic representations than what most baselines consume. As a result, it remains somewhat unclear how much of the gain comes from the granularity-aware architecture vs from higher-quality semantic inputs produced by MSG.
- The method relies on CSD to discover meaningful confusion structure and to route classes into clusters that drive the adapter-based synthesis. While the paper includes sensitivity analysis and ablations comparing routing/clustering variants, it would further strengthen the claim of robustness to report stability of the learned partitions across seeds and across incremental steps, and to quantify how often clusters split/merge, and whether such shifts correlate with performance.

---

> ### Author Rebuttal · Authors · 2026-03-30
>
> We sincerely thank you for your thoughtful feedback. Here are our responses.
>
> **Q1 & W2 & L:** In the main paper, Table 1 reports results under two semantic settings. In the expert setting, all methods use expert attributes only. In the LLM-generated setting, our method uses MSG, while the baselines use mean-pooled multi-view descriptions from CEMIL. Table 2 uses Wiki2Vec attributes for baselines, while ours still uses MSG.
>
> As discussed in the attribution analysis for **Reviewer Yf2c, Q2 & W2**, richer semantics are still constrained by the shared mapping, whereas CSD+ACS changes the synthesis mechanism from semantics to weights, which is the true source of the gain. For a fairer comparison, we further evaluate baselines using MSG (in the table below).
>
> |Dataset|CUB||AWA2||SUN||
> |-|-|-|-|-|-|-|
> |Method|T1|H|T1|H|T1|H|
> |ICIS+MSG|63.1|59.4|62.6|58.2|57.2|33.6|
> |CEMIL+MSG|66.5|60.9|72.7|63.3|58.4|38.6|
> |Ours|71.8|62.5|77.2|68.8|63.3|41.2|
>
> **Q2 & W1:** We model semantic generation as $T=F(A,E,G)$, where $A$ denotes class-intrinsic attributes, and $E$ and $G$ denote two concrete nuisance factors: environmental context (e.g., *natural*, *wild*) and linguistic style (e.g., *scientific*, *distinctive*). In our method, the SCM is used as a structured conceptual tool to distinguish stable class attributes from nuisance factors. Under this formulation, the intervention notation changes $E$ and $G$ while keeping $A$ fixed, thereby generating a semantic set.
>
> A standard prompt ensemble only increases description diversity, but it does not specify which variations are nuisance factors, nor does it constrain the model to treat them as irrelevant to class identity. Thus, the additional prompts may simply introduce more semantic variance, without separating class-intrinsic information from style- or context-induced noise. Our intervention explicitly manipulates predefined nuisance variables and links the resulting variations to the downstream invariance objective. Therefore, it is not just generating more prompts, but exposing specific nuisance modes for suppression (in the table below).
>
> |Dataset|CUB||AWA2||SUN||
> |-|-|-|-|-|-|-|
> |Method|T1|H|T1|H|T1|H|
> |Canonical description only|68.2|61.0|70.5|63.8|59.5|37.9|
> |Generic prompt ensemble|69.4|61.6|71.9|65.1|60.7|39.1|
> |Style intervention only|69.8|61.5|73.8|66.3|61.2|39.3|
> |Context intervention only|70.2|62.1|75.1|67.9|62.5|40.8|
> |Ours|71.8|62.5|77.2|68.8|63.3|41.2|
>
> **Q3 & W3:** We first discuss the estimation of $K$ and $\\{\mu_k,\kappa_k,\pi_k\\}$ for reproducibility. We adopt a simple data-driven strategy that estimates $K$ directly from semantics $F'$, without repeated end-to-end training. Specifically, we search over $\mathcal{K} = [\lfloor \sqrt{|Y|}/2 \rfloor, \lceil \sqrt{|Y|} \rceil]$ based on the total number of classes $|Y|$, then perform offline spherical clustering, and choose the $K$ with the highest cosine-aware silhouette score (Equation (26)). This greatly reduces tuning cost and the estimated values are **10 (0.1537), 6 (0.0874), and 23 (0.0476)** for CUB, AWA2, and SUN, respectively. Based on these estimates, we perform only a local adjustment and select the final $K$. Once $K$ is estimated, $\{\mu_k,\kappa_k,\pi_k\}$ are initialized by EM-based clustering and then jointly optimized end-to-end (see Appendix D.1), which reduces cross-scenario adaptation cost.
>
> Our framework remains stable under local adjustment in $K$. Since CSD provides soft priors rather than hard partitions, and ACS does not rely on any single cluster in isolation, small changes in $K$ only lead to smooth routing adjustments rather than abrupt changes. Moreover, modest variations in $K$ usually affect only the partition granularity, not the main neighborhood relations among classes. Therefore, when the relative geometry is preserved, CSD+ACS remains stable (Figure 5). To quantify partition stability across random seeds, we run the model with 10 seeds and report ARI/NMI (mean ± std over all pairs in the table below).
>
> For stability in incremental steps, we compare consecutive partitions on the shared old classes using ARI and NMI, and measure the proportion of previously seen classes whose cluster assignments change after each expansion step (in the table below).
>
> |Dataset|CUB|AWA2|SUN|
> |-|-|-|-|
> |ARI (seed)|0.71±0.03|0.54±0.09|0.47±0.06|
> |NMI (seed)|0.77±0.01|0.65±0.06|0.63±0.02|
> |ARI (incremental)|0.82±0.02|0.70±0.08|0.73±0.05|
> |NMI (incremental)|0.87±0.01|0.76±0.06|0.80±0.03|
> |Reassignment (incremental)|0.07±0.03|0.13±0.07|0.09±0.03|
>
> In continuous expansion scenarios, refinements occur only when the current granularity becomes insufficient, while large-scale restructuring remains rare. The intra-cluster fission occurs only in the first and third of five steps, while otward branching only at the initial step in Figure 12(b). Notably, each structural update also reduces the performance drop, indicating that the strategy can discover and adjust the structure effectively.

---

> > ### Author Rebuttal · Reviewer_q8pY · 2026-04-01
> >
> > The rebuttal adequately addresses my main concerns. At this point, my remaining concerns are mostly about final presentation and claim calibration rather than unresolved rebuttal-level issues.

---

> > > ### Author Response · Authors · 2026-04-02
> > >
> > > Thanks again for your consideration. We have revised the manuscript, and updated figures and tables are provided in **Anonymous Link:** https://anonymous.4open.science/r/ICML_anonymous-2235. Below we summarize the main revisions on settings, gain attribution, intervention-related claims, and stability.
> > >
> > > For **Q1 & W2 & L**, we first clarified the settings and added a fairer comparison in **Section 4.2**, then provided a gain attribution analysis in **Section 4.3**, and finally expanded the discussion about the dependence on specific semantic sources in **Appendix F**.
> > >
> > > **Section 4.2.2 Comparable Scenarios**
> > >
> > > Line 361: In the expert-based setting, all methods use expert-constructed attributes only. In the LLM-generated setting, our method uses the multi-source semantics produced by MSG, whereas the baselines use the mean-pooled multi-view descriptions from CEMIL. For fairer comparison, we further evaluate two strongest baselines with MSG inputs.
> > >
> > > Line 371: Notably, even with the same MSG-derived semantic, strong baselines still underperform our method, suggesting that the gains are not due to richer semantics alone.
> > >
> > > **Section 4.3 Ablation Study**
> > >
> > > Line 364: To further clarify the attribution of gains between richer semantics and the granularity-aware modeling, we conduct a more detailed orthogonal attribution analysis, as shown in Table 3.
> > >
> > > Line 380: Richer semantics alone bring an average gain of 3.8%, smaller than the 8.3% gain from granularity-aware modeling, suggesting that the larger improvement comes from CSD and ACS rather than more semantic inputs alone.
> > >
> > > **Appendix F Limitations**
> > >
> > > Line 1974: MSG together with the granularity-aware design mitigates part of the variability from LLM generation (Figure 7(a)). However, API variability and prompt drift still affect semantics and classifiers. Moreover, matched-semantic comparison only partly alleviates fairness concerns, as differences in semantic richness still complicate comparison with legacy baselines. Thus, the results should be interpreted with this caveat in mind.
> > >
> > > ---
> > >
> > > To address **Q2 \& W1** and improve claim calibration, we revised **Section 3.2** to describe the intervention process, and added a controlled ablation in **Appendix E.3**.
> > >
> > > **Section 3.2.2 Interventional Semantic Expansion**
> > >
> > > Line 212: To construct stable class semantics, we adopt a structured nuisance-controlled formulation of semantic generation as a conceptual tool for exposing nuisance-induced variations. Specifically, we model semantic generation as $T=\mathcal{F}(A,E,G)$, distinguishing class-intrinsic attributes $A$ from two predefined nuisance factors: linguistic style $G$ and environmental context $E$. Controlled semantic variants are then generated by varying $G$ and $E$ while keeping $A$ fixed.
> > >
> > > **Appendix E.3 Internal Ablation of the MSG Module**
> > >
> > > Line 1480: Second, to verify that the intervention targets specific noise modes rather than serving as generic ensembling, the interventional expansion is ablated into four variants: 1) canonical description only; 2) generic prompt ensemble; 3) style intervention only; and 4) context intervention only. (Table 8)
> > >
> > > Line 1515: Generic prompt ensembling increases description diversity, but does not identify nuisance factors or constrain them as irrelevant to class identity, thus introducing extra semantic variance rather than separating class-intrinsic information.
> > >
> > > ---
> > >
> > > For **Q3 \& W3**, we added the estimation of semantic cluster granularity in **Section 3.3**, and provided partition stability analysis in **Section 4.4 and Appendix E.6**.
> > >
> > > **Section 3.3.1 Coarse-grained Structural Prior**
> > >
> > > Line 222: Spherical K-means partitions them into $K$ clusters, where $K$ is estimated by a data-driven strategy. Specifically, we search over $\mathcal{K} = [\lfloor \sqrt{|Y|}/2 \rfloor, \lceil \sqrt{|Y|} \rceil]$ and choose the $K$ with the highest cosine-aware silhouette score, followed by only a small local adjustment that remains stable (Appendix E.6).
> > >
> > > **Section 4.4.3 Sensitivity Analysis**
> > >
> > > Line 430: As reported in Figure 5, our framework remains stable under local adjustment in $K$. Since CSD provides soft priors and ACS does not rely on any single cluster in isolation, small changes in $K$ only lead to smooth routing adjustments rather than abrupt changes. Moreover, modest variations mainly affect the partition granularity rather than core neighborhood relations, allowing granularity-aware modeling to remain stable.
> > >
> > > **Appendix E.6 Further Sensitivity Analysis**
> > >
> > > Line 1691: Beyond the sensitivity in Section 4.4, we quantify structural stability across seeds and incremental steps. We use ARI and NMI to compare class partitions over 10 seeds, and measure reassignment ratios on shared old classes across expansion steps. Table 13 shows that the partitions remain reasonably stable, where the low reassignment ratios indicate that structural changes during expansion are local.
> > >
> > > We hope these revisions further address the remaining concerns.

---

### Official Review · Reviewer_D62N · 2026-03-08

**Soundness:** 3
**Presentation:** 2
**Significance:** 3
**Originality:** 3
**Overall Recommendation:** 4
**Confidence:** 3

**Summary:**

This paper studies the problem of image-free zero-shot classifier expansion, where classifiers for unseen classes are synthesized from semantic descriptions without access to visual data. The authors argue that existing methods rely on a shared semantic-to-weight mapping, which fails to simultaneously handle distinct classes (coarse granularity) and similar classes (fine granularity). They attribute this limitation to a granularity mismatch that leads to optimization interference.

To address this issue, the paper proposes a granularity-aware adaptive framework consisting of three main components:

1. Multi-Source Semantic Generation (MSG) that generates diverse semantic descriptions via LLM-based interventions on style and environment to reduce non-causal noise.

2. Class-Level Structure Discovery (CSD) that discovers latent class structure by clustering semantic embeddings and modeling them with a vMF mixture model to produce routing priors.

3. Adaptive Classifier Synthesis (ACS) that synthesizes classifier weights using a shared backbone with cluster-specific residual adapters and a causal invariance constraint.

The proposed framework is evaluated on standard zero-shot learning benchmarks (CUB, AWA2, SUN), demonstrating improved performance over several baselines under both expert-defined attributes and LLM-generated semantics. The authors seek to discuss a notable context where increasing numbers of classes introduce heterogeneous granularity requirements in classifier synthesis. Overall, a central concept presented by the study is that decoupling coarse and fine semantic discrimination through structural partitioning can improve semantic-to-weight mapping for classifier expansion.

**Compliance With Llm Reviewing Policy:**

Affirmed.

**Final Justification:**

The rebuttal addresses my concerns.

**Key Questions For Authors:**

Although the authors conduct several ablation studies on hyperparameters, some aspects appear to be missing. I would like to ask the following questions.

1. **Multi-source semantic generation**

The paper states that **30 multi-source descriptions** are generated for each class. However, this number seems to depend on the sizes of the style set $G$ and the environment set $E$.
- Have the authors conducted any experiments to analyze the sensitivity to the number of descriptions, or to different configurations of $G$ and $E$?

It would be helpful to understand whether the performance is sensitive to these choices.

2. **Parameters in the vMF mixture model**

In Equations (7) and (8), the parameters $\pi_k$, $\kappa_k$, and $C_{d_a}(\kappa_k)$ are introduced, but their specific values or estimation procedure are not clearly described.
- How are these parameters determined in practice?
- Since the number of clusters differs across benchmarks, are these parameters also dataset-dependent?

This could potentially become more complicated in the continuous class-incremental learning setting, where the number of classes grows over time.

Furthermore, I would like to ask some questions regarding to the training detail.

3. **Question regarding semantic fusion ablation (Table 7)**

In Table 7, the semantic fusion ablation includes a variant “w/o attention refinement.”
However, it is unclear whether the global semantic-to-weight training stage also removes the attention refinement module in this setting.
Appendix C.3 seems to indicate that attention refinement is used during training. Therefore:
- Was the global semantic-to-weight model also trained without attention refinement in this ablation or was only the fusion step replaced with mean pooling?

If the latter is the case, the comparison might not be entirely fair.

4. **Use of $\hat{F}$ vs $F$ in different training stages**
Related to the previous question:
- The paper uses the fused representation $\hat{F}$ for global semantic-to-weight training,
- while the mean of multi-source semantics $F$ is used for granular training.

To better leverage the benefits of attention refinement, it seems more consistent to maintain $\hat{F}$ and use the **attention-aggregated semantic representation**, which is the output of equation (6) without MP, instead of raw $F$ in Equations (19) and (20), particularly in the term $\hat{\mathcal{E}}_{\mathcal{T}} (F)$. Could the authors clarify the reason for this design choice?

5. **Design of the causal invariance constraint**

In the causal invariance constraint, two mappings $\mathcal{P}_c$ and $\mathcal{P}_n$ are introduced to separate causal identity and non-causal variations. However, it seems possible to enforce invariance by directly minimizing the distance between different semantic index i, j.
- Have the authors experimented with such a simpler formulation?
- If so, how does it compare to the proposed approach?

**Limitations:**

The authors discuss some limitations of the proposed approach, particularly regarding potential biases inherited from semantic descriptions. However, several additional limitations could be discussed more explicitly.

First, the proposed framework consists of multiple components, including ontology filtering, semantic intervention, clustering-based structure discovery, and cluster-specific adapters. While these components contribute to the overall performance, the resulting pipeline is relatively complex and may introduce challenges in terms of reproducibility and sensitivity to hyperparameters.

Finally, although the method improves performance on unseen classes, the experimental results suggest that the top-1 accuracy on seen classes decreases across several benchmarks. This trade-off between unseen-class generalization and performance on seen classes could be further discussed to clarify the practical implications of the approach.

**Strengths And Weaknesses:**

**Strengths**

- The concept of granularity mismatch between distinct and similar classes is intuitive and well motivated.

- The paper provides a reasonable discussion of the limitations of shared semantic-to-weight mappings, particularly their inability to capture subtle attributes. This issue is further supported by theoretical analysis in Appendix D.2.

- To address this limitation, the authors propose to design robust class semantics and implicitly decouple coarse- and fine-grained relationships among classes through semantic cluster-specific adapters.

- The authors conduct extensive experiments, including ablation studies, which help demonstrate the effectiveness and plausibility of the proposed framework.

**Weaknesses**

- The proposed method contains multiple stages:
(1) ontology filtering and semantic intervention,
(2) semantic clustering, and
(3) cluster routing with cluster-specific adapters.
This pipeline introduces considerable complexity, which raises concerns about reproducibility and sensitivity to hyperparameters.

- Although the authors provide many experiments, several important analyses appear only in the appendix. In particular, the paper claims that the proposed method can better capture subtle attributes, which is one of the main motivations of the work. However, the corresponding qualitative results are presented only in the appendix. Moving Figure 9 or Figure 10 into the main paper could improve the clarity and presentation quality.

- Although the primary goal of the paper is to synthesize classifier weights for unseen classes, the proposed method consistently shows lower top-1 accuracy on seen classes across all benchmarks compared to some baselines. This trade-off between unseen performance and degradation on seen classes deserves further discussion.

---

> ### Author Rebuttal · Authors · 2026-03-30
>
> We sincerely thank you for your thoughtful feedback. Here are our responses.
>
> **Q1:** CSD+ACS mainly relies on the stable consensus and relative structure across multiple descriptions. When $G$ and $E$ are too small, the relevant nuisance modes are insufficiently exposed, and the structural signals available to the downstream modules become weaker. Once their scale reaches a reasonable range, further increasing them mainly enriches the semantics with diminishing returns, rather than changing the downstream mechanism itself. We explore the effect of description quantity and different $(G,E)$ configurations on performance in the table below (AWA2, harmonic mean).
>
> |Desc num|4|9|20|30|42|56|
> |-|-|-|-|-|-|-|
> |H|64.3|66.1|67.5|68.8|68.6|69.1|
> |**(G, E)**|**(3,10)**|**(4,7)**|**(5, 6)**|**(6, 5)**|**(7,4)**|**(10, 3)**|
> |H|67.3|68.1|68.8|68.5|67.9|66.7|
>
> **Q2:** Thank you for this insightful comment on practicality. We discuss the estimation of clustering-related parameters ($K$, $\pi_k$, $\kappa_k$, and $C_{d_a}(\kappa_k)$) in **Reviewer q8pY, Q3 & W3**. These parameters are initialized during $K$ estimation and then optimized in the subsequent end-to-end training, rather than manually tuned for each dataset.
>
> In the continual setting, the method does not rebuild a new vMF model from scratch at every step. Instead, it updates the existing cluster structure and routing, with $K$ adjusted accordingly (Appendix C.5).
>
> **Q3:** Thank you for pointing out this ambiguity. In the "w/o attention refinement" variant, attention refinement is removed throughout global mapping training, rather than replaced by mean pooling only at the fusion step. This makes the comparison fair, since the variant does not benefits from attention refinement during global mapping training. We have revised the manuscript to clarify the setting explicitly.
>
> **Q4:** We use $F$ and $\hat{F}$ at different stages because they serve different purposes. $\hat{F}$ suppresses outlier descriptions and provides a stable class-level semantic consensus, making it suitable for global backbone learning and structure discovery. In contrast, the granularity-training stage is no longer to compress semantics, but to preserve the intra-class semantic diversity introduced by multi-source intervention. Using $\hat{F}$ there would compress intervention-induced instance-level (i.e., style and context) variations too early, thereby weakening the signals needed for fine-grained adaptation and causal invariance. Therefore, $\hat{F}$ is used for stable class-level anchoring, while $F$ is retained for granularity modeling and disentangling intra-class variation.
>
> **Q5:** We consider a direct distance-minimization variant for invariance, where Equation (10) is rewritten as
> $\mathcal{L}\_{dinv} = \sum\_{y \in S} \mathbb{E}\_{i \neq j} d(\mathcal{D}\_W(\mathcal{E}\_{\mathcal{T}}(F\_{y,i})), \mathcal{D}\_W(\mathcal{E}\_{\mathcal{T}}(F\_{y,j}))).$
> This objective enforces consistency among different semantic descriptions of the same class in the output space, without explicit causal disentanglement. Since multi-source semantics within the same class contain both non-causal variations that should be suppressed and complementary semantics that are helpful for class representation, direct minimization fails to distinguish them, and thus tends to over-compress the latent space and weaken fine-grained cues. In contrast, our method preserves semantic diversity while constraining only non-causal variations not to affect the final synthesized classifier. Our disentanglement performs better than direct invariance, while both outperform the variant without any invariance loss (in the table below).
>
> |Dataset|CUB||AWA2||SUN||
> |-|-|-|-|-|-|-|
> |Method|T1|H|T1|H|T1|H|
> |Ours w/o invariance loss|68.7|61.3|72.6|65.3|60.9|38.8|
> |Direct invariance|70.1|61.6|74.8|67.0|61.7|39.4|
> |Ours|71.8|62.5|77.2|68.8|63.3|41.2|
>
> **W1 & L1:** The modules are sequentially connected but functionally decoupled. MSG is an offline preprocessing module, where semantics are reproduced using prompts in Appendix G, after which the global backbone follows the shared mapping. For the reproducibility and hyperparameter sensitivity involved in CSD and ACS, we estimate clustering-related parameters from data and define the structural-evolution threshold by statistical significance rather than dataset-specific heuristics. Please refer to **Reviewer q8pY, Q3 & W3 and Reviewer SbVo, W4 & L3** for details.
>
> **W2:** We have moved qualitative results and further explained why it's effective at capturing subtle attributes.
>
> **W3 & L2:** In GZSL, the goal is to reduce seen-class bias while preserving unseen-class discrimination. Granularity awareness is designed to refine the decision boundaries of newly synthesized unseen classifiers, so the model may become slightly less biased toward seen classes but achieves better unseen-class performance and a stronger overall trade-off, which is more important in evolving real-world settings.

---

> > ### Author Rebuttal · Reviewer_D62N · 2026-04-04
> >
> > The rebuttal addresses my concerns.

---

> > > ### Author Response · Authors · 2026-04-04
> > >
> > > We sincerely appreciate your detailed review and constructive suggestions, which are highly valuable for the improvement of the completeness and clarity of this work.
> > >
> > > In the manuscript, we have strengthened the presentation by making the evidence for subtle-attribute modeling more visible. Meanwhile, we have clarified key design choices and training details to make the pipeline more transparent.
> > >
> > > To facilitate reproduction and further broaden the practical accessibility of the framework, we have organized the updated parameter-selection strategy and integrated it into our existing framework, thereby providing readers with more flexible ways to apply the method. We have also included a small step-by-step case example to guide the implementation and support future extensions.
> > >
> > > We hope that these refinements further enhance the practical value. Thank you once again for the valuable guidance during this review process.

---

### Official Review · Reviewer_SbVo · 2026-03-08

**Soundness:** 3
**Presentation:** 3
**Significance:** 3
**Originality:** 3
**Overall Recommendation:** 4
**Confidence:** 4

**Summary:**

This paper proposes a granularity-aware adaptive framework to address the granularity mismatch and gradient-driven optimization interference issues arising from existing methods' reliance on a shared semantic-weight mapping. The framework employs a Multi-Source Semantic Generation module to generate robust, multi-dimensional class semantics via causal intervention. A Class Structure Discovery module is proposed to mine latent class structures for coarse-grained clustering and soft routing, and an Adaptive Classifier Synthesis module is utilized to optimize weight synthesis using cluster-specific residual adapters and causal invariance constraints, which achieves a balanced modeling of the general attributes of visually significantly different classes and the subtle attributes of easily confused similar classes, effectively solving the challenges of feature homogenization and accuracy decay in scenarios with continuous class expansion. On multiple zero-shot learning benchmark datasets such as CUB, AWA2, and SUN, this method comprehensively outperforms contemporary state-of-the-art methods in both zero-shot classification Top-1 accuracy and generalized zero-shot learning harmonic mean.

**Compliance With Llm Reviewing Policy:**

Affirmed.

**Final Justification:**

My concerns have been adequately addressed.

**Key Questions For Authors:**

NA

**Limitations:**

1. Despite the inclusion of an attribute-ontology filtering mechanism, the illusion problem in LLMs cannot be completely eliminated; incorrect visual attributes can cause bias in classifier weights.

2. The MSG module requires multiple LLM calls, resulting in high inference time and call costs, with significant overhead during large-scale category expansion.

3. Key hyperparameters such as the number of clusters K and the gain threshold of structural evolution need to be manually tuned based on the dataset, and the cost of tuning for cross-scenario adaptation is high.

**Strengths And Weaknesses:**

Strengths
1. This paper introduces a granularity-aware adaptive strategy that decouples the learning objectives of different granularity categories through a paradigm of "coarse-grained structure decomposition + fine-grained adapter optimization."
2. The paper integrates causal intervention into semantic generation and weight synthesis, which effectively alleviates the illusion and spurious correlation problems in LLM semantic generation.

Weaknesses
1. The framework's performance is entirely limited by the LLM's pre-trained knowledge and generation quality. For niche, long-tail categories and species/objects that are insufficiently covered in LLM pretraining, the LLM cannot generate high-quality, fine-grained visual attribute descriptions, directly degrading performance in subsequent modules.

2. Despite the inclusion of an attribute-ontology filtering mechanism, the illusion problem in LLMs cannot be completely eliminated; incorrect visual attributes can cause bias in classifier weights.

3. The MSG module requires multiple LLM calls, resulting in high inference time and call costs, with significant overhead during large-scale category expansion.

4. Key hyperparameters such as the number of clusters K and the gain threshold of structural evolution need to be manually tuned based on the dataset, and the cost of tuning for cross-scenario adaptation is high.

---

> ### Author Rebuttal · Authors · 2026-03-30
>
> We sincerely thank you for your detailed review. Here are our responses.
>
> **W1 & W2 & L1:** Thanks for this insightful comment on semantic quality. Low-quality semantics mainly appear as missing subtle attributes and noise contamination. MSG mitigates this through ontology filtering and interventions, which suppress hallucinated or weakly grounded attributes while preserving stable class identity across descriptions. CSD+ACS further reduces the effect of outlier descriptions on structure modeling and refines cluster-specific discriminative directions using factors that remain stable across descriptions. Therefore, lower generation quality usually reduces semantic completeness rather than directly breaking the framework. When basic inter-class proximity and difference patterns are preserved, the downstream CSD+ACS modules can still extract useful structure and prevent local noise from propagating uncontrollably into the final classifier. As discussed in **Reviewer Yf2c, Q3**, even weaker or open-source LLMs still achieve competitive performance. To further quantify sensitivity, we inject controlled incorrect attributes into MSG (CUB, harmonic mean in the table below).
>
> |Error Proportion|0%|5%|10%|20%|40%|
> |-|-|-|-|-|-|
> |CEMIL|60.7|58.9|56.2|52.5|43.7|
> |Ours|62.5|61.0|59.2|56.9|51.3|
>
> When only a small number of classes are insufficiently covered by LLM pretraining, the structural partitioning and soft routing mechanisms help localize their influence, preventing a few low-quality classes from globally distorting other well-described classes as in a traditional shared mapping. When such under-covered classes become more prevalent, recent studies suggest that external knowledge enhancement is a promising remedy. MSG can incorporate external knowledge (Wikipedia and ConceptNet) by concatenating external embeddings with the multi-source semantics $F$ along the feature dimension. We evaluate this design on three datasets, as shown in the table below. Appendix F also discusses enhancing MSG with RAG.
>
> |Dataset|CUB||AWA2||SUN||
> |-|-|-|-|-|-|-|
> |Method|T1|H|T1|H|T1|H|
> |Ours|71.8|62.5|77.2|68.8|63.3|41.2|
> |+Wiki2Vec|72.0|62.5|78.4|69.3|63.8|41.6|
> |+ConceptNet|72.8|63.4|78.9|79.6|63.6|41.5|
>
> **W3 & L2:** Thanks for raising this concern on inference time and call costs. LLM calls are a one-time offline preprocessing cost for newly introduced classes, not a repeated cost during classifier synthesis or inference. To support large-scale expansion, we adopt the parallel querying with **5 classes per batch**, yielding total generation times of about **15 and 8 minutes** on CUB and AWA2, respectively. Even on the large-scale SUN dataset (717 classes), the total inference time is only about **25 minutes**. Regarding call costs, we allow semantic generation using open-source LLMs with stable gain (**Reviewer Yf2c, Q3**).
>
> **W4 & L3:** Thanks for this excellent suggestion on reproducibility. Since MSG is an offline preprocessing module, the semantics can be reproduced using prompts in Appendix G, after which the global backbone follows the shared mapping. In practice, we focus on how hyperparameters are determined in CSD+ACS. Specifically, we first adopt a data-driven strategy to estimate the number of clusters $K$ and parameters $\\{\mu_k,\kappa_k,\pi_k\\}$. Please refer to **Reviewer q8pY, Q3 & W3** for details.
>
> For continual learning, the structural-evolution hyperparameters $\\{\tau,\mathcal{G}_0,\rho_0\\}$ are defined by statistical significance rather than dataset-specific heuristics.
>
> - $\tau$ measures whether a new class deviates significantly from the current structure. After training, we compute the cosine similarity between each seen class and the center of its assigned cluster, and define $\tau$ using a lower-tail statistic of this distribution like the 10th percentile.
> - $\mathcal{G}\_0$ determines whether a cluster should be split internally. We use the silhouette scores of seen classes under the initial clustering as a reference and define $\mathcal{G}\_0$ using an upper-tail statistic of this distribution like the 75th percentile.
> - $\rho_0$ controls the difficulty of triggering structural evolution. To balance the computational cost of more frequent updates against performance, we set $\rho_0=0.1$ on all datasets, meaning evolution is triggered when 10% of a new batch is not well covered.
>
> We conduct a experiment (CUB, in the table below) and observe only a minor gap relative to the previous tuning strategy. Overall, our data-driven parameter estimation and percentile-based rules improve reproducibility and reduce tuning cost without dataset-specific tuning.
>
> |(Intra) Step|1|2|3|4|5|(Otward) Step|1|2|3|4|5|
> |-|-|-|-|-|-|-|-|-|-|-|-|
> |CEMIL|66.1|61.7|59.8|56.0|54.8||61.7|56.3|53.2|47.6|44.3|
> |Ours without evolution|67.3|62.6|61.2|57.4|55.2||62.2|58.7|54.6|46.8|44.7|
> |Ours with statistical significance|68.1|64.4|62.9|60.3|59.5||62.9|61.0|58.2|54.8|52.5|
> |Ours|68.3|65.0|63.3|61.2|59.9||63.4|61.3|58.6|56.6|54.3|

---

> > ### Author Rebuttal · Reviewer_SbVo · 2026-04-03
> >
> > My concerns have been adequately addressed.

---

> > > ### Author Response · Authors · 2026-04-04
> > >
> > > We sincerely thank you for your valuable suggestions, which have helped us to further improve the quality of this manuscript, especially in making the practical scope, semantic robustness, and reproducibility of the framework more clearly presented.
> > >
> > > In response, we have refined the manuscript by clarifying the behavior under different semantic qualities and making the discussion about how the framework mitigates the impact of low-quality LLM semantic knowledge.
> > >
> > > Furthermore, we have completed the released code package for reproducibility, including the original implementation and the updated parameter-selection strategy. While the framework primarily leverages the automation of LLM-based semantic generation to construct rich class knowledge, inspired by your comments, we also take into account the practical need for broader category coverage. Therefore, we further provide an interface for integrating external knowledge when certain categories are insufficiently covered by the LLM. This makes the framework easier to reproduce and provides a stronger foundation for future extensions.
> > >
> > > We hope these refinements further enhance the practical value and overall presentation of the manuscript. Thank you again for the valuable feedback during this review process.

---

### Official Review · Reviewer_Yf2c · 2026-03-09

**Soundness:** 2
**Presentation:** 3
**Significance:** 2
**Originality:** 3
**Overall Recommendation:** 4
**Confidence:** 5

**Summary:**

This paper addresses the limitations of the "shared mapping" paradigm in zero-shot learning (ZSL) for classifier expansion, particularly in scenarios with a mix of distinct and similar classes. The authors propose a granularity-aware adaptive framework consisting of three core modules: 1) Multi-Source Semantic Generation (MSG), which uses LLMs and causal intervention to produce semantically robust representations; 2) Class-level Structure Discovery (CSD), which learns a coarse-grained class structure via differentiable soft clustering to serve as a routing prior; and 3) Adaptive Classifier Synthesis (ACS), which refines classifier generation using a frozen global mapping and lightweight residual adapters guided by the routing prior, further enforced by a causal invariance constraint. Experiments on several benchmark datasets demonstrate competitive performance against state-of-the-art methods.

**Compliance With Llm Reviewing Policy:**

Affirmed.

**Key Questions For Authors:**

1.	Your framework integrates three main modules. Have you experimented with a significantly simpler baseline, such as using only the augmented semantics from the MSG module fed into a single, well-tuned Mixture-of-Experts (MoE) layer, without the causal separation and soft clustering? If so, what were the results? If not, what do you believe would be the critical obstacle for such a simplified model?
2.	The ablation study (Figure 3) shows incremental improvements leading to the full framework. Can you provide a more nuanced breakdown? Specifically, how much of the final gain can be attributed to the "granularity-aware" concept (the synergy between CSD and ACS) versus simply having "more data" (the multi-source semantics from MSG)? What would the performance be if you fed the averaged multi-source semantics from MSG into a more powerful, single classifier, omitting CSD and ACS?
3.	Your experiments primarily use semantics generated by GPT-4o. If you were to switch to a less capable or open-source LLM (e.g., LLaMA-3-8B), how much would the performance degrade? Would this drop be primarily due to the lower quality of the MSG module's output, or would the downstream CSD and ACS modules also fail to extract useful structure from the impoverished semantics? This is crucial for assessing the framework's utility in resource-constrained scenarios.

**Limitations:**

Yes. Appendix F provides a candid and detailed discussion of several limitations, including the dependency on LLMs, the semantic-visual gap, and challenges in open-world generalization. This is a strong point of the submission. However, the discussion does not fully address the practical limitations arising from the framework's own complexity, such as reproducibility, the difficulty of hyperparameter tuning, and computational costs, which are significant barriers to adoption.

**Strengths And Weaknesses:**

Strengths:
1.	The paper correctly identifies the "granularity mismatch" and "optimization interference" issues in shared mapping approaches for ZSL, which is a valid and insightful observation.
2.	The framework creatively combines concepts from causal inference, soft clustering, and adapter-based tuning to tackle the ZSL problem, representing a non-trivial combination of existing techniques.
3.	The experimental section is comprehensive, covering comparisons with SOTA methods on multiple datasets, extensive ablation studies, and exploratory analyses like continual learning, demonstrating a solid amount of work.
4.	The analysis in the appendix, such as viewing the shared mapping limitation through the lens of ridge regression, is a worthwhile attempt to provide theoretical grounding for the observed problem.
Weaknesses:
1.	The framework is heavily engineered, stacking three major modules and numerous sub-components. This complexity raises a fundamental question: is it truly necessary? The paper fails to convincingly argue why a simpler approach (e.g., improved semantic quality combined with a well-designed, single adapter network) couldn't achieve similar results. The burden of proof for such complexity is not fully met.
2.	The central idea of "granularity-awareness" is obscured by the intricate machinery. With MSG, CSD, and ACS so tightly coupled, it becomes difficult to definitively attribute the performance gains to the core insight versus the extensive engineering. Ablation studies, while present, may not fully capture the synergistic effects in such a complex system, making the core contribution's impact ambiguous.
3.	Key design decisions, like freezing the global semantic encoder, are not deeply justified. While it simplifies training, its potential negative impact on adapting to truly novel semantics is not explored. Similarly, the continual learning scenario relies on multiple heuristically set thresholds (\(\tau, \mathcal{G}_0, \rho_0\)), making the system fragile in practice, and the paper lacks robustness analysis for these parameters.
4.	Although the method outperforms SOTA, the improvements on some datasets (e.g., the H value on SUN in Table 1) are modest. Given the framework's immense complexity, its "cost-performance" ratio is questionable. A reviewer might reasonably ask: is this level of complexity justified for these incremental gains?

---

> ### Author Rebuttal · Authors · 2026-03-30
>
> We sincerely thank you for your insightful comments. Here are our responses.
>
> **Q1 & W1:** We have tried the simplified MSG+MoE baseline, but its performance remains inferior to ours, mainly due to two issues. First, MoE assigns experts directly from dense multi-source semantics, without stable and interpretable structural priors. As a result, the routing is more easily dominated by globally coarse patterns, making it difficult to separate confusing classes and capture subtle differences. Second, although MSG introduces style/environment interventions, MoE cannot explicitly distinguish class identity from non-causal nuisance variations. In contrast, our method introduces soft structural priors for adaptive synthesis, and further combines causal separation with invariance modeling, leading to more stable and interpretable classifier synthesis (methods (D) and (F) in the table below).
>
> |Dataset|CUB||AWA2||SUN||
> |-|-|-|-|-|-|-|
> |Method|T1|H|T1|H|T1|H|
> |(A) single semantic + shared mapping (**Q2**)|59.2|56.4|57.8|54.6|54.4|29.5|
> |(B) single semantic + CSD + ACS (**Q2**)|68.2|61.0|70.5|63.8|59.5|37.9|
> |(C) multi-source semantics + shared mapping (**Q2**)|63.1|59.4|62.6|58.2|57.2|33.6|
> |(D) multi-source semantics +  CSD + ACS (**Q1, Q2**)|**71.8**|**62.5**|**77.2**|**68.8**|**63.3**|**41.2**|
> |Semantic gain ((C-A)+(D-B))/2 (**Q2**)|3.8|2.2|5.8|4.3|3.3|3.7|
> |Granularity-aware modeling gains ((B-A)+(D-C))/2 (**Q2**)|8.9|3.9|13.7|9.9|5.6|8.0|
> |(E) multi-source semantics + powerful shared mapping (**Q2**)|64.8|59.8|65.3|60.3|57.9|34.8|
> |(F) multi-source semantics + MoE (**Q1**)|66.3|60.5|67.1|61.8|58.3|35.2|
>
> **Q2 & W2:** Thank you for this insightful comment on clarifying the attribution of gains. MSG mainly improves semantic coverage and robustness compared with a single semantic input. However, if the shared mapping remains unchanged, the additional semantics are still absorbed into the same synthesis function, whose optimization is mainly driven by general attributes that are easier to fit and more effective for separating distinct classes. Therefore, the model still cannot resolve the problem that general attributes dominate while subtle attributes are overwhelmed.
>
> CSD+ACS changes the synthesis mechanism from semantics to classifier weights, where CSD introduces structural priors to separate confusing classes and ACS further performs cluster-specific refinement to strengthen discriminative directions for confusing classes. Therefore, even if multi-source semantics are fed into a more powerful, single classifier, it still remains a shared mapping in essence. Our orthogonal attribution shows that richer semantics alone bring an average gain of **3.8%**, smaller than the **8.3%** gain from granularity-aware modeling (methods (A) to (E) in the table above).
>
> **Q3:** Both LLMs and embedding models in MSG are replaceable (Appendix E.4). CSD+ACS relies on relative structural relationships between different classes within the semantic space. Specifically, CSD depends on the relative geometry among classes rather than on whether any single description is highly detailed, so it can still extract useful soft priors when lower-quality semantics retain basic inter-class relations. ACS then uses these priors for residual refinement based on the global shared mapping. Therefore, weaker MSG semantics mainly reduce the completeness and richness of details, but still preserve fundamental separation patterns among classes (the table below). Only when semantic quality degrades enough to collapse inter-class structure would the benefits of CSD and ACS drop substantially, as discussed in **Reviewer SbVo, W1 & W2**.
>
> |Dataset|CUB||AWA2||SUN||
> |-|-|-|-|-|-|-|
> |Method|T1|H|T1|H|T1|H|
> |(a) LLaMA semantics + shared mapping|61.2|58.1|59.7|56.1|55.7|31.6|
> |(b) LLaMA semantics + CSD + ACS|69.1|61.3|73.6|64.7|60.4|39.5|
> |(c) GPT-4o semantics + shared mapping|63.1|59.4|62.6|58.2|57.2|33.6|
> |(d) GPT-4o semantics +  CSD + ACS|71.8|62.5|77.2|68.8|63.3|41.2|
>
> **W3:** The frozen global semantic encoder is designed to preserve semantic topology and avoid absorbing adapter-specific behaviors. We discuss its drawbacks in Appendix F. For instance, when the semantic distribution of new classes differs from that of the training set, the method may need to retrain the global backbone.
>
> **W4:** Although the gain in H on SUN is limited, the accuracy on unseen classes is improved substantially, indicating reduced bias toward seen classes. Beyond standard benchmarks, our method also expands more classes under data-scarce settings while maintaining competitive performance, especially in continuous expansion with dense attributes.
>
> **W3 & L:** To improve reproducibility and reduce tuning burden, we estimate the clustering-related parameters directly from the data and define the structural-evolution threshold by statistical significance rather than dataset-specific heuristics. Please refer to **Reviewer q8pY, Q3 & W3 and Reviewer SbVo, W4 & L3** for details.

---

> > ### Author Rebuttal · Reviewer_Yf2c · 2026-04-03
> >
> > The rebuttal addresses my main concerns. I keep my original score.

---

> > > ### Author Response · Authors · 2026-04-04
> > >
> > > We sincerely thank you for your thoughtful feedback, which is highly helpful for further improving the rigor, clarity, and practical relevance of this work.
> > >
> > > Based on the suggestions provided, we have incorporated the clarifications on performance gains, as well as the motivation and complementary roles of the three modules into the main text, which is a crucial step toward highlighting the significance of our work in a more comprehensive manner.
> > >
> > > To further enhance the practical accessibility of this work, we have refined the released code and supporting materials with clearer modular organization and more detailed parameter-selection guidance, making the framework easier to reproduce and use in practice. At the same time, we also integrate previous comparison baselines and empirical settings, enabling readers to more effectively reproduce the methods and utilize them as a foundation for exploring further related tasks.
> > >
> > > We hope these refinements further enhance both the technical soundness and presentation quality of the manuscript. Thank you once again for the valuable guidance during this review process.

---

### Decision · Program_Chairs · 2026-04-30

**Decision:**

Accept (regular)

**Comment:**

This paper studies zero-shot classifier expansion and proposes a granularity-aware adaptive framework with interventions that generates multi-source semantics by intervening on non-causal noise, discovers latent class structure to separate distinct classes, and refines similar classes to synthesize classifier weights that are robust to noise.

After rebuttal, all the reviewers are satisfied with the response of the authors and finally recommended weak accept, acknowledged the insightful observation, sound pipeline and comprehensive experiments. The AC thus would like to recommend acceptance and strongly encourages the authors to revise the final version based on the rebuttal.